# A nutrient responsive lipase mediates gut-brain communication to regulate insulin secretion in *Drosophila*

Alka Singh[1,5], Kandahalli Venkataranganayaka Abhilasha [2,5], Kathya R. Acharya[1,2,4], Haibo Liu[1], Niraj K. Nirala[3], Velayoudame Parthibane[2], Govind Kunduri [2], Thiruvaimozhi Abimannan[2], Jacob Tantalla [2], Lihua Julie Zhu [1], Jairaj K. Acharya [2] ✉ & Usha R. Acharya [2] ✉

Pancreatic β cells secrete insulin in response to glucose elevation to maintain glucose homeostasis. A complex network of inter-organ communication operates to modulate insulin secretion and regulate glucose levels after a meal. Lipids obtained from diet or generated intracellularly are known to amplify glucose-stimulated insulin secretion, however, the underlying mechanisms are not completely understood. Here, we show that a *Drosophila* secretory lipase, Vaha (CG8093), is synthesized in the midgut and moves to the brain where it concentrates in the insulin-producing cells in a process requiring Lipid Transfer Particle, a lipoprotein originating in the fat body. In response to dietary fat, Vaha stimulates insulin-like peptide release (ILP), and Vaha deficiency results in reduced circulatory ILP and diabetic features including hyperglycemia and hyperlipidemia. Our findings suggest Vaha functions as a diacylglycerol lipase physiologically, by being a molecular link between dietary fat and lipid amplified insulin secretion in a gut-brain axis.

Pancreatic β cells synthesize and secrete insulin in response to elevated glucose to maintain blood glucose levels within the physiological range. Alterations in this function perturb glucose homeostasis - insufficient insulin secretion results in hyperglycemia while excess secretion leads to hypoglycemia. Defective insulin secretion is causal in most forms of diabetes. While glucose is the primary stimulus for insulin secretion by the β cells, it is well recognized that dietary proteins (amino acids) and lipids (fatty acids) can amplify glucose-stimulated insulin secretion (GSIS)[1,2]. Amino acids such as arginine, glutamine, and alanine influence GSIS either directly or indirectly through generation of metabolic coupling factors. Among lipids, acute exposure to fatty acids results in a substantial increase in glucose-stimulated insulin secretion while chronic stimulus results in suppression of its secretion[1-6]. Fatty acids are also converted to long chain acyl CoA, which can be metabolized intracellularly to generate lipid signals that can augment insulin secretion[7,8]. There is broad consensus that lipids are important in the amplification of GSIS and that impairment of this process contributes to diabetes pathophysiology[8,9]. However, the source of the lipid pool (dietary fat versus endogenously synthesized), the lipid signals, and the enzymes that play a significant role in amplifying GSIS are not yet well established. This is a challenging endeavor because of the involvement of complex, multi-organ communication networks in the maintenance of glycemic regulation[10,11]. After ingestion of a meal with mixed nutrients, the maintenance of glucose homeostasis requires coordination of digestion, absorption, relay of metabolic cues from other organs, and integration of information at the pancreatic islets culminating in the modulation of insulin response. While direct sensing of glucose by the

[1]Department of Molecular, Cell and Cancer Biology, UMass Chan Medical School, Worcester, MA 01605, USA. [2]Cancer and Developmental Biology Laboratory, National Cancer Institute, Frederick, MD 21702, USA. [3]Program in Molecular Medicine, UMass Chan Medical School, Worcester, MA 01605, USA. [4]Present address: University of Cincinnati College of Medicine, 3230 Eden Ave, Cincinnati, OH 45267, USA. [5]These authors contributed equally: Alka Singh, Kandahalli Venkataranganayaka Abhilasha. ✉e-mail: acharyaj@mail.nih.gov; acharyaur@nih.gov

β cells is central to insulin secretion, these cells also integrate stimuli which include other hormones, secretory factors from metabolic organs, and neuronal signals to tune insulin secretion[10,11].

*Drosophila* is increasingly being appreciated as a model to understand metabolism and explore inter-organ communication[12–18]. Many metabolic pathways are conserved between flies and humans[19–22]. This includes the insulin pathway, which coordinates many physiological functions such as metabolism, growth and longevity[20,23,24]. In *Drosophila* there are eight insulin-like peptides, ILP1-8, of which ILP2 is most closely related to human insulin. ILP2, ILP3 and ILP5 are synthesized and secreted from 14 median neurosecretory cells called the insulin-producing cells (IPCs) in the brain[20,23,24]. Studies have shown that secretion of ILPs from the IPCs is governed by nutritional, environmental and neuronal inputs[20,25–27]. During the larval growth phase, nutritional status is conveyed to the IPCs by peripheral organs, such as the fat body and gut, that perceive nutritional information and then relay it to the brain. Elegant studies have identified factors produced by the fat body and gut that communicate with the IPCs to modulate ILP production and / or release in the larvae to promote growth and development[28–35]. Relatively less is known about mechanisms that maintain normal glucose concentrations in the post growth adult stage[36–39].

In earlier studies, we have shown that the secretory lipases, CG8093 and CG6277, are involved in mediating the survival of *Drosophila* ceramide kinase mutant that has defects in energy metabolism due to accumulation of ceramide[40,41]. Ceramide is a sphingolipid intermediate known to induce diverse stress responses including lipotoxicity in metabolic syndrome that encompasses obesity, dyslipidemia and insulin resistance[42]. Gut specific knockdown of CG8093 in the ceramide kinase mutant resulted in hyperlipidemia, cardiac defects, and accelerated death[40]. While trying to understand the functions of CG8093, we have discovered an unexpected role for this gut lipase in the insulin-producing cells. In the tradition of naming *Drosophila* genes either based on wild type function or characteristics associated with mutant phenotypes, we refer to CG8093 as Vaha, the Sanskrit word for movement. In this study, we show Vaha is synthesized and secreted from the midgut in response to dietary fat and moves to the brain. Here it concentrates in the IPCs, in a process relying on the *Drosophila* lipoprotein, Lipid Transfer Particle (LTP). Vaha participates in fat amplified insulin secretion and thus functions as a nutrient dependent regulator of ILP2 release to maintain glucose homeostasis in the adult fly.

## Results

### Vaha expressed in the gut is enriched in the insulin-producing cells in the brain

Using the binary UAS Gal4 system, we expressed C-terminal GFP tagged Vaha (CG8093) in the gut via the enterocyte specific NP1Gal4 driver. Surprisingly, we detected Vaha GFP in the pars intercerebralis (PI) region of the brain (Fig. 1a). The PI region is a neuroendocrine center consisting of various types of neurosecretory cells including the insulin-producing cells (IPCs) that synthesize and secrete ILP2, ILP3, and ILP5. Leaky expression of the Gal4 driver in other tissues could provide a trivial explanation for the above results. To probe this possibility, we expressed UAS GFP in the gut using the NP1Gal4 driver which would drive the expression of GFP only in tissues expressing Gal4. This did not result in staining of the PI region (Supplementary Fig. 1a). UAS Vaha GFP without a driver was also tested to ensure that the transgene was not leaky and as seen in Supplementary Fig. 1b, there was no expression in the PI region. While the focus of this study is the intense Vaha GFP staining in the PI region, GFP staining was also observed in the subesophageal ganglion area (marked 2 in Supplementary Fig. 1c) and a few additional areas in the brain which would be of interest to characterize in a future study (marked 3, 4, 5 in Supplementary Fig. 1c). We performed Western blot analysis on extracts

prepared from gut and CNS (brain and ventral nerve cord) dissected from flies expressing Vaha GFP using the gut Gal4 driver. Immunoblots confirmed that the lipase was detected both in the gut as well as the CNS when expressed in the gut (Fig. 1b, left panel). Unlike the full-length lipase, when a truncated form of the lipase lacking the signal peptide (VahaΔ30GFP) was expressed in the gut, it was not detected in the CNS by Western blot analysis (Fig. 1b, right panel). This experiment shows secretion of Vaha from the gut cells is important for its detection in the brain. Colocalization experiments with epitope tagged *Drosophila* insulin-like peptide 2 (ILP2HF expressed under its own promoter, also referred to as gd2HF) revealed considerable overlap between Vaha GFP and ILP2HF staining (Fig. 1c). However, signal peptide deleted Vaha was not detected in the IPCs (Fig. 1d). Higher magnification views of the IPC region in Fig. 1c, d are shown in Supplementary Figs. 2a and Fig. 2b. No GFP staining was detected in the ventral nerve cord when Vaha GFP or VahaΔ30GFP was expressed in the gut. Overexpression of Vaha GFP with a second gut driver, Mex1 Gal4, also resulted in its detection in the IPCs (Supplementary Fig. 3a and high magnification view of IPCs in Supplementary Fig. 3b). These results suggest that transgenic expression of the lipase in the gut results in its secretion and concentration in the IPCs in the brain. To assess if endogenous lipase could be detected in the IPCs, we made attempts to raise a high-titer antibody to the lipase but were unsuccessful. We opted for epitope tagging instead and generated transgenic lines expressing C-terminal V5 tagged Vaha under the control of its own promoter. Colocalization experiments with the insulin reporter showed there was considerable overlap of Vaha V5 staining with that of ILP2HF (Fig. 1e and high magnification view of IPCs in Supplementary Fig. 4a). V5 staining was not detected in the ventral nerve cord (Fig. 1e). Western blot analysis of these flies showed the presence of lipase both in the gut and head extracts (Supplementary Fig. 4b). A similar construct lacking the signal peptide (Vaha V5Δ30) was expressed in the gut, however, it could not be detected in the head extracts (Supplementary Fig. 4b). These experiments were done in a wild type background that contained unlabeled endogenous Vaha. Collectively, these results indicate that Vaha is a gut secretory lipase that can concentrate in the IPCs.

### Vaha is synthesized in the posterior midgut

To test if Vaha is synthesized in the gut or brain or both organs, we evaluated its endogenous transcript levels by quantitative PCR (RT-qPCR) in the CNS and gut samples of wild type flies ($w^{1118}$). When compared to the transcript in the gut, the lipase was not detected in CNS samples (Fig. 2a), head or brain samples (Supplementary Fig. 5a, b). Endogenously tagged *Vaha* (Vaha V5) transcript was also not detected in the brain when compared to the gut (Supplementary Fig. 5c). To further examine if gut is the source of Vaha, we knocked down Vaha specifically in the gut and examined *Vaha* transcript levels in the gut and CNS (Supplementary Fig. 5d). While there was considerable reduction in transcript in the gut RNAi samples, Vaha transcript was not detected in CNS. We also expressed Vaha RNAi using NP1 Gal4 (gut), Elav Gal4 (neurons) and Ppl Gal4 (fat body) drivers and examined Vaha V5 expression in the IPCs. Only gut specific knockdown of Vaha, significantly reduced Vaha V5 signal in the IPCs (Fig. 2b, c, d, e and Supplementary Fig. 6a and b which show all the panels including brightfield, DAPI, Vaha V5, ILP2HF, and merged images).

To characterize the expression pattern of Vaha, we generated a Vaha-Gal4 transgenic line in which 2 kb of the lipase promoter sequence was fused to Gal4. We crossed the Vaha Gal4 line to UAS-mCherry.NLS that expresses nuclear localized mCherry under the control of UAS. Live imaging using this reporter showed mCherry expression in the gut but not in the brain (Fig. 2f). Confocal imaging of fixed tissue showed mCherry staining in the posterior midgut, mostly in the R4 region (Fig. 2g). Antibody staining for Cut protein defined the adjacent acidic R3 region of the midgut (Fig. 2g). We further confirmed

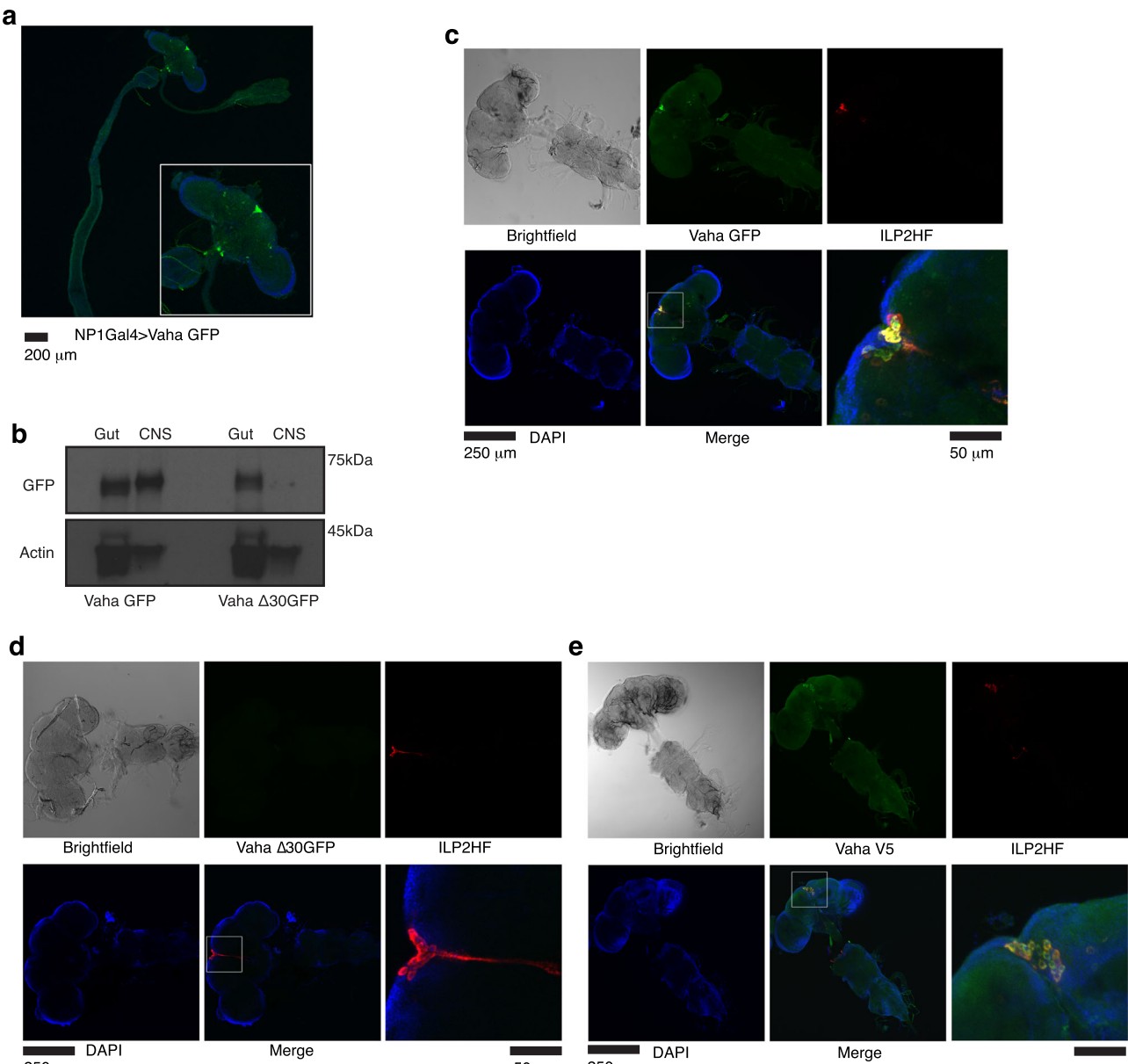

**Fig. 1 | Vaha expressed in the gut concentrates in the IPCs. a** Immunostaining for GFP in the brain dissected with a portion of the gut from 5-7 day old adult flies expressing Vaha GFP in the gut (*w*; NP1 Gal4/+; UASVaha GFP/+). Inset shows 3x magnified image. *n* = 6 guts with brains, scale bar 200 μm. **b** Immunoblotting with GFP antibody of extracts from guts and CNS (brain and ventral nerve cord) from flies expressing Vaha GFP (left panel) or signal peptide deleted Vaha Δ30GFP (right panel) driven by NP1Gal4. Actin is used as loading control. *n* = 3, 7 guts and 7 CNS for each replicate. **c** Immunostaining for Vaha GFP (anti GFP, green) and ILP2HF (anti HA, red) in CNS dissected from 5-7day old adult flies expressing Vaha GFP in the gut and ILP2HF under the control of its own promoter. The genotype is *w*; NP1Gal4/+; UASVaha GFP/gd2HF. ILP2HF (gd2HF) marks IPCs. Nuclei are stained with DAPI (blue). A higher magnification image of the boxed region in the merged panel is shown in the panel to its right, scale bar 50 μm. The confocal images are

projection of z stacks, *n* = 6 CNS, scale bar 250 μm. **d** Immunostaining for Vaha (anti GFP) and ILP2HF (anti HA) in CNS dissected from 5-7day old adult flies expressing Vaha Δ30GFP in the gut and ILP2HF under the control of its own promoter (*w*; NP1Gal4/+; UASVaha Δ30GFP/gd2HF). Nuclei are stained with DAPI. A higher magnification image of the boxed region is shown, scale bar 50 μm. The confocal images are projection of z stacks, *n* = 6 CNS, scale bar 250 μm. **e** Immunostaining for Vaha V5 (anti V5, green) and ILP2HF (anti HA, red) in CNS isolated from 5-7day old adult flies expressing Vaha V5 under the control of its own promoter and ILP2HF (*w*; Vaha V5/VahaV5; gd2HF/gd2HF). Nuclei are stained with DAPI (blue). A higher magnification image of the boxed region is shown, scale bar 50 μm. Confocal images are projection of z stacks, *n* = 6 CNS, scale bar 250 μm. Source data for 1b are provided as a source data file.

the gene expression pattern of Vaha by performing Hybridization Chain Reaction (HCR) on wild type gut. As seen in Supplementary Fig. 7, *Vaha* RNA expression was visualized in the R4 region of the midgut in the HCR images. We also performed HCR on different tissues from third instar larvae. *Vaha* RNA expression was detected in the larval gut but not in the brain, ventral nerve cord, wing and eye imaginal discs, fat body and salivary gland (Supplementary Fig. 8a and b). In addition to the midgut, both the adult and larval guts show some

staining in the hind gut region for *Vaha*, the specificity of which requires further exploration (Supplementary Figs. 7 and 8b).

**LTP and LRP1, LRP2 mediate Vaha gut-brain communication**

Thus far, our results suggest that Vaha is synthesized in the posterior midgut and moves to the IPCs in the brain. We next decided to examine components that could be involved in this inter-organ communication. We first tested if Vaha is detected in circulation by

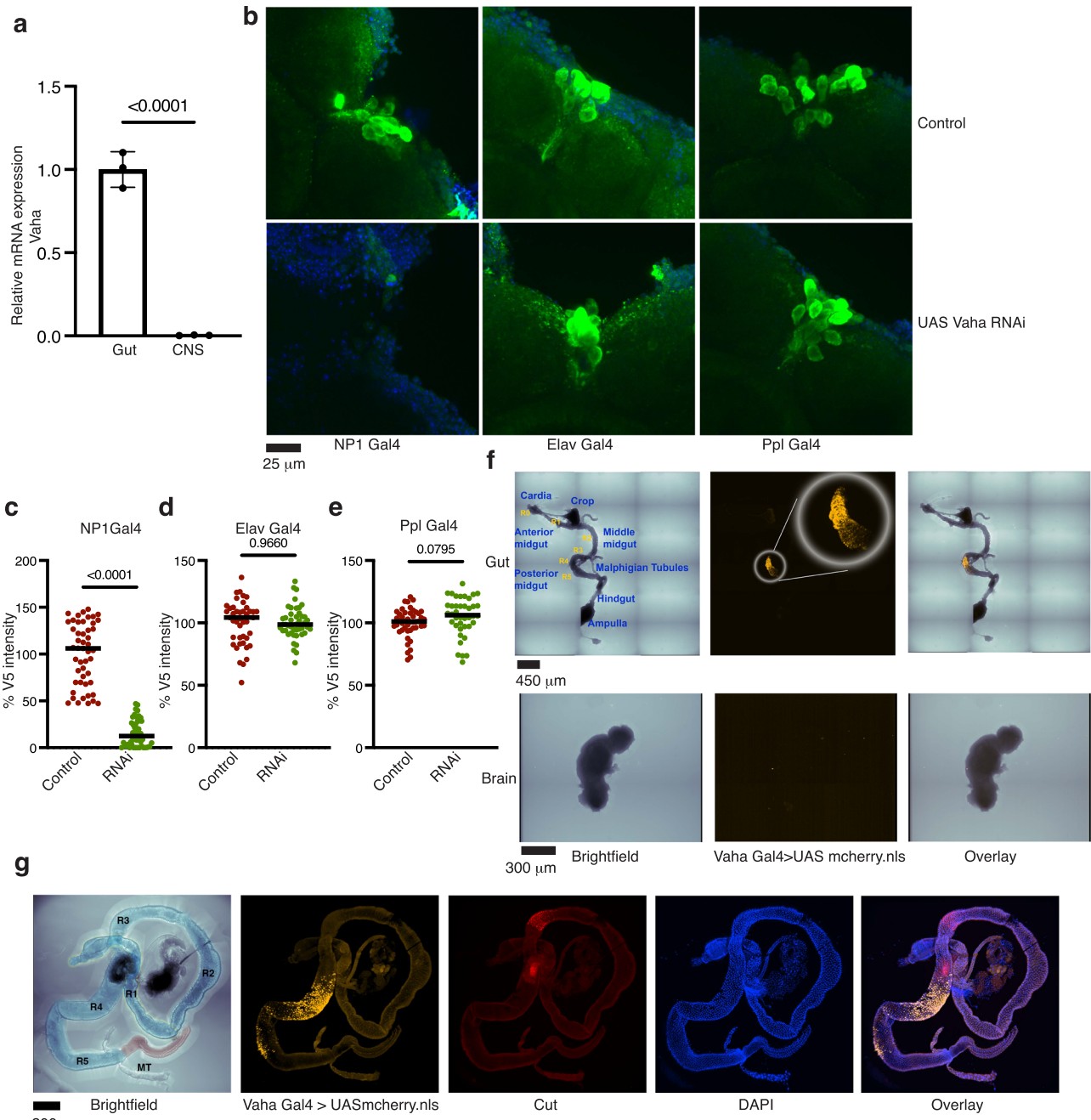

**Fig. 2 | Vaha is synthesized in the gut and not in the brain. a** RT-qPCR analysis of *Vaha* mRNA level in gut and CNS samples isolated from $w^{1118}$. *Vaha* expression is normalized to *GAPDH*. $n = 3$, 10 guts and 10 CNS per replicate. Data is assessed by two-tailed Student's t-test and presented as mean ± SD. **b** Vaha RNAi in the gut ($w$; NP1 Gal4/Vaha V5; Vaha RNAi/gd2HF), neurons ($w$; Elav Gal4/Vaha V5; Vaha RNAi/gd2HF) fat body ($w$; Ppl Gal4/Vaha V5; Vaha RNAi/gd2HF) and imaging of Vaha V5 in the IPCs. Control flies are $w$; NP1 Gal4/Vaha V5; +/gd2HF, $w$; Elav Gal4/Vaha V5; +/gd2HF and $w$; Ppl Gal4/Vaha V5; +/gd2HF. Confocal images are projection of z stacks, $n = 6$ brains, scale bar 25 μm. **c** Quantification of Vaha V5 intensity in the IPCs of Vaha RNAi in the gut. Each data point represents fluorescence intensity in one IPC, $n = 49$ IPCs (control), $n = 46$ IPCs (RNAi). Data is assessed by two-tailed Student's t-test and horizontal line indicates median. **d** Quantification of Vaha V5 intensity in the IPCs of neuronal knockdown of Vaha. Each data point represents

fluorescence intensity in one IPC, $n = 43$ IPCs (control), $n = 45$ IPCs (RNAi). Data is assessed by two-tailed Student's t-test and horizontal line indicates median.
**e** Quantification of Vaha V5 intensity in the IPCs of fat body knockdown of Vaha. Each data point represents fluorescence intensity in one IPC, $n = 46$ IPCs (control), $n = 40$ IPCs (RNAi). Data is assessed by two-tailed Student's t-test and horizontal line indicates median. **f** Live imaging of gut and corresponding brain dissected from flies expressing UAS nuclear localized mCherry driven by Vaha Gal4 driver ($w$; Vaha Gal4/+; UASmcherry.NLS/+). $n = 6$ guts and brains. Scale bar gut 450 μm, brain 300 μm. **g** Immunostaining of guts dissected from above flies. Colored brightfield image highlights the midgut in blue and hindgut in brown. R1-R5 subdivisions of the midgut are marked, MT are malpighian tubules. mCherry is pseudocolored orange while Cut antibody marking R3 is pseudocolored red. $n = 6$ guts, Scale bar 200 μm. Source data are provided for 2a, c, d, e as a source data file.

collecting hemolymph from Vaha V5 flies or flies expressing Vaha GFP via NP1 Gal4 driver and performing Western analysis. Indeed, endogenously tagged Vaha and gut derived Vaha were detected in the hemolymph (Fig. 3a and Supplementary Fig. 9a respectively).

Lipoproteins serve as inter-organ carriers of lipids and lipid-linked morphogens[43]. In mammals, apoA containing HDL mediates cholesterol transport from tissues to the liver, ApoB containing lipoproteins include chylomicrons secreted by the gut, and VLDL and LDL secreted

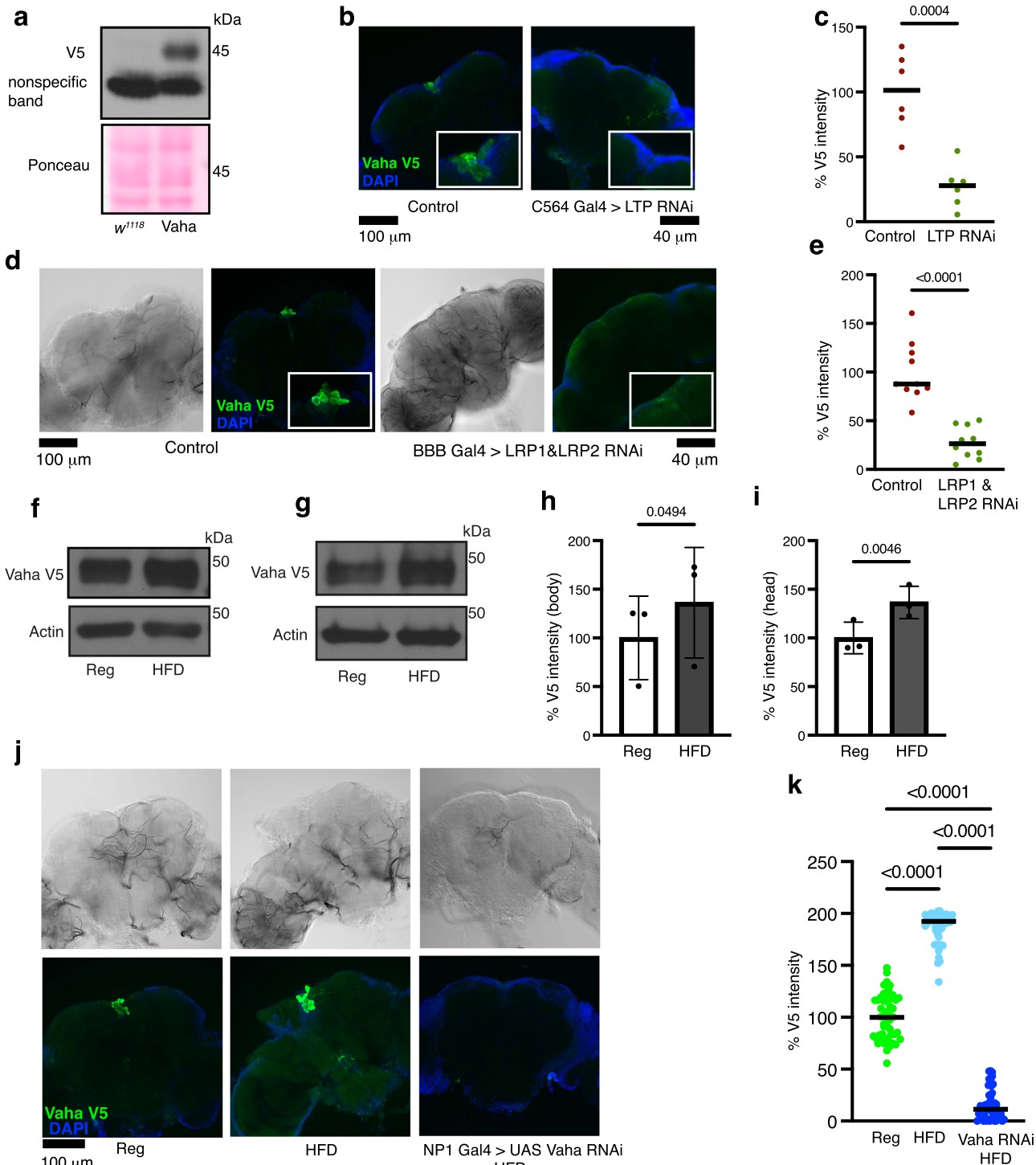

**Fig. 3 | Vaha enrichment in the IPCs requires LTP and Vaha expression is stimulated by dietary fat. a** Immunoblotting with V5 antibody of hemolymph from *w*⁻; VahaV5/VahaV5; + flies. *n* = 3, 40 flies per replicate. **b** Immunostaining for Vaha V5 (anti V5, green) and DAPI (blue) in brains isolated from *w*⁻;VahaV5/C564Gal4; Gal80ᵗˢ /TM6B and *w*⁻; VahaV5/C564Gal4; Gal80ᵗˢ /LTPRNAi flies. Confocal images are projection of z stacks, *n* = 6 brains, scale bar 100 μm. **c** Each dot represents V5 fluorescence intensity in IPC region of one brain of flies described in b. *n* = 6 brains, data is assessed by two-tailed Student's t-test and horizontal line indicates median. **d** Immunostaining for Vaha V5 (anti V5, green) and DAPI (blue) in brains isolated from *w*⁻;VahaV5/+; BBBGal4/+) and *w*⁻; VahaV5/LRP2RNAi; LRP1RNAi/BBB Gal4 flies. Confocal images are projection of z stacks, *n* = 10 brains, scale bar 100 μm. **e** Each dot represents V5 fluorescence intensity in IPC region of one brain of flies in d. *n* = 10 brains, data is assessed by two-tailed Student's t-test and

horizontal line indicates median. **f–g** Immunoblotting for Vaha V5 in body extract (**f**) and head extract (**g**) from flies on regular food (Reg) and regular food containing coconut oil (HFD) for 1 h after starvation overnight. Actin serves as loading control. **h–i** Quantification of blots shown in f and g. *n* = 3, 1 body per replicate (**h**) and *n* = 3, 4 heads per replicate (**i**). Data is assessed by two-tailed paired t-test and presented as mean *n* ± SD. **j** Immunostaining for Vaha V5 (anti V5, green) and DAPI (blue) in brains dissected from flies raised on Reg and HFD for 1 h after starvation overnight and gut specific RNAi on HFD. Confocal images are projection of z stacks, *n* = 6 brains, scale bar 100 μm. **k** Each dot represents V5 fluorescence intensity in one IPC, *n* = 50 (Reg), *n* = 44 (HFD), *n* = 53 (Vaha RNAi HFD). Data is assessed by One-way ANOVA followed by Tukey's multiple comparison test and horizontal line indicates median. Source data are provided for 3a, c, e, f, g, h, i, k as a Source data file.

by the liver to deliver fat to peripheral tissues. The main lipoproteins of *Drosophila* are lipophorin (LPP) and lipid transfer particle (LTP) which are similar to mammalian apoB containing lipoproteins[43]. LPP carries the bulk of the lipids in circulation, the most abundant of which is DAG[43]. Both LPP and LTP are synthesized in the fat body and function together to mobilize lipids from the gut[43]. Tissue specific LPP receptors have been shown to tether LTP at the membrane surface in lipid accepting tissues like the larval imaginal discs, oocytes, and brain[44]. An elegant study in the larval brain has shown that in response to yeast lipids, LTP accumulates on specific neurons called DRNs that contact the IPCs to promote ILP2 release[45]. Since LTP can convey dietary information to the brain, we decided to test if LTP could be involved in the movement and / or enrichment of the lipase in the IPCs. We performed conditional knockdown of LTP in the fat body using RNAi and the Gal4-Gal80$^{ts}$ system. The efficiency of LTP knockdown was confirmed by quantitative PCR (Supplementary Fig. 9b). We examined Vaha immunostaining in the brain and it was significantly reduced in the IPCs when LTP was knocked down compared to the control IPCs (Fig. 3b and c). Two LDL receptor like proteins, LRP1 and LRP2 (Megalin) have been shown to transport LTP across the larval blood brain barrier to control insulin release and signaling in response to yeast lipids[45]. We asked if LRP1 and LRP2 could be involved in the transport of Vaha by knocking down LRP1 and LRP2 in the blood brain barrier glia and examining Vaha immunostaining in the brain. Vaha immunostaining in the IPCs is considerably reduced when LRP1/LRP2 are knocked down in the BBB (Fig. 3d and e). Together, these results suggest that the transport of Vaha across the BBB and its enrichment in the IPCs is dependent on LTP and LRP1/LRP2.

## Vaha is stimulated in response to dietary fat

Since dietary information is mostly perceived by the gut before being conveyed to the brain and other organs, we next examined the nutritional dependence of Vaha expression. Coconut oil has been established as a source to increase the fat content, particularly medium chain triglycerides, while a sucrose diet increases the glucose content of food[46–48]. V5 tagged Vaha transgenic flies were starved for 16 hours (supplemented with water) and then fed with regular food or regular food containing approximately 20% coconut oil (HFD, lipid source) or regular food containing 2 M sucrose (sugar source) for 1 hour. We have used this fasting / refeeding paradigm as our experimental setting throughout this study. We examined transcriptional regulation of Vaha by measuring *Vaha* transcript levels under these dietary conditions. *Vaha* transcript level was increased upon feeding high fat while it did not change significantly upon feeding high sucrose. (Supplementary Fig. 9c, d). Western blot analysis of body extracts from these flies showed Vaha protein expression increased upon feeding high fat (Fig. 3f and h). However, its expression did not change significantly upon feeding high sucrose containing food (Supplementary Fig. 9e, f). Lipase level was also increased in the head extracts of flies fed food containing coconut oil (Fig. 3g and i). These flies also had higher V5 lipase staining in the IPCs compared to flies on regular food (Fig. 3j and k). Furthermore, knockdown of Vaha in the gut by RNAi significantly reduced the accumulation of Vaha V5 in the IPCs upon feeding high fat (Fig. 3j and k). These results show Vaha expression in the gut is increased by dietary fat ingestion and fat stimulation increases enrichment of the lipase in the IPCs.

## Vaha mediates gut-brain communication in the regulation of ILP2 secretion

To characterize Vaha function, we generated a deletion in the lipase gene using the CRISPR-Cas9 system (design shown in Supplementary Fig. 10a)[49,50]. The gRNAs were designed to delete most of the coding region resulting in essentially a null allele. qPCR analysis confirmed that the deletion indeed resulted in a null mutant (Supplementary

Fig. 10b). After isolation, the mutant was backcrossed several rounds to *w$^{1118}$* to isogenize the chromosomes and outcross off-target mutations. Mutant flies (*vaha*) are viable, fertile and do not display obvious morphological defects. However, *vaha* mutants exhibited developmental delay. We monitored control and mutant animals from embryo through adult stages (Supplementary Fig. 10c). 48 h after egg laying, the mutant larvae were smaller compared to control, however, they caught up in size and attained normal pupal and adult size. *vaha* mutant development is slower throughout life with a 48 h delay in the eclosion of adults (Supplementary Fig. 10d). Of the 400 *w$^{1118}$* first instar larvae, 388 (97%) developed to adults and of the 400 *vaha* mutant larvae, 383 developed to adults (95.75%). The smaller size of mutant larvae and delay in eclosion could be rescued by gut specific overexpression of Vaha using the NP1 Gal4 driver (Supplementary Fig. 10c, d). The size, and weight of adult *vaha* mutant flies were not significantly different from control *w$^{1118}$* and their lifespan was comparable to control flies (Supplementary Fig. 11a–c). Food intake as measured by the CAFE assay showed that it was not significantly different in mutants compared to control flies (Supplementary Fig. 11d).

The enrichment of Vaha in the IPCs suggested that it could have an important function in these cells. We first tested if Vaha has a role in *ILP2* gene expression by analyzing its transcript level during the fasting (16 h) / refeeding (1 h) paradigm. qPCR analysis indicated that *ILP2* levels were not significantly different in the *vaha* mutants compared to control under these conditions (Supplementary Fig. 11e). Since insulin activity is also regulated during release, we examined if *vaha* mutants displayed defects in insulin secretion. We used an experimental approach previously employed both in larvae and adults, assaying for steady state levels of ILP2 in the IPCs using the fasting/refeeding model[28,38]. Under nutrient deprivation, ILP2HF secretion was reduced and thus it accumulated in the IPCs of fasted control animals (Fig. 4a top panel, Fasted). Feeding reduced ILP2HF accumulation in the IPCs as it was released into circulation in response to food (Fig. 4a top panel, Fed Reg). Feeding high fat diet resulted in further reduction of ILP2HF in the IPCs compared to regular food (Fig. 4a top panel, Fed HFD). In *vaha* mutants, ILP2HF accumulation in the IPCs was comparable to control flies in the fasted state (Fig. 4a, bottom panel). In contrast, they retained more ILP2HF in the cell bodies in response to feeding regular food (Fig. 4a, bottom panel). The fluorescence intensity of ILP2HF in IPCs in the control and *vaha* mutants under the different conditions is quantified in Fig. 4b. The failure to secrete insulin in response to dietary stimulation was further worsened on high fat food as there was a wider differential (approximately 30%) in ILP2HF intensity in the IPCs of control and *vaha* mutant on high fat diet (80%) versus control and *vaha* mutant on regular food (50%, Fig. 4b). The mutant phenotype was noted both in male and female flies. Gut specific expression of a wild type copy of Vaha (UAS-Vaha GFP) rescued the mutant phenotype both in regular and high fat food (Fig. 4a, bottom panel and Fig. 4b). Ubiquitous expression of UAS-Vaha GFP using tubulin>Gal4 driver or expression of Vaha using Vaha Gal4 driver also rescued the mutant phenotype (Supplementary Fig. 12a–d). We also confirmed that the mutant phenotype originated from the Vaha locus by putting the *vaha* mutant over a deficiency and monitoring insulin secretion (Supplementary Fig. 12e, f). Another interesting observation in *vaha* mutants is that immunostaining revealed ILP2HF in a small number of specific neurons in addition to the IPCs, albeit the intensity being far less compared to the IPCs (Supplementary Fig. 12g). The identity of these neurons, whether they communicate with the IPC neurons, and if there is a causal role for Vaha in this process are interesting questions for a future study.

To test if increased staining in the mutant IPCs in the fed state was due to decreased ILP2HF released into circulation, we measured ILP2HF levels in the hemolymph of control and mutant flies under the

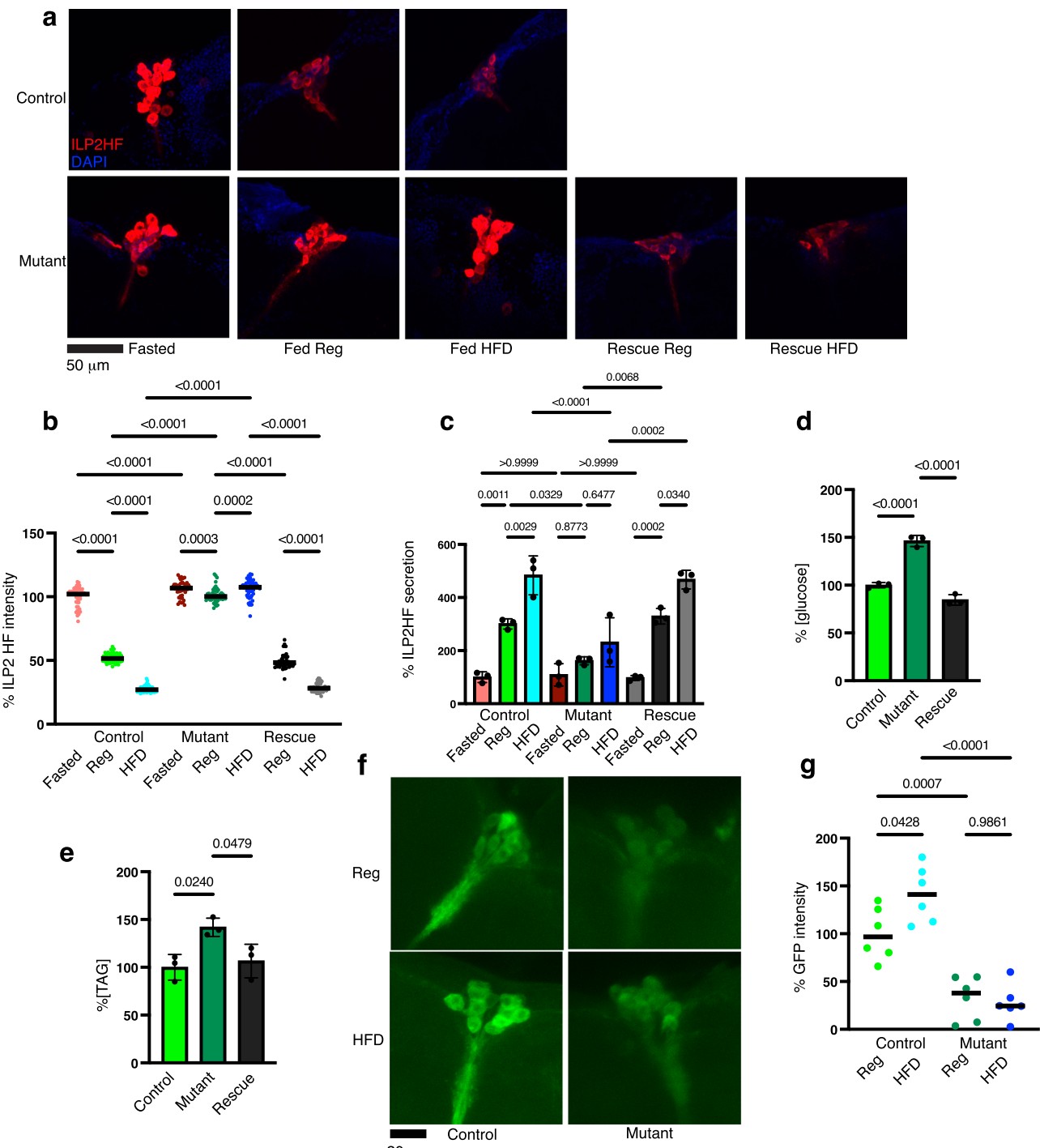

**Fig. 4 | *vaha* mutants are defective in ILP2HF secretion. a** Immunostaining of brains from 5-7 day old control, *vaha* mutant and rescue flies for ILP2HF (anti HA, red) and DAPI (blue). The genotypes are control *w⁻; +/+; gd2HF/+; vaha* mutant *w⁻; vaha/vaha*; gd2HF/+ and rescue *w⁻*; NP1 Gal4.*vaha/vaha*; gd2HF/UAS VahaGFP. Flies fasted overnight are fed either regular food (Reg) or regular food containing coconut oil (HFD) for 1 h. Confocal images are projection of z stacks *n* = 6 brains, scale bar 50 μm. **b** Quantification of ILP2HF fluorescence intensity in the IPCs of control, *vaha* mutants and rescue flies in the fasted and fed states (described in a). Each data point represents fluorescent intensity in one IPC, *n* = 47 (control fasted), *n* = 51(control fed Reg), *n* = 48 (control fed HFD), *n* = 47 (mutant fasted), *n* = 50(mutant fed Reg), *n* = 52(mutant fed HFD), *n* = 49 (rescue Reg) and *n* = 49 (rescue HFD). One-way ANOVA followed by Tukey's multiple comparison test is used and horizontal line indicates median. **c** ILP2HF levels in circulation assayed in control, *vaha* mutants and rescue in the fasted and fed states

(genotypes described in a). Each data point represents value from *n* = 3, 15 male flies per replicate. One-way ANOVA followed by Tukey's multiple comparison test is used and data presented as mean*n* ± SD. **d** Glucose content **e** TAG content measured in homogenates from flies described in a. Each data point represents value from 10 flies. *n* = 3, data is assessed by two-tailed Student's t-test and presented as mean ± SD. Rescue is shown for flies on regular food. **f** IPC activity measured in control and *vaha* mutants using the CaLexA system. The genotype of control flies is *w⁻*; LexAopCD8GFP2ACD8GFP/dILP2Gal4; CaLexALexAop CD2GFP/gd2HF and mutant flies is *w⁻*; LexAopCD8GFP2ACD8GFP.*vaha /dILP2Gal4.vaha*; CaLexALexAopCD2GFP/gd2HF. Confocal images are projection of z stacks, *n* = 6 brains, scale bar 20 μm. **g** Each dot represents intensity of GFP fluorescence in one brain, flies are described in **f**. *n* = 6 brains. Data is assessed by One-way ANOVA followed by Tukey's multiple comparison test and horizontal line indicates median. Source data are provided for 4b, c, d, e, g as a Source data file.

different conditions[37]. ELISA results showed that while fasting ILP2HF levels were not significantly different between control and mutant, circulating ILP2HF levels were considerably decreased in *vaha* mutants both in regular and high fat food (Fig. 4c). The decrease in ILP2HF levels in the mutant in regular and high fat food could be rescued by gut specific expression of Vaha (Fig.4c). It is well established that genetic ablation of IPCs in flies is accompanied by increase in glucose and lipid levels[23,24,51]. To test if decreased ILP2HF secretion in the *vaha* mutant affects glucose and TAG levels, we measured whole body glucose and TAG levels in homogenates from control, mutant and rescue flies fed regular food for one hour after fasting overnight. As can be seen in Fig. 4d, e, *vaha* mutants show elevated glucose and TAG compared to control and these were rescued by gut specific expression of Vaha. To test if IPC activity is compromised in the *vaha* mutant, we used the CaLexA (calcium-dependent nuclear import of LexA) system since IPC activation is accompanied by an increase in intracellular calcium levels. Here, when calcium levels rise in the IPCs, CaLexA enters the nucleus and binds the LexA operator to activate downstream CD8 GFP and CD2 GFP reporter expression which can be visualized by staining with an antiGFP antibody[52]. IPC activity thus measured in the *vaha* mutant is reduced compared to the control on regular food as well as HFD (Fig. 4f and quantification in Fig. 4g). In this experiment, ILP2HF staining was also included to mark the IPCs and Supplementary Fig. 13a and b show all the panels including LexA GFP, ILP2HF, DAPI and the merged images on regular food and HFD respectively.

The defect in insulin secretion was also observed in gut (enterocyte) specific RNAi knockdown of the lipase. As seen in Supplementary Fig. 14a, b more ILP2HF was retained in the IPCs of Vaha knockdown flies compared to controls. ELISA results confirmed that the levels of circulatory ILP2HF were decreased in the Vaha knockdown flies (Supplementary Fig. 14c). On the other hand, gut specific overexpression of Vaha, resulted in reduced staining of ILP2HF in IPCs and increased levels of circulatory ILP2HF (Supplementary Fig. 14d–f). Collectively, these experiments show that Vaha modulates insulin secretion since lack of Vaha results in defective insulin-like peptide release, decreased levels of circulatory insulin-like peptide and decreased IPC activity when transitioning from the fasted to fed state. Overexpression of Vaha, on the other hand, increased ILP2HF secretion.

We next examined if targeted expression of Vaha in IPCs of *vaha* mutant could correct insulin secretion, glucose, and TAG levels. Expression of UAS-Vaha GFP using the dILP2Gal4 driver in the *vaha* mutant rescued the insulin secretion defect as seen in Fig. 5a, b and the metabolic defects as seen in Fig. 5c and d. Since Vaha is a secreted protein, we addressed the local requirement of Vaha by expressing the signal peptide deleted transgenic (UAS VahaΔ30 GFP) in a similar manner. This also corrected the insulin secretion and metabolic defects suggesting that the presence of Vaha within IPCs is sufficient for this function. We asked if lipase activity is important for the function of Vaha in the IPCs. Classical lipase function depends on a catalytic triad (Ser-His-Asp) mechanism with the Ser located in the middle of the GXSXG motif found in lipases that adopt the α/β hydrolase fold[53]. The Ser-His-Asp catalytic triad is also present in Vaha and we generated GFP tagged UAS transgenic flies wherein each of these three active site residues was replaced by Ala. When this construct was expressed in the IPCs, ILP2HF levels were considerably higher compared to control suggesting that lipase activity is essential for ILP2 release from IPCs (Fig. 5a, b). The increase in glucose and TAG levels in the *vaha* mutant were not corrected by expression of the active site mutant (Fig. 5c, d). We also expressed the active site mutant in the gut using the NP1Gal4 driver, dissected brains, and stained for the GFP tagged lipase (Supplementary Fig. 15a). The mutant lipase colocalized with ILP2HF in the IPCs, suggesting lipase activity was not essential for its movement from the gut to the brain.

## Metabolic changes and developmental delay in *vaha* mutants result from compromised insulin secretion

We explored the consequences of modulating IPC activity on the different phenotypes of *vaha* mutant. We asked if activating ILP secretion could rescue *vaha* mutant phenotypes. To achieve this, we depolarized the IPCs by expressing the bacterial sodium channel (NaChBac) under the control of dILP2 Gal4[54]. Under these conditions, ILP2HF accumulation in the IPCs of *vaha* mutant was considerably reduced (Fig. 6a and b). The activation also rescued the metabolic changes (increased glucose and increased TAG) observed in the *vaha* mutant during fasting/refeeding (Fig. 6c and d). We also examined whether activation of the IPCs would rescue the developmental delay in *vaha* mutants and indeed it did so (Fig. 6g). On the other hand, we reduced ILP activity by using *ilp2,3,5 / ilp2,3* mutants which were viable compared to *ilp2,3,5 / ilp2,3,5* mutants that were lethal in our hands[55]. We then made mutants of *vaha* with *ilp2,3,5 / ilp2,3* (combined mutant) and measured glucose and TAG levels in this background. As seen in Fig. 6e, f, the metabolite levels in the combined mutant were not different from *ilp2,3,5 / ilp2,3* mutants alone. We also monitored the development from embryos to adults in these mutant backgrounds. As shown in Fig. 6g, the eclosion time for 100% of the combined mutant flies was like that of the *ilp2,3,5 / ilp2,3* mutant. To further test the idea that Vaha affects glucose metabolism through insulin, we performed a glucose tolerance test (GTT). The flies were fasted and transferred to a glucose diet for one hour and then re-fasted for one hour (Fig. 6h). The impairment in glucose clearance upon re-fasting was not significantly different in the combined mutants compared to *ilp2,3,5 / ilp2,3* mutants. This result shows loss of Vaha does not significantly affect GTT in insulin mutant background. Additionally, overexpression of Vaha in the gut reduced glucose and TAG levels in wild type flies during fasting/feeding, however their levels were not changed when Vaha was overexpressed in *ilp2,3,5 /ilp2,3* mutant background (Supplementary Fig. 15b–e). Collectively these results suggest that the developmental delay and metabolic changes observed during fasting/refeeding in Vaha deficient animals are a consequence of altered insulin release.

## *vaha* mutants display features of diabetes

To examine global metabolic changes triggered by the loss of Vaha in adult flies, we used UPLC-MS/MS to profile the metabolome in *vaha* mutants and compared them to control $w^{1118}$ flies (30-35day old flies). Metabolic profiling revealed 156 out of 428 biochemicals and lipidomic profiling revealed 294 out of 921 lipid species with statistically significant changes ($p \le 0.05$). The profiled lipids, the pathways they belong to, the fold change (mutant/control) and statistical significance is shown in Supplementary Data 1 (additional Supplementary File). Similar information for various biochemicals is shown in Supplementary Data 2 (additional Supplementary File). The raw data used to generate the fold change (mutant/control) for lipids is provided as Supplementary Data 3 (additional Supplementary File). The raw data used to generate the fold change (mutant/control) for metabolites is provided as Supplementary Data 4 (additional Supplementary File). Lipidomic and metabolomic analyses revealed that *vaha* mutants display features of diabetes. Flies raised on high sugar, like human diabetics, develop hyperglycemia and hyperlipidemia, both of which are hallmarks of diabetes[48,56]. 30-35 day *vaha* mutant flies showed elevated whole body glucose and TAG levels (Fig. 7a–b). We performed differential abundance analyses of the metabolites listed in Supplementary Data 1 and Supplementary Data 2. A heatmap representation of the differentially expressed lipid species is shown in Fig. 7c while that for the other metabolites is shown in Fig. 7d. Among the lipid classes, differential abundance analysis showed that number of TAG species accumulate in the mutant compared to control flies (Fig. 7c). Other than TAG, some DAG species were also elevated in the mutant. Glucose metabolic pathways that regulate its cellular fate are also important considerations in the pathophysiology of diabetes[57]. The three major

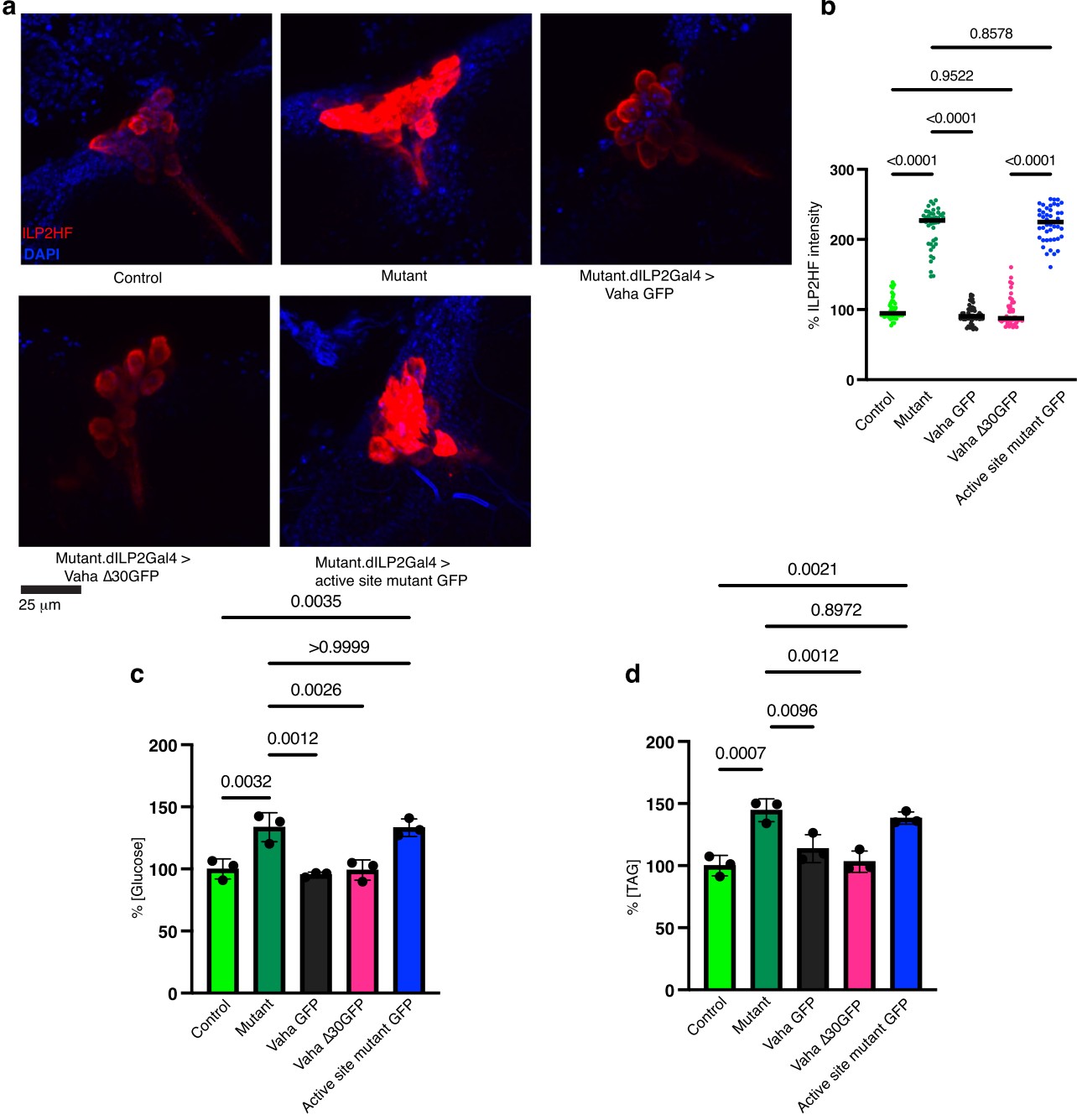

**Fig. 5 | Expression of Vaha GFP, Vaha Δ30GFP in the IPCs rescue whereas lipase active site mutant does not rescue *vaha* mutant phenotypes. a** Immunostaining of brains dissected from 5-7 day old control, *vaha* mutant and Vaha GFP, Vaha Δ30GFP, and active site mutant flies for ILP2HF (anti HA, red) and DAPI (blue). Flies were fasted overnight and put on regular food. In the active site mutant, the Ser-His-Asp catalytic triad required for lipase activity are replaced by Ala. The genotype of the control flies is ($w^-$; +/+; gd2HF/+; *vaha* mutants are $w^-$; *vaha/vaha*; gd2HF/+ and different transgenic flies are $w^-$; dILP2Gal4.*vaha/vaha*; gd2HF/UAS VahaGFP and $w^-$; dILP2Gal4.*vaha/vaha*; gd2HF/ Vaha Δ30GFP and $w^-$; dILP2Gal4.*vaha/vaha*; gd2HF/ UAS active site mutant GFP. Confocal images are projection of z stacks, *n* = 6 brains, scale bar 25 µm. **b** Quantification of ILP2HF fluorescence intensity in the IPCs of flies described in (**a**). Each data point represents fluorescent intensity in one IPC, *n* = 45 (control), *n* = 45 (mutant), *n* = 53 (mutant.dILP2Gal4>Vaha GFP), *n* = 44 (mutant.-dILP2Gal4>Vaha Δ30GFP), *n* = 45 (mutant.dILP2Gal4>active site mutant GFP). Data is assessed by One-way ANOVA followed by Tukey's multiple comparison test and horizontal line indicates median. **c** Glucose content measured in flies described in a. Each data point represents value from 10 flies. *n* = 3, data is assessed by One-way ANOVA followed by Tukey's multiple comparison test and presented as mean ± SD. **d** TAG content measured in flies described in a. Each data point represents value from 10 flies. *n* = 3, One-way ANOVA followed by Tukey's multiple comparison test is used and data is presented as mean ± SD. Source data are provided for 5b, c, d as a source data file.

pathways include (i) oxidation to pyruvate by glycolysis and further oxidation of pyruvate through the TCA cycle, (ii) conversion to other sugars and intermediates for biosynthetic and metabolic purposes (iii) and storage as glycogen for later use. *vaha* mutants showed elevated glycolytic intermediates (glucose 6-phosphate, phosphoenolpyruvate,

pyruvate, Fig. 7d). 6-phosphogluconate and sedoheptulose 7-phosphate, intermediates of the pentose phosphate pathway, an alternate route for glucose metabolism, were also elevated in *vaha* mutants (Fig. 7d). Metabolomic data indicated that long chain acylcarnitines (cerotoylcarnitine, dihomo-linoleoylcarnitine, behenoylcarnitine)

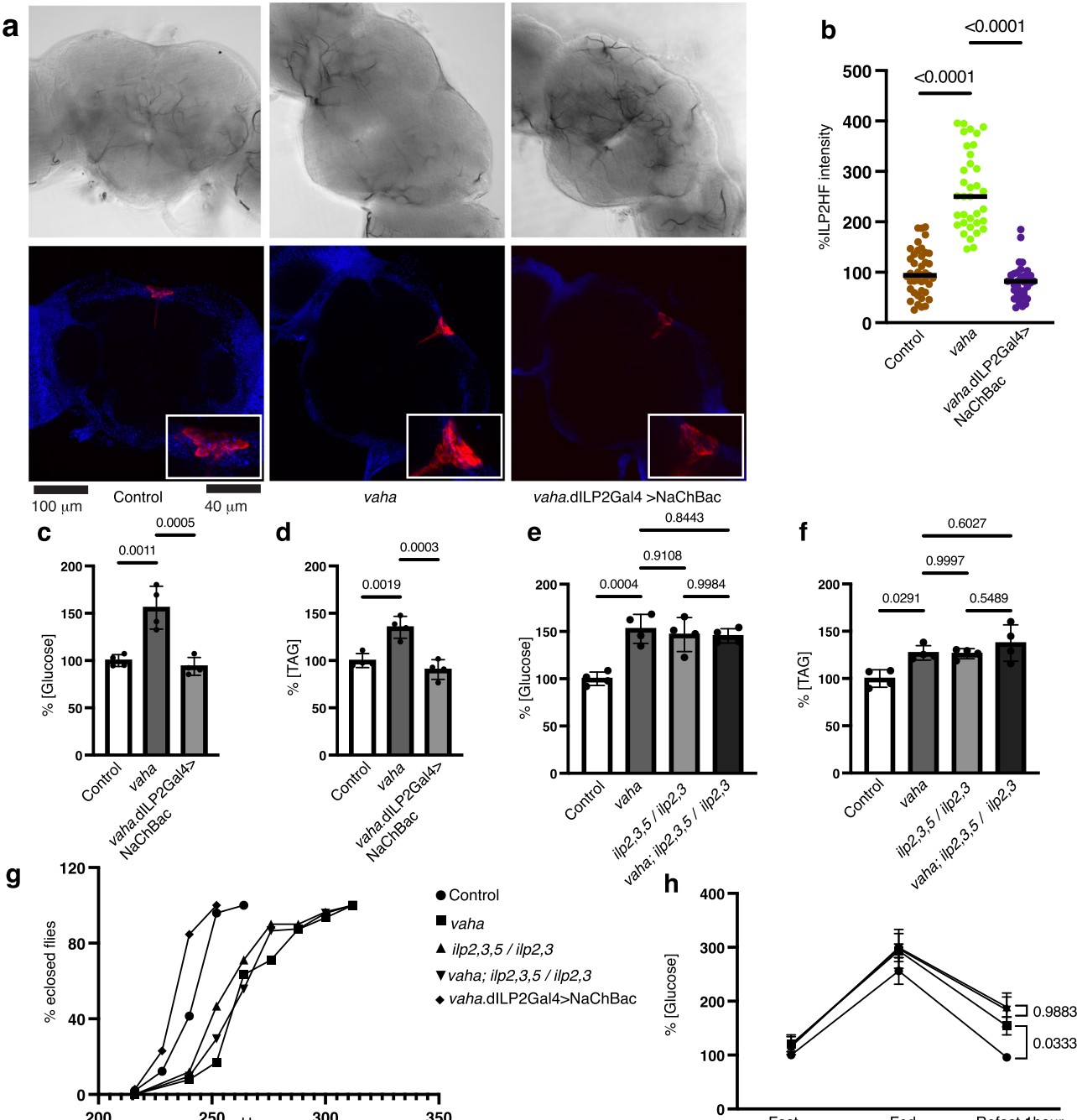

**Fig. 6 | Vaha influences glucose metabolism through ILP. a** Immunostaining of brains dissected from 5-7 day control, *vaha* and *vaha* expressing bacterial sodium channel via dILP2Gal4 driver for ILP2HF (red) and DAPI (blue). The top panel shows brightfield images. Confocal images are projection of z stacks *n* = 6 brains, scale bar 100 µm. **b** Quantification of ILP2HF fluorescence intensity in the IPCs of flies described in a. Each data point represents fluorescent intensity in one IPC, *n* = 41(control), *n* = 35 (*vaha* mutant), *n* = 35 (*vaha*.dILP2Gal4>NaChBac). One-way ANOVA followed by Tukey's multiple comparison test is used and horizontal line indicates median. **c** Glucose content and **d** TAG content measured in flies described in a. Each data point represents value from 10 flies, *n* = 4, One-way ANOVA followed by Tukey's multiple comparison test is used and data is presented as mean ± SD. **e** Glucose content and **f** TAG content measured in vaha mutant, insulin mutant and

combined *vaha* and insulin mutant flies. The genotypes of flies are control *w^{1118}*; *vaha* mutants *w*; *vaha/vaha*; insulin mutants *w*; +/+; *ilp2,3,5/ilp2,3* and combined mutants is *w*; *vaha/vaha*; *ilp2,3,5/ilp2,3*. Each dot represents value from 10 flies, *n* = 4, One-way ANOVA followed by Tukey's multiple comparison test is used and data is presented as mean ± SD. **g** Eclosion of adult flies described in a and e-f is followed between 216 to 312 hours after egg laying. **h** Glucose tolerance test performed on adult flies described in e-f. Flies fasted overnight are fed on glucose containing media for 1 h and then re-fasted for 1 h. Circulatory glucose is measured, and each data point represents value from 40 flies. *n* = 3, One-way ANOVA followed by Tukey's multiple comparison test is used and data is presented as mean ± SD. Source data are provided for 6b, c, d, e, f, g, h as a source data file.

were elevated in the mutants compared to the control flies (Fig. 7d). Acylcarnitines are intermediate oxidative metabolites consisting of esterified fatty acid of carnitine, which is involved in the transport of fatty acids to mitochondria for β oxidation[58]. Under physiological

conditions, organisms switch between glucose and fatty acid metabolism based on the availability of substrates to maintain energy homeostasis. To meet energy demands, the mutant flies appear to rely on glycogenolysis as indicated by higher levels of the maltose sugars,

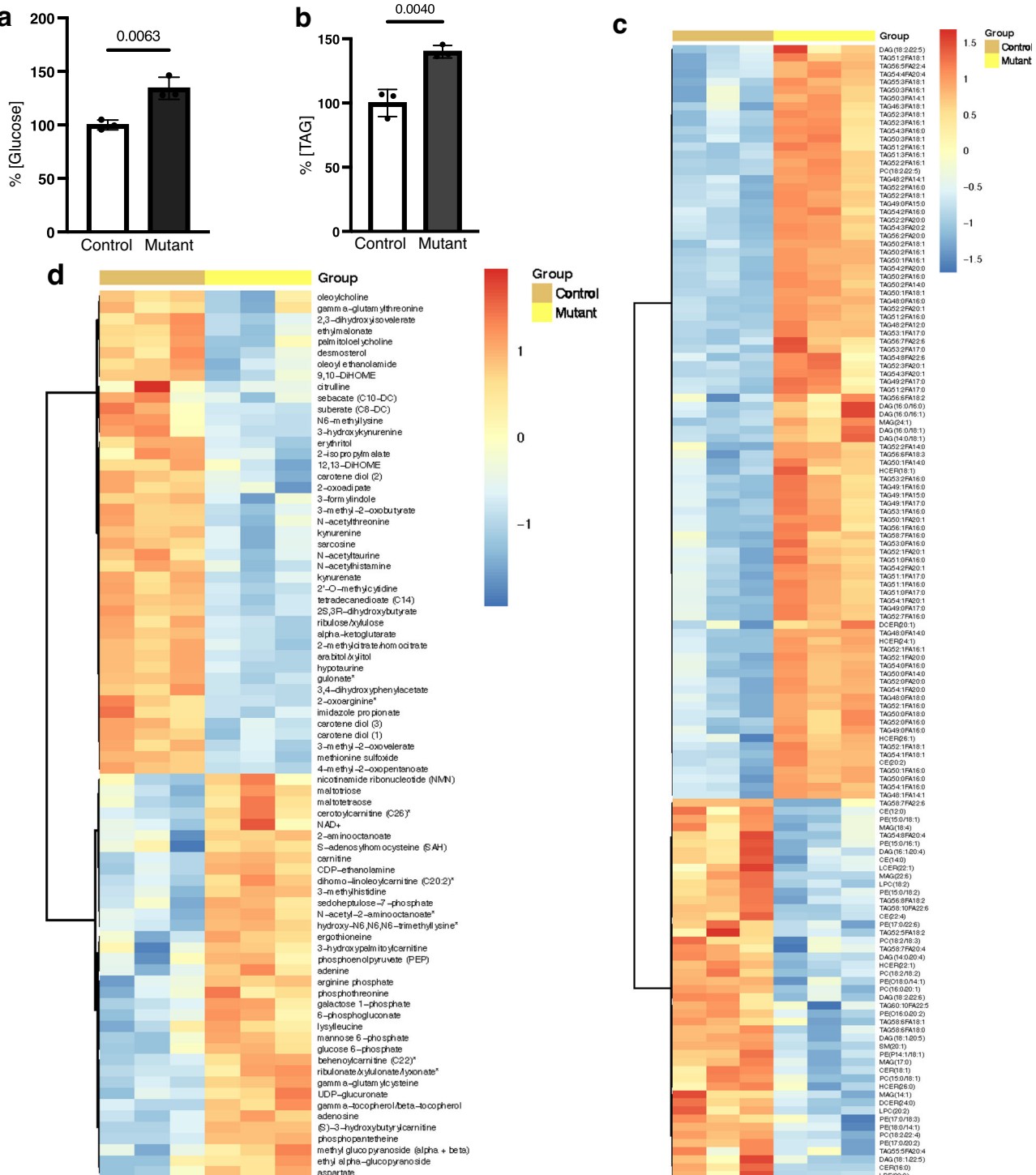

**Fig. 7 | *vaha* mutant flies display features of diabetes. a** Glucose levels were estimated in whole fly homogenates of 30–35 day old control and *vaha* mutant flies by UPLC-MS/MS. Data is obtained from three biological replicates with 100 flies per replicate. Data is assessed by two-tailed Student's t-test and presented as mean ± SD. **b** Triglyceride (TAG) levels were estimated in whole fly homogenates of 30–35 day old control and *vaha* mutant flies by UPLC-MS/MS. Data is obtained from three biological replicates with 100 flies per replicate. Data is assessed by two-tailed Student's t-test and presented as mean ± SD. **c** Heat map showing lipids with significantly altered abundance ((log₂ (fold change) ≥ 0.59 and adjusted *p* value ≤ 0.05) in whole fly samples of the control compared to those of the *vaha* mutants. Columns represent samples and rows represent lipid species. The color key indicates

Z-scores of the relative concentrations of lipid species with blue being the lowest and red the highest. Data is obtained from three biological replicates with 100 flies (30–35 day old) per replicate. **d** Heat map showing metabolites with significantly altered abundance ((log₂ (fold change) ≥ 0.59 and adjusted *p* value ≤ 0.05) in whole fly samples of the control compared to those of the *vaha* mutants. Columns represent samples and rows represent metabolites. The color key indicates Z-scores of the relative concentrations of metabolites with blue being the lowest and red the highest. Data is obtained from three biological replicates with 100 flies (30–35 day old) per replicate. Source data are provided for 7a, b as a source data file.

maltotriose and maltotetraose. The increased glycogenolysis could lead to more glucose production further compounding hyperglycemia. Dysregulated fatty acid oxidation is reflected by elevated plasma acylcarnitine levels in humans and studies have linked elevated plasma acylcarnitines to type 2 diabetes mellitus, insulin resistance, and obesity[59]. To explore other metabolic changes that could contribute to diabetes, we compared metabolites that show differential changes in *vaha* mutants with metabolites (biomarkers) that have been robustly associated with diabetes or risk of diabetes in human patients[59]. Supplementary Table 1 shows changes in glutathione metabolism, pentose phosphate pathway and amino acid metabolism including changes in arginine, tryptophan, and branched chain amino acid metabolism. Altogether, these results suggest *vaha* mutants exhibit changes in metabolites strongly associated with progression of diabetes in humans.

### Vaha functions as a DAG lipase

To understand the role of Vaha in lipolytic breakdown and define the lipase activity of Vaha, we sought to characterize the purified enzyme. Toward this objective, we generated stable, inducible S2 cells expressing C-terminal V5-His-tagged Vaha. Upon induction for 24 h or 48 h, robust accumulation of Vaha was observed in the secreted media (Supplementary Fig. 16a). The lipase was purified to homogeneity from the 48 h supernatant using nickel affinity chromatography and gel filtration utilizing fast protein liquid chromatography (Fig. 8a). The purity of the protein was confirmed by silver staining and verified by Western blot analysis (Fig. 8a). To explore the substrate preference of the enzyme, we used a well-established assay that involves incubating the enzyme with radiolabeled [$^3$H] triolein (TAG) or [$^3$H] diolein (DAG) or [$^3$H] monolein (MAG) as substrate[60]. The lipolytic activity of the enzyme leads to the release of radiolabeled fatty acids which are resolved by thin layer chromatography, scraped, and counted. Using this assay, the substrate concentration curves for TAG, DAG and MAG were generated and the kinetic parameters Km, kcat and kcat/Km were calculated (Fig. 8b–d respectively). The purified lipase showed highest affinity for DAG with a Km of $17.95 \times 10^{-6}$M, followed by TAG with a Km of $19.71 \times 10^{-6}$M, and then MAG with a Km of $25 \times 10^{-6}$M. The ratio of kcat/Km is a useful indicator of the relative efficiency of the enzyme when it has more than one substrate. Based on the observation that the enzyme has highest kcat/Km values for DAG, Vaha is likely a DAG lipase (Fig. 8c). Furthermore, the increase in total DAG and decrease in MAG levels in *vaha* mutants observed in our lipidomic profiling suggests that the lipolytic cascade is interrupted at the stage of DAG hydrolysis (Fig. 8e, f). The individual species of DAG and MAG used to calculate the total DAG and MAG concentrations in the control and mutant groups are shown in Supplementary Table 2. These results further strengthen the idea that Vaha functions as a DAG lipase under physiological conditions. Unlike TAG, DAG and MAG, total free fatty acid (FFA) levels did not show significant differences in the *vaha* mutants (Supplementary Fig. 16b). However, certain monounsaturated FFA (16:1, 20:1, 22:1) and polyunsaturated FFA (18:2, 22:2) levels were decreased in the mutant compared to control flies (Supplementary Data 1).

### Lipid supplementation stimulates ILP2HF release from IPCs

Our results show enhanced release of ILP2 from the IPCs into circulation after fat ingestion and Vaha mediates this process. It has been shown that exogenous lipids can be successfully delivered to flies by supplementing food with the desired lipids[61,62]. To test if TAG, DAG, or MAG could cause release of ILP2HF from the IPCs, we evaluated dietary supplementation of these lipids for 2-3 h, followed by dissection of brains and staining for ILP2HF. As seen in Fig. 8g, all three lipids could induce release of ILP2HF from the IPCs with MAG and DAG being more effective than TAG. The fluorescent images are shown in Supplementary Fig. 16c. We next measured ILP2HF release in *vaha* mutants

supplemented with these lipids. While TAG and DAG did not cause release of ILP2HF, MAG could partially rescue ILP2HF secretion in the *vaha* mutant (Fig. 8h, Supplementary Fig. 16d). These data suggest that MAG and/or its downstream product(s) likely mediate the effects of Vaha.

In summary, we have shown that Vaha produced in the midgut moves to the brain when stimulated by dietary fat. It concentrates in the IPCs in the brain in a process requiring the lipoprotein LTP. Vaha functions as a DAG lipase releasing MAG and free fatty acid to induce ILP2 secretion from the IPCs to maintain adult glucose homeostasis (Fig. 8i model).

### Discussion

The metabolic amplification of glucose-stimulated insulin secretion by dietary fat and protein allows for optimal secretion of insulin after a meal. In mammals, fatty acids are one of the lipid signals that amplify GSIS[1–4]. They can be obtained exogenously from circulation or by hydrolysis of circulating triglycerides in chylomicrons via the action of lipoprotein lipase, which has been proposed to augment GSIS[63]. Fatty acids can also be generated endogenously through lipolysis of intracellular triglycerides or by conversion to fatty acyl CoA which is metabolized to generate lipid signals that potentiate insulin secretion[7,8]. The impairment of GSIS by orlistat, a pan lipase inhibitor, is important evidence for the involvement of lipolysis and lipase activity in GSIS[64]. While mouse knockout models of adipose triglyceride lipase and hormone sensitive lipase that catalyze the lipolysis of TAG and DAG respectively suggest defects in GSIS, the results are not unequivocal[65–67]. ABHD6 mouse knockout results in elevated MAG and enhanced GSIS, suggesting MAG is an important lipid mediator[68]. On the other hand, pharmacological inhibition of MAG lipase reduces GSIS despite an increase in MAG[69]. The complexities of mammalian systems add to the difficulty in defining mechanisms that augment GSIS after a meal. *Drosophila* is a powerful yet simple organism to study inter-organ communication. Larvae serve as a model for the identification of organ to organ communicating factors that contribute to growth and development, while the adult flies are a model for metabolic homeostasis, reproduction and aging[12]. The *Drosophila* fat body secretes hormonal factors in response to dietary nutrients which modulate insulin secretion to regulate larval growth and body size. The positive regulators include the cytokine Unpaired 2, a circulating peptide Stunted, growth-blocking peptides GBP1 and GBP2, a peptide hormone CCHamide-2 and the Activin like ligand Dawdle[29–32,34,35]. The negative regulators include the *Drosophila* TNF-α homolog Eiger, and insulin-like growth factor Impl2[33,70]. These signals act either through their receptors in the IPCs or through a relay involving neurons that are in contact with the IPC neurons. In contrast to growth and development, less is known about factors involved in homeostatic glycemic regulation. Unlike larval IPCs which respond to amino acids for ILP release, adult IPCs respond to glucose for ILP release and are therefore amenable to study GSIS, its amplification, and maintenance of glucose concentrations[36,71]. Our results (Fig. 4a–c) show that there is indeed robust fat mediated amplification of insulin secretion in adult fly IPCs. *vaha* mutants are defective in fat amplified insulin secretion and exhibit hyperglycemia like human diabetics. They provide a context to probe glucose homeostasis in adult flies uncoupled from growth defects.

Our attempt to understand how Vaha moves from the gut to the IPCs suggests lipase activity is not required for Vaha movement since it is detected in the IPCs of flies deficient in catalytic activity. In adult flies, when the lipoprotein LTP is reduced in the fat body, or LDL receptor like proteins LRP1/LRP2 are knocked down in the blood brain barrier glia, Vaha fails to concentrate in the IPCs showing its transport across the blood brain barrier and enrichment in the IPCs involves LTP and LRP1/ LRP2. Future experiments probing how Vaha and LTP interact would shed light on Vaha's mode of transport and action at the

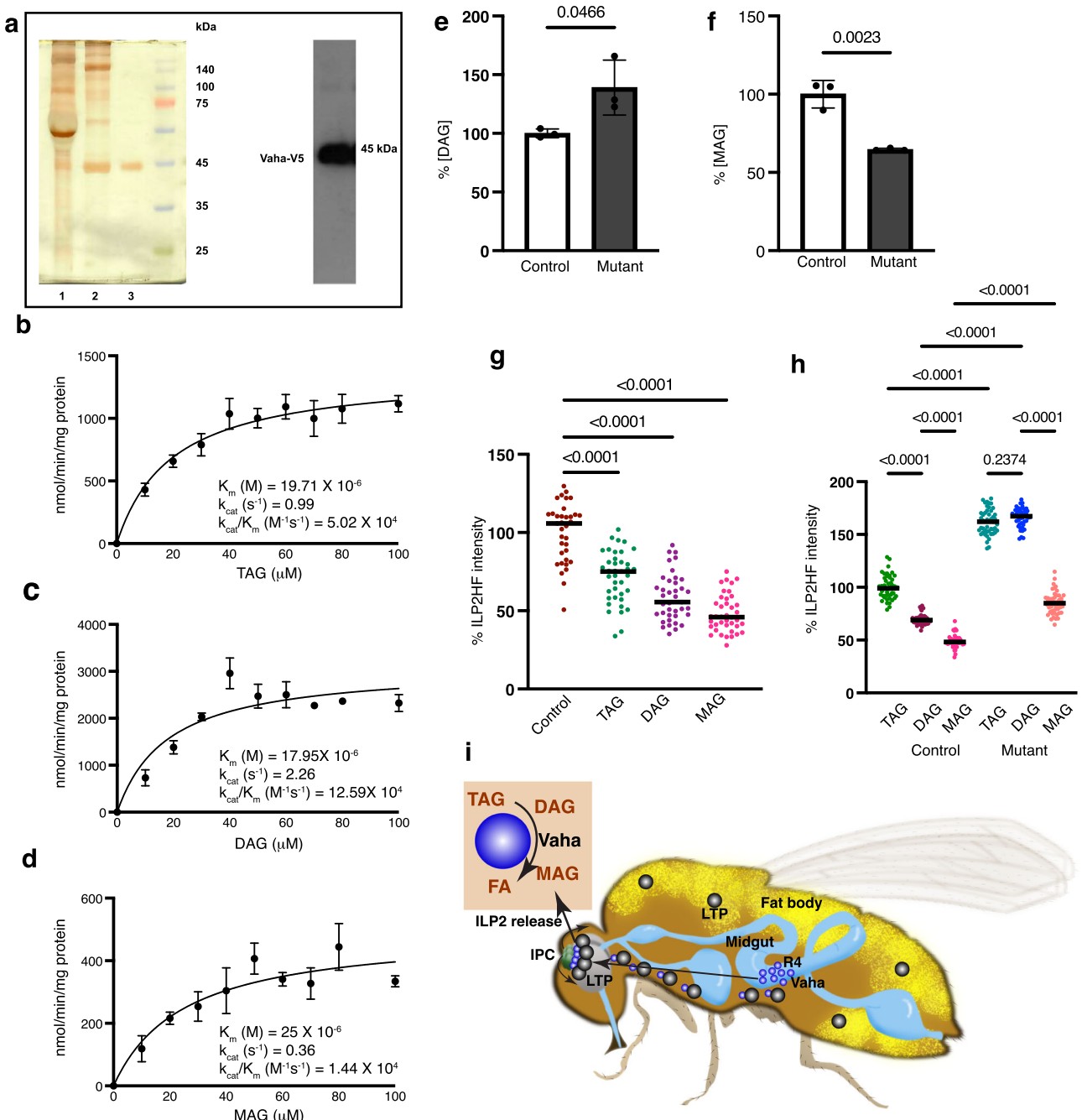

**Fig. 8 | Vaha functions as a DAG lipase and dietary supplementation of TAG, DAG, or MAG increases insulin secretion from IPCs. a** Silver stained SDS-PAGE gel showing purification steps. Lane 1: 50% ammonium sulfate precipitate of culture supernatant from Vaha V5 stable cells, lane 2: nickel affinity chromatography, lane 3: gel filtration by FPLC. Western blot of the purified protein with V5 antibody. $n = 2$. **b–d** Purified Vaha protein is incubated with increasing concentrations of (**b**)TAG (**c**) DAG or (**d**) MAG (cold + radiolabeled) as a lipid substrate and the lipase activity is determined by measuring the released fatty acid. The substrate concentration curve is used to calculate Km, kcat and kcat/Km. For each concentration, 3 independent replicates are performed. Data are presented as mean ± SD. **e, f** DAG (**e**) or MAG levels estimated in fly homogenates of 30(**f**) 35-day-old control and *vaha* mutant by LC-MS/MS (**f**). $n = 3$, 100 flies per replicate. Data assessed by two-tailed Student's t-test and presented as mean ± SD. **g** Quantification of ILP2HF fluorescence intensity in the IPCs of control (*w⁻*; +/+; gd2HF/gd2HF) flies, and fed either TAG, DAG, or MAG. Each data point represents fluorescent intensity in one IPC, $n = 35$ (control), $n = 39$ (TAG), $n = 39$ (DAG), $n = 40$ (MAG). Data assessed by One-way ANOVA followed by Tukey's multiple comparison test and horizontal line indicates median. **h** Quantification of ILP2HF fluorescence intensity in the IPCs of control (*w⁻*; +/+; gd2HF/gd2HF) and *vaha* mutant flies (*w⁻*; *vaha/vaha*; gd2HF/gd2HF) fed either TAG, DAG, or MAG. Each data point represents fluorescent intensity in one IPC, $n = 47$ (control TAG), $n = 45$ (control DAG), $n = 46$ (control MAG), $n = 47$ (mutant TAG), $n = 44$ (mutant DAG), $n = 48$ (mutant MAG). Data assessed by One-way ANOVA followed by Tukey's multiple comparison test and horizontal line indicates median. **i** A model of Vaha-mediated inter-organ communication in ILP2 release. Vaha (dark blue circles) secreted from R4 region of the midgut (light blue) concentrates in IPCs (green) in a process requiring LTP (black circles) from the fat body (yellow) to regulate ILP2 release. Source data are provided for 8b–h as a source data file.

IPCs. The stimulation of Vaha expression by dietary fat to modulate ILP2 release and the involvement of LTP suggest chylomicrons whose concentration increase after a meal could be hydrolyzed locally by secretory lipases like Vaha. These could move to the IPCs as needed to generate lipid signals for amplification of insulin secretion. Our results showing Vaha functions physiologically as a DAG lipase align with the fact that chylomicrons in *Drosophila* are rich in DAG, unlike mammalian chylomicrons that are rich in TAG[43]. Vaha reporter experiments show that it is synthesized in the R4 region of the posterior midgut, a region that specializes in absorptive functions. Sexual dimorphism in the R4 region of the midgut has been observed particularly for carbohydrate metabolism genes[15]. Since Vaha is expressed in the R4 region, it would be of interest in the future to explore if there could be sex differences in intestinal Vaha gene expression. Among the different cell types in the gut, the intestinal stem cells, hormone secreting enteroendocrine cells, digestive and absorptive enterocytes, the latter are the predominant cell type and have large nuclear size. Based on these two criteria, we speculate, the enterocytes of the R4 region of the midgut secrete Vaha. While many hormonal factors are secreted by enteroendocrine cells in the gut, much less is known about the involvement of enterocytes in secretion of factors that communicate with other organs to maintain metabolic homeostasis. The idea that enterocytes from the different regions of the *Drosophila* midgut are not homogenous and perform distinct functions is increasingly being appreciated[72].

Sequence-based BLAST searches to identify vertebrate orthologs of Vaha show modest identity to acid lipases, including human or mouse lysosomal acid lipase (38%), gastric TAG lipase (38%), Lip M, K, N and J (37–35%). Through enzyme assays with purified Vaha protein, we observed that its activity is higher at pH 7.4 than acid pH, suggesting it likely does not function as an acid lipase. Whether there is a functional mammalian homolog of Vaha that participates in regulating insulin secretion remains to be determined. In mammals, the gut modulates insulin level by secreting the incretins, glucose-dependent insulinotropic polypeptide (GIP), and glucagon-like peptide 1 (GLP-1)[73]. GIP and GLP respond to ingested glucose (and fat) by acting directly on the pancreatic β cells. Since their incretin effect is diminished in diabetes, both GLP and GIP are of therapeutic importance in its treatment[73]. GIP and GLP-1 homologs have not yet been described in *Drosophila*, and an exciting possibility is that Vaha could assume one of their functional roles in fat responsive ILP release. Interestingly, a recent study identified *Drosophila* Neuropeptide F secreted by the midgut enteroendocrine cells as the first sugar responsive incretin hormone in invertebrates, analogous to mammalian GLP-1[74]. Discovering how secretory factors from other organs can influence insulin-producing cells to maintain glycemic control continues to be valuable in understanding the pathophysiology and management of diabetes.

## Methods

### Fly stocks and husbandry

*Drosophila* stocks were raised on corn meal agar and maintained at 25 °C (agar 7.9 g, yeast 27.5 g, cornmeal 52 g, dextrose 110 g, 2.4 g of tegosept in 9.2 ml of 90% ethanol and 1.2 L of water).

5-7 day old flies were used in all experiments unless otherwise indicated. Experiments involving coconut oil (HFD) were performed at room temperature. Crosses involving conditional *Gal80ts* dependent expression of transgenics were incubated at room temperature until adult flies eclosed out of the pupal case. Adults were kept at room temperature for 5-7 days before they were incubated at 29 °C for 10-12 days prior to dissection. Tubulin Gal4 (5138), fat body Gal4 (6982), tubulin Gal80ts (7018), LTP RNAi (51937), mCherry NLS (38424), Mex 1 Gal4 (91368), BBB Gal4 (50472), deficiency (7879), nos Cas9 (54591), UAS NaChBac (9469), CaLexA (66542), *ilp2,3* mutants (30888), and

*ilp2,3,5* mutants (30889) were obtained from Bloomington Stock Center. 8093 RNAi (19561), UAS LRP1 RNAi (8397) and UAS Megalin RNAi (36389) were obtained from Vienna Drosophila Resource Center. NP1 Gal4 flies on the second chromosome were obtained from Dr. Tony Ip's lab (UMass Worcester)[75]. gd2HF flies on the third chromosome were obtained from Dr. Seung Kim's lab (Stanford University). *vaha* mutant, UAS Vaha GFP, UAS VahaΔ30 GFP, Vaha V5, VahaΔ30 V5, Vaha Gal4, and Vaha active site mutant flies were generated in this study as described below.

### Generation of UAS Vaha GFP, UAS VahaΔ30 GFP and UAS-Vaha active site mutant GFP transgenic flies

A fusion protein of C-terminally EGFP tagged Vaha was designed. For this the open reading frame of CG8093 was fused in frame with cDNA sequence of EGFP (GenBank: AAB02574.1) with a 13 aa linker sequence coding for AVDGTAGPGSIAT between the two sequences. The cDNA coding for the fusion protein was cloned into pUAST using Not1-Xho1 site (synthesized and cloned by GenScript). Similarly, C-terminally EGFP tagged Vaha with sequences for the first 30 amino acids deleted from the N-terminus was synthesized and cloned into pUAST as a Not1-Xho1 fragment by GenScript. Vaha active site mutant construct tagged with C-terminal EGFP was generated wherein active site residues Serine (167), Aspartic acid (341) and Histidine (374) were replaced by Alanine and cloned as Not1-EcoR1 fragment into pUAST vector (synthesized by GenScript). The constructs were injected into *w1118* embryos to generate transgenic flies (BestGene Inc).

### Generation of genomic Vaha V5 and genomic VahaΔ30 V5 transgenic flies

1739 bp genomic fragment (containing protein-coding sequence of Vaha together with 0.5 kb upstream regulatory region) was amplified by single fly PCR and cloned as EcoR1-Kpn1 fragment into pUAST vector (the upstream oligo 5′-GCG AAT TCT AAT TAT AGT AAT CAC TTT AAA AT-3′ maps 473 bp upstream of the start of CG8093 with an EcoR1 site; and the downstream oligo 5′-AAA ATA GGT ACC TTA CGT AGA ATC GAG ACC GAG GAG AGG GTT AGG GAT AGG CTT ACC TGC ATT ATT TAT GTC GTT AAT TAC C-3′ has the sequence for V5 inserted just before the stop codon). The clone was confirmed by sequencing.

The VahaΔ30 V5 having the upstream sequence and V5 epitope tag as above but lacking the coding sequence for the first 30 amino acids of CG8093 was synthesized and cloned as an EcoR1-Not1 fragment into pUAST by GenScript. Transgenic flies were generated by BestGene Inc. Although the above two clones were generated in pUAST vectors, the endogenous promoter drives the expression of these genes without the need for Gal4 driver.

### Generation of Vaha Gal4 transgenic flies

The Vaha-Gal4 construct includes DNA 2 kb upstream of Vaha start codon (the sequence including 2 kb upstream of the Start and 2 kb downstream of the Stop codons respectively is listed as 'extended gene sequence' of CG8093 in Flybase) that is fused in frame to the ATG of the Gal4 sequence (Gen Bank: K01486.1) and cloned as a Not1-EcoR1 fragment into pUAST (synthesized by GenScript). Transgenic flies were generated by BestGene Inc.

### Generation of *vaha* mutant using the CRISPR-Cas9 system

Due to the lack of a *Vaha* mutant fly line, the CRISPR-Cas9 system was used to generate a deletion mutant of the gene. Two guide RNAs targeting the coding regions, towards the N-terminal and C-terminal parts, were expressed using a single vector (Supplementary Fig. 10a). The targeting vector was generated within the framework of the pCFD4 vector that utilizes U6-1 and U6-3 promoters to drive the expression of the guide RNAs[50]. The vector was assembled using high fidelity PCR and Gibson assembly as described[50]. A transgenic fly line

generated by BestGene Inc was utilized to induce germ line knockout by crossing with nos Cas9 transgenic flies. The resultant mutant flies were balanced and backcrossed to $w^{1118}$ flies ($n = >5$) to isogenize and outcross off-target mutations.

## Immunohistochemistry

Immunostaining of fly brain: The brains were dissected in S2 cell media (ThermoFisher Scientific R69007), fixed in 4% paraformaldehyde in PBS for an hour, and blocked in PBS containing 0.1% Triton X-100 and 1% BSA overnight at 4 °C. Samples were incubated with primary antibodies overnight at 4 °C (mouse anti GFP B-2 1:100 Santacruz Biotech sc9996; rabbit anti GFP 1:100 Chromotek pabg1; anti V5 1:100 EMD Millipore AB3792; rabbit anti HA C29F4 1:500 Cell Signaling Technology 3724; mouse anti HA 6E2 1:500 Cell Signaling Technology 2367). Samples were washed in PBST and incubated overnight at 4 °C with secondary antibodies (1:1000 goat anti mouse Alexa Fluor Plus 488 nm or 555 nm; goat anti rabbit Alexa Fluor Plus 488 nm or 555 nm Invitrogen A32723, A32727, A32731, A32732). Samples were washed in PBST followed by PBS and mounted in Vectashield containing DAPI (Vector Labs H-1200). Samples were imaged at 20X on a Zeiss LSM880 confocal microscope or 60X on a Nikon spinning disk confocal microscope.

Immunostaining of fly gut: The guts were dissected in PBS, fixed in 4% paraformaldehyde for an hour, and permeabilized with 0.1% Triton X-100. 5% normal goat serum was used for blocking and in antibody solutions that contained PBS and 0.1% TX-100. Samples were incubated with primary antibodies overnight at 4 °C (rabbit anti-DsRed 1:100 Clontech 632596; mouse anti-Cut 1:10 cell culture supernatant DSHB 2B10). Samples were washed in PBST and incubated with Alexa conjugated secondary antibodies overnight at 4 °C (donkey anti-rabbit 568 nm 1:100 ThermoFisher A10042; donkey anti-mouse 647 nm 1:100 ThermoFisher A31571). Washed samples were incubated with DAPI for 5 min, washed 3X in PBST, and mounted in Vectashield. They were imaged at 10X on a Leica-Andor spinning disk confocal microscope.

## Image acquisition and quantification

The same confocal settings were applied for control and experimental samples. Fiji was used to process confocal images. For quantification of fluorescence intensity in the IPCs of the brain, maximum projection image of z stacks in gray scale was analyzed by Fiji. The freehand tool in Fiji was selected to draw individual cells and the mean pixel intensity was calculated after subtracting the background intensity. To visualize the whole gut, after z-stacked images were obtained, sets of image fields with 5% overlap were stitched to obtain final composites of individual gut using native software. The projected images were viewed using Imaris Image Viewer and assembled using Photoshop and Illustrator.

## Western blotting

For SDS-PAGE, fly samples (either body, head, or gut) were homogenized in 2xLaemmli sample buffer containing β-mercaptoethanol. 10% SDS-PAGE gels were run for all samples except for ILP2HF samples which were separated on 15% gels. Proteins were transferred to PVDF membranes and probed overnight at 4 °C with primary antibodies. The secondary antibody incubation was for 2 h at room temperature. The blots were developed with Western Lightning Plus ECL (Perkin Elmer NEL103001). The primary antibodies used were rabbit anti GFP 1:1000 Chromotek pabg1; mouse anti-V5 1:500 Invitrogen 46-1157; mouse anti actin 1:1000 DSHB JLA20; and mouse anti actin 8H10D10 1: 1000 Cell Signaling 3700. The secondary antibodies used were goat anti rabbit HRP or goat anti mouse HRP 1:3000 or 1:5000 Jackson ImmunoResearch). Fiji was used for the quantification of Western blots. Uncropped, unprocessed scans of full blots are provided in the Source Data file.

## Feeding experiments

Preparation of HFD: The agar in regular food was increased to 9 g/L to keep the food solidified after addition of coconut oil and prevent flies from sticking to the food. 225 ml of Coconut oil (organic triple filtered, Trader Joe's) liquified in a warm water bath was added to 1 L of yellow food and mixed thoroughly until the contents were homogenous. 10 ml of food was dispensed per vial and was left to solidify overnight at 4 °C before use the next day. Water was used instead of coconut oil in the control food.

Preparation of 2 M sucrose food: During preparation of regular food, sucrose was added as a solid to 2 M concentration at the last step, and the contents were thoroughly mixed. For all short-term feeding experiments (1–3 h), 0.1% bromophenol blue was added to the food and flies whose guts were dyed blue were used for experiments.

For feeding experiments, flies were starved overnight for 16 h (20 flies per vial) supplemented with water via a wet sponge at the bottom of each vial or 1% agar. 10 flies were then transferred to regular food and 10 flies to regular food containing coconut oil or 2 M sucrose. For feeding lipids, TAG, DAG, or MAG (Chem Service Inc NG-S237, NG-S236, NG-S235) was added at 10% (approximately 100 mM) to regular food. Flies were starved overnight and then transferred to lipid-supplemented food for 2–3 h before dissection.

## ELISA for estimation of circulatory ILP2HF

ILP2 level in hemolymph was measured as previously described[37]. A day before the experiment, 8-well Nunc-Immuno modules (Thermo Fisher 468667) were coated with FLAG antibody (Sigma-Aldrich F1804) in 200 mM NaHCO₃ buffer for 14–16 h at 4 °C. This was followed by washing with PBST (0.2% Tween20) and blocking in 1% BSA for 2 h at room temperature. 50 µl of diluted anti-HA peroxidase (Roche 12013819001, 1:2000) in PBS containing 1% Triton X-100 was added to the FLAG coated wells and stored in a wet slide chamber until samples were ready. Hemolymph samples from male flies (80 flies per sample) were collected and transferred to an eppendorf tube containing 60 µl of PBST. Alternatively, the posterior end of male abdomens (15 flies/sample) were dissected and transferred to a tube containing 60 µl of PBST. Samples were incubated in a microtube holder with vortexing at low speed for 20–30 min. After spinning down, 50 µl of supernatant was transferred to coated wells and incubated overnight at 4 °C. The plate was washed with PBST. A total of 100 µl of 1-step TMB-ELISA substrate (Thermo Fisher 34029) was added and incubated for 30 min at room temperature. The reaction was stopped by adding equal amount of 2 M sulfuric acid. The absorbance was measured at 450 nm in a plate reader (Biotek Synergy HT) or a Multi-Mode Microplate Reader (SpectraMax iD3, Molecular Devices).

## Glucose and TAG estimations

Flies (10 flies/sample) were homogenized in PBS containing 0.1 % Triton X-100 using pellet pestle and heated at 70 °C for 10 min. The homogenate was centrifuged at $18,000 \times g$ for 15 min at 4 °C. Whole body glucose was measured using Hexokinase reagent (ThermoFisher Scientific, TR15421) and TAG estimation was carried out using triglyceride reagent (Fisher Scientific, SB- 2100-430). All values were normalized for protein content, which was quantified using Bradford Reagent (Bio-Rad, 500006).

## Oral glucose tolerance test (OGTT)

Flies (40 flies/ sample) were starved for 16 h on 1% agar and transferred to vials containing 1% agar + 15% dextrose for 1 h. After 1 h of feeding, flies were re-starved on 1% agar for 1 h. Hemolymph was collected from initial starvation, after 1 h feeding, and after re-starvation for 1 h. The level of circulatory glucose was measured using Hexokinase reagent (ThermoFisher Scientific, TR15421).

## CaLexA measurements

Flies bearing *LexAop-CD8-GFP-2A-CD8-GFP* on the second chromosome were recombined to the *vaha* mutant. CaLexA was expressed in the IPC neurons using dILP2 Gal4 in control and *vaha* mutant flies. Both sets of flies also carried an additional *LexAop-CD2GFP* reporter on the third chromosome and ILP2HF to mark the IPCs. Control and *vaha* mutant flies were starved overnight, transferred to regular or high fat food, brains were dissected, fixed, and stained. CaLexA driven GFP signal in the brains was enhanced by staining with rabbit anti GFP antibody (rabbit anti GFP 1:200 Chromotek pabg1).

## Metabolic profiling

One hundred flies each of 30–35 day old $w^{1118}$ and *vaha* mutants (per replicate) were collected and frozen. The samples were analyzed in triplicate by Ultrahigh Performance Liquid Chromatography-Tandem Mass Spectroscopy (UPLC-MS/MS) by Metabolon Inc (Morrisville, NC). Lipids were extracted from samples via a modified Bligh-Dyer extraction using methanol/water/ dichloromethane in the presence of deuterated internal standards. The extracts were concentrated under nitrogen and reconstituted in 0.25 ml of 10 mM ammonium acetate dichloromethane:methanol (50:50). The extracts were transferred to inserts and placed in vials for infusion-MS analysis, performed on a Shimazdu LC with nano PEEK tubing and the Sciex SelexIon-5500 QTRAP. The samples were analyzed via both positive and negative mode electrospray. The 5500 QTRAP scan was performed in MRM mode with a total of more than 1100 MRMs. Individual lipid species were quantified by taking the peak area ratios of target compounds and their assigned internal standards, then multiplying by the concentration of internal standard added to the sample. Lipid class concentrations were calculated from the sum of all molecular species within a class.

## Metabolomics data analysis

Metabolomics data analyses were carried out using R (version 4.0). Raw concentration data of metabolites detected by UPLC-MS/MS was first filtered to keep only metabolites detected in at least three samples and then *log*-transformed. The minimal concentration ($C_{min}$) and standard deviation (SD) for each metabolite were calculated and used for imputation of the corresponding metabolites as follows. Missing values for a given metabolite were imputed with random numbers from a normal distribution, $N(\mu = C_{min} - 2 * SD, sd = SD)$. A smooth quantile normalization (PMID: 29036413) was performed on the imputed data and the resulting data was used for downstream analyses. Differential abundance analysis was performed using the limma package (PMID: 25605792). The nominal *p* values were adjusted using the Benjamini–Hochberg (BH) method ([https://www.jstor.org/stable/2346101](https://www.jstor.org/stable/2346101)). Metabolites with adjusted *p*-values ≤ 0.05 and absolute log2 (fold change) ≥ 0.59 were visualized as a heatmap by using the pheatmap package.

## Generation of stable cells

Schneider's Drosophila Line 2 (D.Mel. (2), SL2) was obtained from ATCC, CRL-1963. These cells were cultured in Schneider's Drosophila medium (Thermo Fisher R69007) supplemented with 10 % FBS, 10 units/ml penicillin, and 10 μg/ml streptomycin at 25 °C. For generation of V5 Vaha stable cell line, Vaha cDNA was cloned into pMTV5-HisA vector as an EcoR1-Xho1 fragment. Stable lines were generated using standard techniques[76]. S2 cells were transfected with Vaha expression vector and selection vector pCoHygro using Lipofectamine. Hygromycin B was used to select for stable transfectants. Media was replaced every 4 days until resistant cells appeared in 3-4 weeks. Resistant cells were replated into 6-well plates, expanded, and induced with copper sulfate for expression, master stocks were prepared from cells with high expression and frozen.

## Expression, and purification of Vaha from stable S2 cells

Stable S2 cells expressing Vaha C-terminally tagged with V5 and His were stimulated with 1 mM $CuSO_4$ for 48 h at 25 °C. The culture supernatant was subjected to 50% ammonium sulfate precipitation. The resulting pellet was dissolved in 50 mM Tris-HCl (pH 7.4) containing 2 mM β-mercaptoethanol, protease cocktail inhibitor, and dialyzed against 50 mM Tris-HCl, pH 7.4, containing 2 mM β-mercaptoethanol for 12 h at 4 °C. The dialyzed protein sample was loaded onto a nickel affinity column (HisTrap HP 1 ml GE Healthcare 17-5247-01). The column was pre-equilibrated with 50 mM Tris HCl, pH 7.4, containing 20 mM imidazole, and 500 mM NaCl and eluted with 50 mM Tri-HCl containing 400 mM imidazole, 2 mM β-mercaptoethanol, and 500 mM NaCl. The pooled fractions were concentrated using 10 kD protein concentrator and the resulting partially purified protein was subjected to Fast Protein Liquid Chromatography on a Superdex 200 column. The eluted lipase fractions were pooled and dialyzed against 50 mM Tris-HCl, pH 7.4.

## Determination of Vaha enzyme activity using different substrates

Vaha activity was measured using 3 different substrates - [³H] TAG (Perkin Elmer NET431001MC), [³H] DAG (American Radiolabeled Chemicals ART0643) and [³H] MAG (American Radiolabeled Chemicals ART1158). The cold substrates were from Cayman Chemical Company TAG 26871, DAG 26896, MAG 16537. The cold substrates were emulsified with 5 mM CHAPS and sonicated for 5-10 sec using a probe sonicator (Ultrasonic Processor, Cole-Parmer). To determine substrate specificity of lipase, we used 10 μM-100 μM of TAG/DAG/MAG and 0.1% of respective radiolabeled [³H] substrate in the total substrate concentration. The FPLC purified enzyme was added to the assay mixture containing 50 mM Tris-HCl (pH 7.4), cold substrate, and [³H] substrate in a total volume of 200 μl. The reaction mixture was incubated at 30 °C for 15 min and terminated by adding acidified water and 0.6 ml of chloroform: methanol (1:2 v/v). The lipids were extracted using the Bligh and Dyer method and the extracted lipids were separated on TLC silica gel plates (Sigma, 1055540001) using the - petroleum ether: diethyl ether: acetic acid (70:30:1 v/v) solvent system. The released fatty acid was quantified in a scintillation counter (TRI-CARB 4910TR, PerkinElmer).

## Quantitative reverse transcription PCR

To measure changes in gene expression, total RNA was extracted from 10 whole flies, 30 guts, or 40 heads by TRIZOL RNA isolation protocol. Total RNA was extracted from 10 brains and 10 CNS using PicoPure RNA isolation kit (ThermoFisher Scientific). The isolated RNA samples were reverse transcribed by transcriptor first strand cDNA synthesis kit (Roche). Gene expression was analyzed by real-time qPCR using the iTaq universal SYBR Green supermix (BIO-RAD) in a QuantStudio™ 5 Real-Time PCR System (Applied Biosystems). The RT-qPCR cycling conditions are as follows: initial denaturation at 95 °C/5 min followed by denaturation (95 °C/15 sec) and annealing and extension (60 °C/60 sec) for 40 cycles. GAPDH was used to normalize Vaha gene expression and relative fold changes were calculated by comparative Ct method 2^-(ddCt). The primers used to quantify *Vaha* transcript levels were forward 5'GAACTCGCCATATATCAAC3' and reverse 5'GTAGATCCGTAGTCGAATTG 3'. The primers for *ILP2* were forward 5' ATGAGCAAGCCTTTGTCCTTC3' and reverse 5' ACCTCGTTGAGCTTTTCACTG3'. The primers for *LTP* were forward 5'AATCCGAGTCGCAAAACTATGAT3' and reverse 5'GCCGTCGTGGAAAGCAAAT3'. The primers used for GAPDH were forward 5'TAAATTCGACTCGACTCACGGT3' and reverse 5'CTCCACCACATACTCGGCTC 3'.

## In situ hybridization chain reaction (HCR)

HCRs were performed following the instructions provided by Molecular Instruments (MI) and based on published protocols[77]. Probes for

in situ HCR for *Vaha* (CG8093), *Myo31DF* (CG7438) and the hairpins were designed and manufactured by MI. The probe set size was 20 for each gene and the amplifiers were B2 (Alexa fluor 647) for *Vaha* and B3 (Alexa fluor 546) for *Myo31DF*. The transcripts for the two genes were used for designing the probes. While the company does not provide the individual probe binding sequences, the lot numbers provided by the company for the probes are as follows: CG7438 (*Myo31DF*) – PRG995 and for CG8093 (*Vaha*) – PRG996 and will aid in design and reproduction of the probes used in the study. The probes are designed to maximize target specificity while minimizing off-targeting complementarity. Male and female adult guts and adult wandering third instar larvae were dissected in PBS, fixed in 4% paraformaldehyde for an hour and washed in PBS containing Triton X-100. After methanol, ethanol and xylene treatments, the guts were incubated in hybridization buffer for 30 min at 37 °C and then with 16 nM of each probe overnight. The following day the samples were washed in 5X SSCT buffer and incubated with heated and snap-cooled hairpin amplifiers at room temperature in the dark. The following day, the excess hairpins were removed by washing in 5X SSCT, and after the final wash, the samples were incubated with DAPI for counterstaining the nuclei and mounted in Vectashield for microscopy. The images were viewed at 10X in Leica-Andor spinning disk confocal microscope. Z-stacked images of gut with a minimum of 5% overlap for individual fields were stitched into a composite using native software. The projected images were viewed with Imaris Image Viewer and assembled in Photoshop CC.

## CAFE assay

The capillary feeding assay was used to determine food consumption[78]. For the feeding assay, 3-day-old flies were collected and transferred to fresh food vials for 48 h (20 flies per vial) before starvation overnight. The feeding system was created by cutting a vial plug in half and making 4 holes. 20 µl pipette tips were cut and inserted in the plug holes to hold marked glass capillaries (VWR 53432-706). Holes were made in the vial for ventilation and a wet sponge was kept at the base of fly vial to retain moisture. The base food for feeding was 5% w/v sucrose with 5% w/v yeast extract. Tubes with filled capillaries but without flies were kept as evaporation control. Food consumed was tracked as volume change in capillary tubes.

## Developmental timing experiments

Adult flies were allowed to lay eggs for 24 h at 25 °C on apple juice plates. After hatching, *w¹¹¹⁸* and *vaha* mutant first-instar larvae were collected from the plates and transferred to food vials. Developmental stage and timing were monitored until adult flies were closed.

## Statistics

GraphPad Prism 10 software was used to plot data and perform statistical analysis. Data were analyzed either by the two-tailed Student t-test or paired t-test for pairwise comparison and one-way ANOVA with a Tukey multiple comparison test for multiple comparisons. Exact *p* values have been provided and $p < 0.05$ was considered as statistically significant.

## Reporting summary

Further information on research design is available in the Nature Portfolio Reporting Summary linked to this article.

## Data availability

Source data for the experiments are provided as a source data file with this paper. Metabolomics and Lipidomics data that are provided in Supplementary Data 3 and 4 have also been deposited to the EMBL-EBI MetaboLights database https://www.ebi.ac.uk/metabolights/MTBLS8441 (DOI: 10.1093/nar/gkz1019, PMID:31691833) with the identifier MTBLS8441. Source data are provided with this paper.

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

## Acknowledgements

We thank the Bloomington Drosophila Stock Center, the Vienna Drosophila Resource Center for fly stocks, and the Developmental Studies Hybridoma Bank for antibodies. We thank Dr. Seung Kim and Dr. Takashi Nishimura for generously sharing reagents and fly lines. We are grateful to Dr. Ira Daar for his support and valuable discussions. We thank Drs. Susan Mackem, Mark Lewandoski, Terry Yamaguchi, and Erin Davies for insightful suggestions. We thank Kenneth Kim for technical support, OMAL for microscopy assistance, and Dr. Sitara Niranjan for helpful discussions. AS was supported by grant R01GM110288 to URA. This study was funded by the Center for Cancer Research (JKA Intramural Research Program), National Cancer Institute, NIH, Division of Health and Human Services. The content of this publication does not necessarily reflect the views or policies of the Department of Health and Human Services, nor does mention of trade names, commercial products, or organizations imply endorsement by the U.S. Government.

## Author contributions

The study was conceptualized by U.R.A. and J.K.A. Methodology and design of experiments were by A.S., K.V.A., K.R.A., H.L., N.K.N., V.P., G.K., T.A., J.T., L.J.Z., J.K.A., U.R.A. Experiments were performed and analyzed by A.S., K.V.A., K.R.A., V.P., G.K., T.A., J.T., J.K.A., U.R.A. Analytical tools were contributed by H.L., N.K.N., L.J.Z. The manuscript was written by U.R.A. with input from K.V.A., while other authors edited the manuscript.

## Funding

## Competing interests

The authors declare no competing interests.
