## [Peer Review File · Nature Communications]

A nutrient responsive lipase mediates gut-brain communication to regulate insulin secretion in *Drosophila*REVIEWER COMMENTS

Reviewer #1 (Remarks to the Author):

This study shows that the lipase Vaha is secreted from enterocytes in the gut and travels to the insulin producing cells (IPCs) in the brain where it regulates insulin-like peptide 2 (DILP2) secretion. Although sugar is a main factor that promotes insulin secretion, dietary fat also affects insulin release. In here, the authors show that Vaha is a diacylglycerol (DAG) lipase that is released from the gut in response to intake of dietary fat and accumulates in the IPCs to modulate insulin secretion, providing a possible mechanism by which dietary fat can influence secretion of insulin. Conceptually, this is very interesting and I would like to see this work being published!

However, the data do not fully support the conclusions, although the authors provide several lines of evidence. I therefore suggest a major revision to strengthen the conclusions before publication. A key finding is that Vaha is produced by the gut and accumulates in the IPCs, and that this is mediated by LTP-dependent transport. This is based on expression of GFP-tagged Vaha using the NP1-Gal4 and an endogenously tagged Vaha transgene. They need to include the UAS-Vaha::GFP alone without any drivers to exclude that it is not leaky and that there is no expression in the IPCs. Furthermore, they need to show that the endogenously tagged Vaha is not expressed in the brain or IPCs. One thing is the overexpression, but they should also knock Vaha down specifically in the gut and show it eliminates the IPC Vaha signal. It would also be nice to use a second gut driver and a second RNAi lines as well as perform several more experiments with tissue-specific RNAi in addition to the Vaha null mutant. It is also possible to show that gut Vaha accumulates in the IPCs ex vivo by co-culturing guts and CNS.

The Vaha accumulation on the IPCs is dependent on LTP, but how do they explain that LTP is not required for transport of Vaha into the brain, but only its accumulation in the IPCs? In the paper they author mention, it is shown that LTP are required for trafficking across the blood-brain barrier (BBB). Thus, it would be expected that if LTP transports Vaha to the IPCs that is required for Vaha to cross the BBB and enter the brain. (I think they should also show this in a more quantitative manner, by quantifying the amount of Vaha in the IPCs and CNS

in animals with fat body-specific LPT knockdown. Generally, there are several experiments where conclusions are based only on single image, which needs to be confirmed by replicates and quantification). LpR1 and LpR2 transport LTP across the BBB 1, and thus, presumably also Vaha. This should be tested by double knockdown of LpR1/2 in the BBB glia. It could also be interesting to see whether LpR1/2 are required for Vaha accumulation in the IPCs. It somehow needs to be confirmed that gut-derived Vaha is secreted into circulation and enter that brain. One of the weaknesses is that they do not know the mechanism by which Vaha is recruited by the IPCs, or even its uptake into the brain. Perhaps it is mediated by LpR1/2.

A concern is that Vaha is a lipase, and it is difficult to tease out whether the metabolic changes of Vaha deficient animals are a consequence of altered DILP2 signaling or whether the altered DILP2 signaling is an indirect consequence of metabolic defects in Vaha deficient animals. They show that Vaha mutant animals have altered metabolism, but they need to show that Vaha regulates metabolism (e.g. glucose) through insulin. If Vaha regulates metabolism through DILP2, changing Vaha signaling should have no metabolic effect in a DILP2 mutant. On the other end, activation of IPC activity should rescue metabolic defects of Vaha mutants. I think they should test whether Vaha affects for example glucose tolerance by performing a glucose-tolerance test (GTT). If the hypothesis is correct Vaha is important for GSIS then it should affect glucose tolerance. Of course, this should be dependent on insulin, so loss of Vaha should have no effect on GTT in a DILP2 mutant or animals in which the IPC activity has been silenced. I think it is important to demonstrate that Vaha is indeed affecting glucose metabolism through insulin.

Fig. 2d. Higher resolution images are necessary, also with higher magnification.

Fig. 2e. I agree these are likely enterocytes that produce Vaha in the R4, but it should be shown.

Why do they not show the LPT knockdown data?

For some figures like Fig. 3c,d all the controls are normalized to 100. How is that possible

and it seems to me that you cannot make any statistical analyses based on this way of analyzing the data.

Fig. 3d. Vaha levels goes up in head extracts in animals fed high fat. It should be shown that the gut is the source. For example, knock Vaha down in the gut and show that it does not accumulate in the brain on high fat diet. I think generally for all experiments it is cleaner to use dissected brains or even CNS than using head extracts. The head also contains a lot of fat body, so it is possible that Vaha accumulates in the head fat instead of in the brain itself.

They should also examine Vaha transcript levels on different diets, both in dissected CNS and guts.

Fig. 3e. IPC staining should be quantified using several replicates.

The authors state that Vaha mutants they generated are no different in size, weight, or lifespan without showing the data. They should show that data. However, it is puzzling why all these parameters which are influenced by DILP2 are not affected by loss of Vaha, if Vaha regulates DILP2 as the authors claim? Later on, they actually look into whether Vaha affects growth, even though they already excluded it?

They need to quantify data in Supplementary Fig. 3a. It is not possible to conclude that Vaha mutation does not affect DILP2 from a single Western blot without replicates. It is also much better to do this by staining the IPCs and quantifying DILP2 in the IPCs.

Fig. 4a,b. I am also not sure what comparison they make to say the failure to secrete insulin in response to dietary stimulation is worsened in high fat food. Instead of rescuing the phenotype of Vaha mutants by ubiquitous expression using the Tubulin-Gal4 they should be able to rescue with a gut-specific driver.

Fig. 4b,c. Why is the control fasted not compared to control refed reg or HFD?

I am not sure I understand this correctly, but it seems that they show in Supplementary

Fig.3a that refeeding after starvation does not change DILP2 in heads, but then they show that refeeding does change it in Fig, 4b as would be expected of course.

They should show Vaha in circulation, since the protein is large enough for hemolymph detection and they have a tagged version. This can be done by Western blots or perhaps even ELISA.

I am not sure how they conclude that high fat diet enhances GSIS? That seems to require actual feeding glucose or injection of glucose and measuring circulating insulin.

Fig. 6a. Like many other places this needs to be quantified. Showing one image is not sufficient to conclude that Vaha lipase activity is essential for DILP2 accumulation.

Fig. 6h. It should be shown that TAG, DAG and MAG cannot induce DILP2 release in animals with gut-specific Vaha loss.

Since the study focus on insulin secretion it is important to measure IPCs activity, which can be done using e.g. CaLexA to show that Vaha loss decrease IPCs calcium activity in response to glucose after high fat diet.

If the hypothesis is correct that Vaha is important for GSIS then it should affect glucose tolerance. This can be tested by performing a glucose-tolerance test (GTT). Of course, this should be dependent on insulin, so loss of Vaha should have no effect on GTT in a DILP2 mutant or animals in which the IPC activity has been silenced. I think it is important to somehow demonstrate that Vaha is affecting glucose metabolism through insulin.

Gut Vaha overexpression should lead to changes opposite the loss of function, thus more DILP2 secretion. This needs to be tested.

They need to put the Vaha mutation over a deficiency or another mutant allele. As a geneticist this is the right way to confirm that the phenotype is origination from the Vaha locus. The specific problem with CRISPR is that all mutant alleles (if the same gRNAs are

used) will have the same off-targets. Now I understand that it was backcrossed, but still using a Deficiency is better especially since many of the experiments rely on this mutant.

It needs to be shown that Vaha-RNAi using NP1-Gal4 reduces Vaha transcript levels in the gut without affecting the CNS (brain and ventral nerve cord).

Minor points:

In fig. 1d it is not possible to see unit on scale bar and other figures miss a scale bar.

1 Brankatschk, M., Dunst, S., Nemetschke, L. & Eaton, S. Delivery of circulating lipoproteins to specific neurons in the Drosophila brain regulates systemic insulin signaling. *Elife* 3, doi:10.7554/eLife.02862 (2014).

Reviewer #2 (Remarks to the Author):

The work by Singh et al entitles “A nutrient responsive lipase mediates gut-brain communication to regulate insulin secretion in Drosophila” is the characterization of CG8093, that they name “vaha”. This gene had been partially characterized before, in a previous publication (reference 40 in their manuscript), where they reported that UAS-CG8093 RNAi ubiquitous expression (possibly via an actin-Gal4 driver) is lethal, and that midgut specific expression via *esg*-Gal4 leads to viable flies, albeit with increased TAG, and reduced resistance to starvation. In this paper they use a different Gal4 driver, NP1-Gal4, which is an insertion in the myosin *myo31DF*.

Major points:

1) They perform only a rescue experiment using the *ilp2* secretion phenotype after a fasting/refeeding paradigm, and this by means of ubiquitous expression of *vaha*, and do not perform rescue experiments on the three other phenotypes observed in the CRISPR/Cas allele of *vaha*, which is purportedly a null allele. The mutant phenotypes described for the *vaha* null are: reduced weight at end of first instar larvae, developmental delay, metabolites imbalances in aged flies, and reduced response of *ilp* release in response to dietary fat. Only

this last get rescued. The authors should try to rescue the growth delay, reduced weight, and study whether some of the metabolic changes are reverted in a rescued fly.

2) They use normally first week adults for most experiments, but use 30-35 days old flies for metabolic studies. Are younger flies not affected? Have the authors done a developmental profile?

3) Expression is studied using a tagged construct, and mention is made in reference 40 of another work (Zinke I, Schutz CS, Katzenberger JD, Bauer M, Pankratz MJ (2002) Nutrient control of gene expression in *Drosophila*: Microarray analysis of starvation and sugar-dependent response. *EMBO J* 21: 6162–6173.) where in situ hybridizations showed expression in the midgut, and PCR data in this paper. Yet, flybase shows expression in many other tissues, including neurons, which would be in line with the RNAi results that resulted in lethality (reference 40). Is the RNAi data misleading in that it is also targeting other genes? Or, is the null allele obtained not showing all phenotypes? This should be clarified in order to understand and interpret the current loss-of-function allele, and the RNAi data. Can in situ hybridization be performed in most tissues/ stages to clarify the point, as antibodies were not obtained?

4) What is a good control for PCR experiments: GAPDH is a glucose metabolism enzyme, and so, could be affected in the mutants/constructs? In reference 40 they use rp49 instead. The difference in expression in reference 40 from intestine to brain is five times, and here it is 600 times: the difference should be clarified.

5) Why is data not shown in several instances? (e.g., on page 8) Aren't supplementary figures just for that?

6) The authors should acknowledge that experiments carried out using tagged and other expression constructs were done in a wild type background (except the one rescue experiment), that is, that the endogenous vaha expression is also present, and that that might bias results, as they might have gain-of-function effects, especially since no quantification of expression is generally done. In reference 4 they use a different Gal4 construct to drive expression (esg-Gal4) in the midgut, and here they use the NP1-Gal4, which is more widespread in the intestine yet they see only expression in part of the midgut.

7) Figures. In general, confocal figures should be bettered. The ones right now are very dark, with little detail. The same frame or plane of view image in phase contrast, or Nomarki, should be furnished, to make clear where is the signal generated, and the extent of the

tissue(s) surveyed in the pictures. In the paper differences in expression (florescent signals) are compared, but no quantitation is shown; this should be done, together with the n of the experiments, to back the claims in the text. Also, in several figures bars and lettering is often very small and hard to read (e.g., figure 1b). Figures should be revamped to make sure data is legible, figures are clear, adding scale bars where needed, for example. As they stand now, figures are not acceptable.

8) Accumulation of Vaha in brain cells is only partially coincident with ilp2 expressing IPCs, and some ilp2-positive cells do not accumulate Vaha. The authors mention that non-ilp2, Vaha expressing neurons should be characterized in the future, but no mention is made of ilp2-positive, Vaha negative cells. Is Vaha only accumulated in some of the IPC cells? Why?

9) The sentence “We reason, yeast, the major supplier of steryl esters and triglycerides, and cornmeal, a minor contributor, could be the lipid sources that modulate Vaha in regular food.” needs backing.

10) Dilps should be changed to ilps throughout the text, since ilp is the official name of these genes in Flybase.

Other points:

1) Perhaps add references in supplementary table 3.

2) A revision of the text for typos and grammar. In general, the text is fine, but still some typos and error have not been corrected (e.g., We next examined, which processes could be on page 11, without the comma; use of the correct hyphen throughout the text).

3) “Of these 280 w1118 pupae developed to adults and 303 vaha pupae developed to adults.”, from page 29: quantitation can be included.

4) IPC are usually capitalized.

5) Is “overnight” 14-16 hours?

6) Shouldn't a supplementary figure showing the full extent of the gel be furnished for all Western experiments?

7) The depository for the metabolomics data should be mentioned in the text.

Reviewer #3 (Remarks to the Author):

Singh and colleagues present a very original study on a new gut-expressed and dietary-regulated lipase called Vaha, which relocates to the insulin-like peptide (Dilp) producing

cells in the *Drosophila* brain (IPCs) to exert systemic developmental and metabolic effects by controlling Dilp secretion.

The authors use Vaha fusion transgenes and the heterologous Gal4/UAS system to (i) assess the localization of Vaha in the IPCs and (ii) the expression of the protein in the R4 region of the gut but not in IPCs. Global and tissue-specific mRNA expression data in the gut are provided in support of the findings based on transgene data. The authors demonstrate that Vaha localization at the level of protein secretion depends on the predicted Vaha signal sequence and on a functional LTP (provided by the fat body) at the level of protein uptake into the IPCs. The authors show the Vaha protein levels increase in response to high fat diet, which causes Vaha accumulation in the IPCs. Dilp secretion from the IPCs is impaired in global Vaha mutants and upon knockdown of the gene in the gut. Vaha mutants show developmental delay and – after aging - changes in the metabolite profile including increased total body glucose, DAG and TAG levels, while MAG levels are decreased. Recombinant Vaha has lipase activity accepting DAG, MAG and TAG as substrates. Overexpression of an Vaha lipase active site mutant in the gut does not impair IPC targeting but increases Dilp2 accumulation in the IPCs. Finally, the authors demonstrate that dietary supplementation with TAG, DAG and MAG reduced Dilp2 intensity in the IPCs suggesting increased secretion.

This is a very interesting study, which, however, deserves additional experimental support for the fascinating conclusions drawn and the regulatory concept deduced by the authors (see below). To this reviewer the order and prioritization of the presented data is not intuitive and readability would improve by restructuring. For example, recombinant protein data which prove that Vaha is a bona fide lipase and describe the substrate spectrum are presented in the last figure and not in the first along with the expression data. To my perception the authors provide no evidence that Vaha does not serve additional function(s) in the gut as it is expressed in regions and cells involved in lipid metabolism. So why not proceeding from the biochemical characterization to the expression before introducing the exciting finding that the lipase relocates to the IPCs? With regards to priority: Fig. 1 presents the protein localization data, which are also presented in Fig.2 using a different transgene construct. Why this redundancy?

Points of concern:

1) According to Fig. 3, Vaha abundance is responsive to diet. Is this due to transcriptional regulation in the gut? According to the model of the authors, Vaha differential accumulation in the IPCs impact on Dilp secretion. This argues in favor of a vaha gain-of-function phenotype upon Vaha overexpression in the gut (on regular food). Do the authors observe such a phenotype and if so, is this phenotype absent in the *dilp2,3,5* mutant? Moreover, if the IPC-expressed Dilps mediate the developmental defects of vaha mutants, is the vaha *dilp2,3,5* double mutant phenotypelidentical to the *dilp2,3,5* single mutant? This needs to be tested.

2) According to the model Vaha exerts systemic metabolic effects by lipase function in the IPCs, while the expression of the gene in the gut might be required for its responsiveness to diet. The authors should take advantage of the wonderful tools they developed for an extended mechanistic analysis in the vaha mutant background. What are the phenotypes of flies rescued (or not) by targeted expression in the IPCs of Vaha, of the Vaha signal sequence mutant and of the Vaha active site mutant expression?

3) The rescue experiment provided by the authors uses a ubiquitous driver and accordingly is not appropriate to support the model that Vaha is expressed in the gut but acts in the IPCs. How does the fusion protein distribution/accumulation in the IPCs look like under theses circumstances? A rescue experiment targeting vaha to its endogenous expression domain(s) using the Vaha Gal4 driver or to the gut using the NP1 driver is required.

4) If Vaha lipase activity in the IPCs mediates the effect of dietary lipid supplementation (TAG, DAG, MAG) on Dilp release, are the effects shown in Fig. 6h absent in the vaha mutant?

5) The mechanism of Vaha lipase function in the control of dilp release in the IPCs is still obscure. Does Vaha act as general lipoprotein lipase? LTP mediates transport of Lpp-derived cargo to several tissues including imaginal discs and ovaries. The authors find increased DAG and decreased MAG levels in whole vaha mutant flies. This finding argues in favor of a more general action of Vaha on lipoproteins rather than a local effect at IPCs. In fact, modENCODE

data report on vaha expression in the larval imaginal discs.

6) While most assays were performed with 5-7d old adults, metabolites were measured in adults that are considerably older (30-35d). What was the reason for that? Is this phenotype restricted to this period or present in younger flies as well? To clarify this, the authors should include TAG/Glucose measurements of 5-7d old adults.

7) The statement that vaha is expressed exclusively(?) in enterocytes of the R4 needs more experimental support by co-staining with enterocyte markers. The NP1-Gal4 controlled knockdown, the abundance and the nuclear morphology data are in support that vaha is present in enterocytes but does not exclude its expression in other cell types of the gut. In fact, Fig. 2e seems to indicate that nuclear sizes of Vaha positive gut cells vary.

8) Given the broad vaha expression in the R4 region of the gut compared to the the few cells in the brain, which accumulate Vaha, the relative amount of Vaha detected in gut vs. head is surprising. Ho do the authors exclude that other head tissues except the brain accumulate Vaha? Is there direct evidence for Vaha detection in the hemolymph?

9) Fig. 1a: Should it read NP1Gal4 in the legend?

10) Fig. 3c and d: How is it possible that replicates on regular food show no SEM?

11) ig. 3e needs a quantification.

12) Fig. 4 What are the relative intensities normalized to? For example, in 4e the control mean is around 200%.

13) Fig. 4: Does the genetic rescue of the mutant (as shown in a and b) also restore the secretion of Dilp2 (shown in c) to regular food and HFD?

14) Fig. 4: According to b and c moderate changes in intensity translate into substantial changes in secretion in controls. This appear not to be true in the gut-directed RNAi

situation (e and f). Why?

15) According to Fig. 5c major phospholipid species are downregulated in the vaha mutant (as opposed to many TAG species). What does this mean? Do we look at late metabolic consequences of the developmental delay here?

16) The “Methods” section is insufficient. For example, the description of the fly transgenes deserve revision. All vaha variants (also Vaha V5, Vaha Delta30 and the active site mutants) were cloned into pUAST and are accordingly Gal4-dependent constructs and no knock-ins. Accordingly, UAS vaha V5 etc. is more appropriate. Genomic positions used for cloning need to be precise. Descriptions like “... constructs contained 2kb upstream ...” are not best practice. According to the description the active site mutant changes the three catalytic center amino acids to Ala and was cloned in pUAST, but Supplemental data show that there is an (additional?) Vaha active site GFP fusion construct. Also, “Probes for in situ HCR ... were designed and manufactured by MI” is not sufficient.

To the opinion of this reviewer “data not shown” is not acceptable. There is space in the supplement to present the data.

We greatly appreciate the positive feedback of the reviewers. We thank the three reviewers for their detailed comments and insightful suggestions. The revised manuscript incorporates many new experiments and changes advised by the reviewers. We believe these revisions have strengthened and improved the manuscript. Below, we have provided a point-by-point response to the reviewers' comments.

The reviewers' comments are in black, and our response is in blue. The highlighted versions are the changes made in the text.

Reviewer #1 (Remarks to the Author):

This study shows that the lipase Vaha is secreted from enterocytes in the gut and travels to the insulin producing cells (IPCs) in the brain where it regulates insulin-like peptide 2 (DILP2) secretion. Although sugar is a main factor that promotes insulin secretion, dietary fat also affects insulin release. In here, the authors show that Vaha is a diacylglycerol (DAG) lipase that is released from the gut in response to intake of dietary fat and accumulates in the IPCs to modulate insulin secretion, providing a possible mechanism by which dietary fat can influence secretion of insulin. Conceptually, this is very interesting and I would like to see this work being published!

However, the data do not fully support the conclusions, although the authors provide several lines of evidence. I therefore suggest a major revision to strengthen the conclusions before publication. A key finding is that Vaha is produced by the gut and accumulates in the IPCs, and that this is mediated by LTP-dependent transport. This is based on expression of GFP-tagged Vaha using the NP1-Gal4 and an endogenously tagged Vaha transgene. They need to include the UAS-Vaha::GFP alone without any drivers to exclude that it is not leaky and that there is no expression in the IPCs. Furthermore, they need to show that the endogenously tagged Vaha is not expressed in the brain or IPCs. One thing is the overexpression, but they should also knock Vaha down specifically in the gut and show it eliminates the IPC Vaha signal. It would also be nice to use a second gut driver and a second RNAi lines as well as perform several more experiments with tissue-specific RNAi in addition to the Vaha null mutant. It is also possible to show that gut Vaha accumulates in the IPCs ex vivo by co-culturing guts and CNS.

Our response to each of the reviewer's comments in the above paragraph is provided below.

They need to include the UAS-Vaha::GFP alone without any drivers to exclude that it is not leaky and that there is no expression in the IPCs.

We have now included the UAS-Vaha GFP transgene control without a driver and show that the transgene alone is not leaky (Supplementary Fig.1b). The following sentence is included in the text on page 5:

UAS Vaha GFP without a driver was also tested to ensure that the transgene was not leaky and as seen in Supplementary Fig. 1b, there was no expression in the PI region.

Furthermore, they need to show that the endogenously tagged Vaha is not expressed in the brain or IPCs.

We have performed quantitative PCR (RT-qPCR) for *Vaha V5* (endogenously tagged Vaha) transcript in gut and brain dissected from these flies. We did not detect *Vaha V5* transcript in the brain. The following sentence is included in the text on page 7:

Endogenously tagged *Vaha* (*Vaha V5*) transcript was also not detected in the brain when compared to the gut (Supplementary Fig. 5c).

One thing is the overexpression, but they should also knock Vaha down specifically in the gut and show it eliminates the IPC Vaha signal. It would also be nice to perform several more experiments with tissue-specific RNAi in addition to Vaha null mutant.

As suggested by the reviewer, we have knocked down Vaha specifically in the gut by RNAi and tested for Vaha V5 (driven by its own promoter) signal in the IPCs. For other tissue specific RNAi, we have included Vaha RNAi using neuronal Gal4 driver and fat body Gal4 driver in this experiment. Only gut specific Vaha RNAi reduces Vaha V5 signal in the IPCs. The data from these experiments is shown in main Fig. 2b (showing IPC region) and Supplementary Fig. 6 (showing all the panels including brightfield, DAPI, Vaha V5, ILP2HF and merged images of brains). The quantification of V5 intensity in the IPCs in each of the RNAi experiments is provided in main Figs. 2c (gut knockdown), 2d (neuronal knockdown), and 2e (fat body knockdown). The following paragraph is now included in the main text on page 7:

We also expressed Vaha RNAi using NP1 Gal4 (gut), Elav Gal4 (neurons) and Ppl Gal4 (fat body) drivers and examined Vaha V5 expression in the IPCs. Only gut specific knockdown of Vaha, significantly reduced Vaha V5 signal in the IPCs (Figs. 2b, 2c, 2d, 2e and Supplementary Fig. 6 which shows all the panels including brightfield, DAPI, Vaha V5, ILP2HF, and merged images).

It would also be nice to use a second gut driver.

As a second gut driver, we used *mex1* Gal4 and overexpressed UAS Vaha GFP, dissected brains and stained for Vaha GFP and ILP2HF. In this experiment also, Vaha GFP could be detected in the IPCs. The following line has been included in the text on page 6:

Overexpression of Vaha GFP with a second gut driver, *mex1 Gal4*, also resulted in its detection in the IPCs (Supplementary Fig. 3).

It would also be nice to use a second RNAi line.

As a second RNAi line, we tested the available TriP line (BDSC#56924) for CG8093. We drove expression of this RNAi line using NP1Gal4 driver. However, as shown below for reviewers' perusal, we could not get reduction in *Vaha* transcript level by qPCR and hence could not include a second RNAi in our experiments.

It is also possible to show that gut Vaha accumulates in the IPCs *ex vivo* by co-culturing guts and CNS.

We attempted preliminary co-culture experiments by expressing UAS Vaha GFP using gut Gal4 driver (NP1 Gal4) and collected hemolymph from the flies since Vaha GFP is detected in the hemolymph. We incubated brains dissected from ILP2HF flies with the hemolymph and immunostained the brains for Vaha GFP. However, we were not successful in detecting GFP signal in the IPCs in these experiments. In future, we will attempt standardization of co-culture experiments involving gut, brain and fat body as it would be useful to demonstrate that gut Vaha can be detected in IPCs *ex vivo*.

The Vaha accumulation on the IPCs is dependent on LTP, but how do they explain that LTP is not required for transport of Vaha into the brain, but only its accumulation in the IPCs? In the paper they author mention, it is shown that LTP are required for trafficking across the blood-brain barrier (BBB). Thus, it would be expected that if LTP transports Vaha to the IPCs that is required for Vaha to cross the BBB and enter the brain. (I think they should also show this in a more quantitative manner, by quantifying the amount of Vaha in the IPCs and CNS in animals with fat body-specific LPT knockdown. Generally, there are several experiments where conclusions are based only on single image, which needs to be confirmed by replicates and quantification). LpR1 and LpR2 transport LTP across the BBB 1, and thus, presumably also Vaha. This should be tested by double knockdown of LpR1/2 in the BBB glia. It could also be interesting to see whether LpR1/2 are required for Vaha accumulation in the IPCs. It somehow needs to be confirmed that gut-derived Vaha is secreted into circulation and enter that brain. One of the weaknesses is that

they do not know the mechanism by which Vaha is recruited by the IPCs, or even its uptake into the brain. Perhaps it is mediated by LpR1/2.

We agree with the reviewer. Since Vaha is detected in hemolymph and the head extract will include hemolymph, the Western blot analysis for Vaha showed a band even in LTP knockdown. The blot and the accompanying sentence created ambiguity and hence have been removed. Instead, as suggested by the reviewer, we have performed LRP1 and LRP2 knockdown using a BBB Gal4 driver and examined Vaha V5 staining in the IPCs. In these experiments, staining of Vaha V5 in the IPCs is considerably reduced suggesting LRP1/LRP2 are involved in the transport of Vaha across the BBB and its enrichment in the IPCs.

The intensity of Vaha V5 in the LTP knockdown flies has now been quantified and provided in main Fig. 3c. Since V5 signal was almost absent in many IPCs, the outline of individual IPCs could not be traced and thus quantified. Instead, V5 fluorescence intensity in the IPC region of each brain (area of identical size across genotypes) was quantified using Image J (this method has been used in Redhai et al., (2020) Nature 580:263-268). The following text has been included on pages 8-9.

We performed conditional knockdown of LTP in the fat body using RNAi and the Gal4-Gal80^{ts} system. The efficiency of LTP knockdown was confirmed by quantitative PCR (Supplementary Fig. 9b). We examined Vaha immunostaining in the brain and it was significantly reduced in the IPCs when LTP was knocked down compared to the control IPCs (Figs. 3b and 3c). Two LDL receptor like proteins, LRP1 and LRP2 (Megalin) have been shown to transport LTP across the larval blood brain barrier to control insulin release and signaling in response to yeast lipids⁴⁵. We asked if LRP1 and LRP2 could be involved in the transport of Vaha by knocking down LRP1 and LRP2 in the blood brain barrier glia and examining Vaha immunostaining in the brain. Vaha immunostaining in the IPCs is considerably reduced when LRP1/LRP2 are knocked down in the BBB (Figs. 3d and 3e). Together, these results suggest that the transport of Vaha across the BBB and its enrichment in the IPCs is dependent on LTP and LRP1/LRP2.

A concern is that Vaha is a lipase, and it is difficult to tease out whether the metabolic changes of Vaha deficient animals are a consequence of altered DILP2 signaling or whether the altered DILP2 signaling is an indirect consequence of metabolic defects in Vaha deficient animals. They show that Vaha mutant animals have altered metabolism, but they need to show that Vaha regulates metabolism (e.g. glucose) through insulin. If Vaha regulates metabolism through

DILP2, changing Vaha signaling should have no metabolic effect in a DILP2 mutant. On the other end, activation of IPC activity should rescue metabolic defects of Vaha mutants. I think they should test whether Vaha affects for example glucose tolerance by performing a glucose-tolerance test (GTT). If the hypothesis is correct Vaha is important for GSIS then it should affect glucose tolerance. Of course, this should be dependent on insulin, so loss of Vaha should have no effect on GTT in a DILP2 mutant or animals in which the IPC activity has been silenced. I think it is important to demonstrate that Vaha is indeed affecting glucose metabolism through insulin.

We concur with the reviewer's comment that it is important to demonstrate that Vaha is indeed affecting glucose metabolism through insulin. First, we show that 5-7 day old *vaha* mutant flies have metabolic defects (increase in glucose and triglyceride (TAG) levels) during the fasting/feeding experimental setting and these defects are rescued by gut specific overexpression of UAS Vaha GFP (main Figs. 4d, 4e). As suggested by the reviewer, we tested if stimulation of IPC activity could rescue metabolic defects of *vaha* mutants and whether loss of Vaha had any influence on metabolic defects in DILP mutants. For activation of IPCs, we have used the bacterial sodium channel (NaChBac) under the control of dILP2 Gal4. For silencing of IPCs, we tried expression of inhibitory potassium channel Kir 2.1 in the IPCs using dILP2Gal4 with tubulin-driven temperature sensitive Gal80. Since these experiments had to be performed at 29°C, the metabolism of flies was higher making comparisons of glucose and TAG complicated. Instead, to reduce ILP activity, we have used the triple mutant, *ilp2,3,5/ilp2,3* since *ilp2,3,5/ilp2,3,5* was lethal in our hands. IPC activation rescued metabolic defects as well as the developmental delay observed in *vaha* mutants. On the other hand, the metabolic defects and developmental delay of combined mutants of *vaha* and *ilp2,3,5/ilp2,3* mutants were not different from *ilp2,3,5/ilp2,3* alone. Also, gut specific overexpression of Vaha reduced glucose and TAG levels in wild type flies; however, their levels did not change in *ilp2,3,5/ilp2,3* mutant background. A new main figure showing the data from these experiments (Figs. 6a-6g) and supplementary figures (Figs. 14b-14e) have been added.

As suggested by the reviewer, we performed glucose tolerance test in *vaha* mutants, *insulin* mutants and the combined mutants. Vaha did not influence GTT in the insulin mutant background. This data has been included in main Fig. 6h. A new section describing all the above results is now included in the text on pages 14 and 15:

Metabolic changes and developmental delay in *vaha* mutants result from compromised

insulin secretion

We explored the consequences of modulating IPC activity on the different phenotypes of *vaha* mutant. We asked if activating ILP secretion could rescue *vaha* mutant phenotypes. To achieve this, we depolarized the IPCs by expressing the bacterial sodium channel (NaChBac) under the control of dILP2 Gal4⁵³. Under these conditions, ILP2HF accumulation in the IPCs of *vaha* mutant was considerably reduced (Figs. 6a and 6b). The activation also rescued the

metabolic changes (increased glucose and increased TAG) observed in the *vaha* mutant during fasting/refeeding (Figs. 6c and 6d). We also examined whether activation of the IPCs would rescue the developmental delay in *vaha* mutants and indeed it did so (Fig. 6g). On the other hand, we reduced ILP activity by using *ilp2,3,5 / ilp2,3* mutants which were viable compared to *ilp2,3,5 / ilp2,3,5* mutants that were lethal in our hands⁵⁴. We then made mutants of *vaha* with *ilp2,3,5 / ilp2,3* (combined mutant) and measured glucose and TAG levels in this background. As seen in Figs. 6e, 6f, the metabolite levels in the combined mutant were not different from *ilp2,3,5 / ilp2,3* mutants alone. We also monitored their development from embryos to adults in these mutant backgrounds. As shown in Fig. 6g, the eclosion time for 100% of the combined mutant flies was like that of the *ilp2,3,5 / ilp2,3* mutant. To further test the idea that Vaha affects glucose metabolism through insulin, we performed a glucose tolerance test (GTT). The flies were fasted and transferred to a glucose diet for one hour and then re-fasted for one hour (Fig. 6h). The impairment in glucose clearance upon re-fasting was not significantly different in the combined mutants compared to *ilp2,3,5 / ilp2,3* mutants. This result shows loss of Vaha does not significantly affect GTT in insulin mutant background. Additionally, overexpression of Vaha in the gut reduced glucose and TAG levels in wild type flies during fasting/feeding, however their levels were not changed when Vaha was overexpressed in *ilp2,3,5 / ilp2,3* mutant background (Supplementary Figs. 14b-e). Collectively these results suggest that the developmental delay and metabolic changes in Vaha deficient animals are a consequence of altered insulin release.

Fig. 2d. Higher resolution images are necessary, also with higher magnification.

Original Fig. 2d is now Fig. 2f. Improved live images of gut and brain have been provided. A 3X magnified image of the region of the gut showing staining is included in the middle panel of Fig. 2f.

Fig. 2e. I agree these are likely enterocytes that produce Vaha in the R4, but it should be shown.

We attempted to show that enterocytes produce Vaha by two approaches. In the first approach we used anti Pdm1 (Nubbin) antibody which has been used as a marker for enterocytes. We stained wild type guts or guts dissected from flies expressing Vaha Gal4 > UAS mcherry.nls with anti Pdm1 antibody obtained from two different sources. As shown below, we failed to get enterocyte staining with these antibodies. In the second approach, we used Vnd LexA line based on Vnd Gal4 identified as being expressed in the R4 region of the gut (Lim YS et al., (2021) J Neurogenet. 35: 33-44). However, as shown below we detected diffuse staining in the entire gut. We have provided a high magnification view of a portion of the R4 region from this experiment. Vaha staining (pseudocolored orange) is observed in most cells with large nuclei (enterocytes) and is not seen in cells with small nuclei (other cell types in the gut, indicated by white arrows). The data from our attempts are provide for the perusal of the reviewers. In future, we will attempt to resolve which cell type in the gut secretes Vaha. We have now moved the following sentence from the main text to the discussion section on page 21.

Among the different cell types in the gut, the intestinal stem cells, hormone secreting enteroendocrine cells, digestive and absorptive enterocytes, the latter are the predominant cell type and have large nuclear size. Based on these two criteria, it is likely, the enterocytes of the R4 region of the midgut secrete Vaha.

Immunostaining of guts dissected from flies expressing UAS nuclear localized mCherry under the control of Vaha Gal4 driver with anti Pdm1 antibody (rabbit, a gift from Dr. Cai Yu, Temasek Lifesciences Laboratory, Singapore) and anti DsRed antibody (mouse) or anti Nubbin antibody (Nub2D4, mouse) from DSHB and anti DsRed antibody (rabbit).

Immunostaining of guts dissected from flies expressing UAS nuclear localized Cherry under the control of Vaha Gal4 driver with anti GFP antibody (mouse) and anti DsRed antibody (rabbit).

Why do they not show the LPT knockdown data?

We apologize for the omission. We have now included qPCR data showing LTP knockdown in Supplementary Fig. 9b.

For some figures like Fig. 3c,d all the controls are normalized to 100. How is that possible and it seems to me that you cannot make any statistical analyses based on this way of analyzing the data.

The original Figs. 3c, 3d were quantification of Western blot data. Two of the three replicates were run on the same gel, immunoblotted, and developed at the same time. The quantification values of these bands were close. One replicate was run at a different time. As a result, the developed films showed different densities for signal and background resulting in quantification values that were different from the other two values. So, in the original submission, we took each control as 100% and calculated the sample value. We have now done the calculation incorporating the differences that shows the spread in the control.

The body and head blots were interchanged in the original submission, we have now corrected this mistake and we apologize for the oversight.

Fig. 3d. Vaha levels goes up in head extracts in animals fed high fat. It should be shown that the gut is the source. For example, knock Vaha down in the gut and show that it does not accumulate in the brain on high fat diet.

As suggested by the reviewer, we knocked down Vaha in the gut and tested if we could detect Vaha V5 signal in the IPCs of flies fed high fat diet. RNAi flies on high fat diet show reduced Vaha V5 signal in the IPCs. We have included this data in main Fig. 3j and the quantification of V5 intensity in the IPCs is provided in Fig. 3k. The following lines have been included in the main text on page 10:

Lipase level was also increased in the head extracts of flies fed coconut oil containing food (Fig. 3g and Fig. 3i). These flies also had higher V5 lipase staining in the IPCs compared to flies on regular food (Fig. 3j and Fig. 3k). Furthermore, knockdown of Vaha in the gut by RNAi significantly reduced the accumulation of Vaha V5 in the IPCs upon feeding high fat (Figs. 3j and 3k).

I think generally for all experiments it is cleaner to use dissected brains or even CNS than using head extracts. The head also contains a lot of fat body, so it is possible that Vaha accumulates in the head fat instead of in the brain itself.

We attempted to use CNS (brain and ventral nerve cord) in imaging, qPCR and Western blotting experiments. We were able to successfully use CNS in imaging and qPCR experiments. We were also able to generate clean Western blots with 5-7 guts and 5-7 CNS dissected from NP1 Gal4 **overexpressing** UAS Vaha GFP or UAS Vaha Δ 30GFP (Figs. 1b, 1c, 1d). However, the V5 blots for detection of **endogenous** Vaha in the CNS were not reproducibly clean. In our hands it takes approximately 45 minutes to dissect 5-7 CNS compared to few seconds for heads. We believe during this time there is considerable deterioration of CNS samples. Due to this technical issue, we have used head extracts for Vaha V5 blots.

They should also examine Vaha transcript levels on different diets, both in dissected CNS and guts.

We examined Vaha transcript levels on high fat and high sucrose food. Vaha transcript level is increased on high fat but not on high sucrose food. This data is included in Supplementary Figs. 9c, 9d. In Fig. 2a and Supplementary Figs. 5a, 5b, 5c, 5d we have shown that Vaha is expressed only in the gut, and we could not detect Vaha transcript in CNS, brain, or head samples. Therefore, in this experiment we examined transcript in the bodies of flies fasted overnight and fed regular food or high fat or high sucrose food. The following lines have been added to the main text on page 9:

We examined transcriptional regulation of Vaha by measuring *Vaha* transcript levels under these dietary conditions. *Vaha* transcript level was increased upon feeding high fat while it did not change significantly upon feeding high sucrose. (Supplementary Figs. 9c, 9d).

Fig. 3e. IPC staining should be quantified using several replicates.

Original Fig.3e is new Fig. 3j. The quantification of V5 intensity in the IPCs of regular and high fat food is now shown in Fig. 3k.

The authors state that *Vaha* mutants they generated are no different in size, weight, or lifespan without showing the data. They should show that data. However, it is puzzling why all these parameters which are influenced by DILP2 are not affected by loss of *Vaha*, if *Vaha* regulates DILP2 as the authors claim? Later on, they actually look into whether *Vaha* affects growth, even though they already excluded it?

We have now included the data that show *vaha* mutants are not different in size, weight or lifespan compared to *w¹¹¹⁸* flies in Supplementary Figs. 11a, 11b and 11c. We apologize for not including this data in the original submission. We have rewritten the characterization of mutant phenotypes by first describing the developmental delay in *vaha* mutants and subsequently describing the defect in insulin secretion. Our data shows *Vaha* has an important influence on ILP2 secretion in the transition from fasting to feeding state. We believe this scenario is not an insulin null situation which leads to smaller size, altered weight and lifespan of adult flies. It is possible that there could be compensation from other ILPs such as ILP6 or other regulators of ILP production and release in *vaha* mutants which are currently difficult to comprehend and demonstrate.

They need to quantify data in Supplementary Fig. 3a. It is not possible to conclude that *Vaha* mutation does not affect DILP2 from a single Western blot without replicates. It is also much better to do this by staining the IPCs and quantifying DILP2 in the IPCs.

We apologize for including this experiment in the original submission. This was a mistake in our thinking. Our intent was to test if ILP2 production changes in the IPCs during the 1h of feeding after overnight fasting in the *vaha* mutants compared to control. After giving it some thought, we feel measuring ILP2 transcript levels would be better than documenting protein levels. Western blots for ILP2 from heads will include insulin produced in the IPCs, insulin present in the brain tissue and insulin present in hemolymph. Thus, our experiment fails to truly detect change in insulin production (if any) between control and *vaha* mutants. We have instead replaced original Supplementary Fig.3a with new Supplementary Fig. 11e. The new panel shows qPCR data for ILP2 in heads dissected from control and *vaha* mutant flies fasted overnight and put on regular food or high fat food for 1 h which is the experimental setting for most of our experiments. Under these conditions *ILP2* transcript level is not significantly different between control and *vaha* mutants. The immunofluorescence data shown in Figs. 4a and 4b reflect the true insulin secretory defect in *vaha* since ILP2 accumulation in the *vaha* mutant brain can now be interpreted as peptide retention in the absence of transcriptional changes (Rulifson et al., (2002) *Science* 296: 1118-1120; Geminard et al., (2009) *Cell Metab* 10:199-207). The following text has been included on page 11:

The enrichment of Vaha in the IPCs suggested that it could have an important function in these cells. We first tested if Vaha has a role in *ILP2* gene expression by analyzing its transcript level during the fasting (16h) / refeeding (1h) paradigm. qPCR analysis indicated that *ILP2* levels were not significantly different in the *vaha* mutants compared to control under these conditions (Supplementary Fig. 11e).

Fig. 4a,b. I am also not sure what comparison they make to say the failure to secrete insulin in response to dietary stimulation is worsened in high fat food. Instead of rescuing the phenotype of Vaha mutants by ubiquitous expression using the Tubulin-Gal4 they should be able to rescue with a gut-specific driver.

This sentence was included because the difference in ILP2HF intensity in the IPCs of control and *vaha* mutant on high fat diet vs control and *vaha* mutant on regular diet is about 30% (Supplementary Figs. 12a, 12b). We allude to this wider differential as getting worse on high fat food. As suggested by the reviewer, we also carried out gut specific rescue experiments and this data is shown in main Figs. 4a, 4b and 4c. Gut specific UAS Vaha GFP expression using NP1 Gal4 driver rescues insulin secretion defect in *vaha* mutants. The differential in ILP2HF intensity of control and *vaha* mutant on regular food versus high fat food in the gut specific rescue follows similar trend as before. The following lines have been included on page 11:

The fluorescence intensity of ILP2HF in IPCs in the control and *vaha* mutants under the different conditions is quantified in Fig. 4b. The failure to secrete insulin in response to dietary stimulation was further worsened on high fat food as there was a wider differential (approximately 30%) in ILP2HF intensity in the IPCs of control and *vaha* mutant on high fat diet (80%) versus control and *vaha* mutant on regular food (50%, Fig.4b).

Fig. 4b,c. Why is the control fasted not compared to control refed reg or HFD?

Original Fig. 4b is now Supplementary Fig. 12b. Control fasted data has been compared to control refed reg and HFD as well as all other statistical comparisons have now been included. Since original Fig.4c (ELISA) did not include rescue data, we have replaced this figure with main Fig. 4c which includes rescue data. Here too, all statistical comparisons have now been included.

I am not sure I understand this correctly, but it seems that they show in Supplementary Fig.3a that refeeding after starvation does not change DILP2 in heads, but then they show that refeeding does changes it in Fig, 4b as would be expected of course.

As mentioned earlier, we apologize for including this experiment in the original submission. This was a mistake in our thinking. Our intent was to test if ILP2 production changes in the IPCs during the 1h of feeding after overnight fasting in the *vaha* mutants compared to control. After giving it some thought, we feel measuring ILP2 transcript levels would be better than documenting protein levels. Western blots for ILP2 from heads will include insulin produced in the IPCs, insulin present in the brain tissue and insulin present in hemolymph. Thus, our experiment fails to truly detect change in insulin production (if any) between control and *vaha* mutants. We have instead replaced original Supplementary Fig.3a with new Supplementary Fig. 11e. The new panel shows qPCR data for ILP2 in heads dissected from control and *vaha* mutant flies fasted overnight and put on regular food or high fat food for 1 h which is the experimental setting for most of our experiments. Under these conditions ILP2 transcript level is not significantly different between control and *vaha* mutants. The immunofluorescence data shown in Figs. 4a and 4b reflect the true insulin secretory defect in *vaha* since ILP2 accumulation in the *vaha* mutant brain can now be interpreted as peptide retention in the absence of transcriptional changes (Rulifson et al., (2002) Science 296: 1118-1120; Geminard et al., (2009) Cell Metab 10:199-207). The following text has been included on page 11:

The enrichment of Vaha in the IPCs suggested that it could have an important function in these cells. We first tested if Vaha has a role in *ILP2* gene expression by analyzing its transcript level during the fasting (16h) / refeeding (1h) paradigm. qPCR analysis indicated that *ILP2* levels were not significantly different in the *vaha* mutants compared to control under these conditions (Supplementary Fig. 11e).

They should show Vaha in circulation, since the protein is large enough for hemolymph detection and they have a tagged version. This can be done by Western blots or perhaps even ELISA.

As suggested by the reviewer, we tested if Vaha could be detected in hemolymph collected from Vaha V5 or NP1 Gal4> UAS Vaha GFP flies by Western blotting with V5 or GFP antibody. Vaha is detected in circulation and this data is shown in main Fig. 3a (V5) and Supplementary Fig. 9a (GFP). The following lines have been included in the text on page 8:

We first tested if Vaha is detected in circulation by collecting hemolymph from Vaha V5 flies or flies expressing Vaha GFP via NP1 Gal4 driver and performing Western analyses. Indeed, endogenously tagged Vaha and gut derived Vaha were detected in the hemolymph (Fig. 3a and Supplementary Fig. 9a).

I am not sure how they conclude that high fat diet enhances GSIS? That seems to require actual feeding glucose or injection of glucose and measuring circulating insulin.

We thank the reviewer for this suggestion. We have rewritten the instances (one in introduction and 4 in discussion) where Vaha was directly connected to GSIS. In these instances, we now write that Vaha influences fat amplified insulin release.

Fig. 6a. Like many other places this needs to be quantified. Showing one images is not sufficient to conclude that Vaha lipase activity is essential for DILP2 accumulation.

Original Fig. 6a showed insulin secretion was compromised in a lipase active site mutant. New data expressing the lipase active site mutant in *vaha* using DILP2 Gal4 is shown in main Figs. 5a and 5b. The active site mutant shows defect in insulin secretion and ILP2HF intensity in the IPCs is quantified in Fig. 5b.

Fig. 6h. It should be shown that TAG, DAG and MAG cannot induce DILP2 release in animals with gut-specific Vaha loss.

We performed supplementation of TAG, DAG and MAG in *vaha* mutants and monitored ILP2HF release. While TAG and DAG did not induce ILP2 release in *vaha* mutants, MAG had a partial rescue effect. This data is now included in main Fig. 8h and Supplementary Fig. 15d. The following text is included on page 19:

We next measured ILP2HF release in *vaha* mutants supplemented with these lipids. While TAG and DAG did not cause release of ILP2HF, MAG could partially rescue ILP2HF secretion in the *vaha* mutant (Fig. 8h, Supplementary Fig. 15d). These data suggest that MAG and/or its downstream product(s) likely mediate the effects of Vaha.

Since the study focus on insulin secretion it is important to measure IPCs activity, which can be done using e.g. CaLexA to show that Vaha loss decrease IPCs calcium activity in response to glucose after high fat diet.

As suggested by the reviewer, we measured IPC activity in control and *vaha* mutants on regular food and high fat food using CaLexA system. Vaha mutants indeed show reduced IPC activity compared to control flies as shown in main Figs. 4f and 4g. The following text is included on pages 12 and 13.

To test if IPC activity is compromised in the *vaha* mutant, we used the CaLexA (calcium-dependent nuclear import of LexA) system since IPC activation is accompanied by an increase in intracellular calcium levels. Here, when calcium levels rise in the IPCs, CaLexA enters the

nucleus and binds the LexA operator to activate downstream CD2 GFP reporter expression. This is a membrane reporter that outlines the IPCs, and we have stained with an anti GFP antibody. IPC activity thus measured in the *vaha* mutant is reduced compared to the control on regular food as well as HFD (Fig. 4f and Fig. 4g).

If the hypothesis is correct that Vaha is important for GSIS then it should affect glucose tolerance. This can be tested by performing a glucose-tolerance test (GTT). Of course, this should be dependent on insulin, so loss of Vaha should have no effect on GTT in a DILP2 mutant or animals in which the IPC activity has been silenced. I think it is important to somehow demonstrate that Vaha is affecting glucose metabolism through insulin.

Please see pages 5 and 6 of this document where this comment has been addressed.

Gut Vaha overexpression should lead to changes opposite the loss of function, thus more DILP2 secretion. This need to be tested.

We tested the effect of gut specific Vaha overexpression in insulin secretion and the data is included in Supplementary Figs. 13d, 13e and 13f. Vaha overexpression does lead to more ILP2HF release as monitored by immunostaining of IPCs and more ILP2HF in circulation as monitored by ELISA compared to control. The following text has been included on page 13:

On the other hand, gut specific overexpression of Vaha, resulted in reduced staining of ILP2HF in IPCs and increased levels of circulatory ILP2HF (Supplementary Figs. 13d-f).

They need to put the Vaha mutation over a deficiency or another mutant allele. As a geneticist this is the right way to confirm that the phenotype is origination from the Vaha locus. The specific problem with CRISPR is that all mutant alleles (if the same gRNAs are used) will have the same off-targets. Now I understand that it was backcrossed, but still using a Deficiency is better especially since many of the experiments rely on this mutant.

We tested the effect of *vaha*/deficiency on ILP2HF release by immunostaining. The data is shown in Supplementary Fig. 11f and 11g. The following text is included on page 12:

We also confirmed that the mutant phenotype originated from the Vaha locus by putting the *vaha* mutant over a deficiency and monitoring insulin secretion (Supplementary Figs. 11f, 11g).

It needs to be shown that Vaha-RNAi using NP1-Gal4 reduces Vaha transcript levels in the gut without affecting the CNS (brain and ventral nerve cord).

We performed qPCR for Vaha transcript level in gut and CNS samples dissected from NP1Gal4 > Vaha RNAi flies. This data is shown in Supplementary Fig.5d. The following text is included on page 7:

To further examine if gut is the source of Vaha, we knocked down Vaha specifically in the gut and examined Vaha transcript levels in the gut and CNS (Supplementary Fig. 5d). While there was considerable reduction in transcript in the gut RNAi samples, Vaha transcript was not detected in CNS.

Minor points:

In fig. 1d it is not possible to see unit on scale bar and other figures miss a scale bar.

Scale bars and units have been thickened in all figures including Fig.1d. Scale bars have been added to figures that were missing them.

Reviewer #2 (Remarks to the Author):

The work by Singh et al entitles “A nutrient responsive lipase mediates gut-brain communication to regulate insulin secretion in *Drosophila*” is the characterization of CG8093, that they name “vaha”. This gene had been partially characterized before, in a previous publication (reference 40 in their manuscript), where they reported that UAS-CG8093 RNAi ubiquitous expression (possibly via an actin-Gal4 driver) is lethal, and that midgut specific expression via *esg*-Gal4 leads to viable flies, albeit with increased TAG, and reduced resistance to starvation. In this paper they use a different Gal4 driver, NP1-Gal4, which is an insertion in the myosin *myo31DF*.

Major points:

1) They perform only a rescue experiment using the *ilp2* secretion phenotype after a fasting/refeeding paradigm, and this by means of ubiquitous expression of *vaha*, and do not perform rescue experiments on the three other phenotypes observed in the CRISPR/Cas allele of *vaha*, which is purportedly a null allele. The mutant phenotypes described for the *vaha* null are: reduced weight at end of first instar larvae, developmental delay, metabolites imbalances in aged flies, and reduced response of *ilp* release in response to dietary fat. Only this last get rescued. The authors should try to rescue the growth delay, reduced weight, and study whether some of the metabolic changes are reverted in a rescued fly.

We performed rescue experiments for the developmental delay and metabolic changes (increased glucose and TAG levels) observed in *vaha* mutants. We overexpressed UAS Vaha GFP in the gut using NP1 Gal4 driver in *vaha* mutant background. The smaller larval size, 48 h delay in eclosion of adult flies and increased glucose and TAG levels during fasting and refeeding in *vaha* were all rescued by gut specific overexpression of UAS Vaha GFP. The defect in ILP2 secretion was also rescued by gut specific overexpression of Vaha GFP. These data are included in main

Figs. 4a, 4b, 4c (ILP2 secretion); Fig. 4d (glucose) and Fig. 4e (TAG) and Supplementary Fig. 10c (size) and Supplementary Fig. 10d (delay). The following lines have been included in the text:

The smaller size of mutant larvae and delay in eclosion could be rescued by gut specific overexpression of Vaha using the NP1 Gal4 driver (Supplementary Figs. 10c, 10d). (page 10)

To test if decreased ILP2HF secretion in the *vaha* mutant affects glucose and TAG levels, we measured glucose and TAG levels in control, mutant and rescue flies fed regular food for one hour after overnight starvation. As can be seen in Figs.4d, 4e, *vaha* mutants show elevated glucose and TAG compared to control and these were rescued by gut specific expression of Vaha. (page 12).

We also examined if overexpression of Vaha in the IPCs could rescue insulin secretion and metabolic defects in *vaha* mutants. This data is shown in main Figs. 5a, 5b (insulin secretion) and Fig. 5c (glucose) and Fig. 5d (TAG) and the following text is included on page 13:

We next examined if targeted expression of Vaha in IPCs of *vaha* mutant could correct insulin secretion, glucose and TAG levels. Expression of UAS-Vaha GFP using the dILP2Gal4 driver in the *vaha* mutant rescued the insulin secretion defect as seen in Figs. 5a, 5b and the metabolic defects as seen in Figs. 5c and 5d.

2) They use normally first week adults for most experiments, but use 30-35 days old flies for metabolic studies. Are younger flies not affected? Have the authors done a developmental profile?

One of our reasons for metabolomic profiling was to interrogate if *vaha* mutant flies displayed diabetic features. We reasoned by 30-35 days, metabolic changes associated with diabetes might set in and would be more apparent than one week old flies. Although we have not done a developmental profile, we have carried out glucose and TAG measurements in homogenates from 5-7 day old control and *vaha* mutant flies in our fasting/feeding experimental setting. Young *vaha* mutants also show increase in glucose and TAG levels. This data is shown in Figs. 4d, 4e.

3) Expression is studied using a tagged construct, and mention is made in reference 40 of another work (Zinke I, Schutz CS, Katzenberger JD, Bauer M, Pankratz MJ (2002) Nutrient control of gene expression in *Drosophila*: Microarray analysis of starvation and sugar-dependent response. *EMBO J* 21: 6162–6173.) where in situ hybridizations showed expression in the midgut, and PCR data in this paper. Yet, flybase shows expression in many other tissues, including neurons, which would be in line with the RNAi results that resulted in lethality (reference 40). Is the RNAi data misleading in that it is also targeting other genes? Or, is the null allele obtained not showing all phenotypes? This should be clarified in order to understand and interpret the current loss-of-function allele, and the RNAi data. Can in situ hybridization be performed in most tissues/ stages to clarify the point, as antibodies were not obtained?

Strong, ubiquitous RNAi knockdown of *Vaha* did result in lethality. However, *vaha* null mutants are viable. Genomic sequencing of the *vaha* mutant for CG8093 gene shows precise and complete deletion of the sequence bracketed by the gRNAs indicating this to be a deletion null mutant. qPCR analysis also supports this conclusion. Additionally, the phenotype of tissue specific RNAi knockdown using NP1 Gal4 is like that of *vaha* mutant phenotype. Therefore, we are confident that our *vaha* mutant is a null and the lethal phenotype observed during ubiquitous expression of the RNAi line is most likely due to off target effects (like many of the available long hairpin VDRC lines), as pointed out by the reviewer.

We performed in situ hybridization chain reaction (HCR) for *Vaha* and *Actin* (control) in *w¹¹¹⁸* third instar larvae. We examined larval brain, ventral nerve cord, ventral wall muscles, wing and eye imaginal discs, fat body, salivary gland and larval gut. We could detect staining for *Vaha* only in the larval gut. We have included this data in Supplementary Fig. 8. The following text has been added in pages 7 and 8:

We also performed HCR on different tissues from third instar larvae (Supplementary Fig. 8).

Vaha RNA expression was detected in the larval gut but not in the brain, ventral nerve cord, wing and eye imaginal discs, fat body and salivary gland (Supplementary Fig. 8).

4) What is a good control for PCR experiments: GAPDH is a glucose metabolism enzyme, and so, could be affected in the mutants/constructs? In reference 40 they use rp49 instead. The difference in expression in reference 40 from intestine to brain is five times, and here it is 600 times: the difference should be clarified.

We thank the reviewer for raising this point and we looked at this conundrum in detail during the revision process. We are consistently getting the results we have reported in this manuscript in several experimental settings where qPCR was performed for *Vaha* transcript in the gut, CNS, brain, or head samples from *w¹¹¹⁸* flies (main Fig. 2a and Supplementary Figs. 5a, 5b and 5d). While we do not know the exact reason for under reporting the differential expression of *Vaha*, a probable reason for the 5-fold difference that was reported in reference 40 was not because of using rp49 but rather likely due to background signal in the earlier RNA preparations. Since we show through different approaches that gut is the source of *Vaha*, we have now normalized *Vaha* expression in the gut as 1 unit and performed comparisons to other instances (CNS, brain, head). We believe this is more appropriate representation of the data than before.

5) Why is data not shown in several instances? (e.g., on page 8) Aren't supplementary figures just for that?

We apologize for the omissions. We have now included the data previously not shown in Supplementary Figures. These include LTP knockdown data (Supplementary Fig. 9b), *vaha* mutant fly size (Supplementary Fig. 11a), mutant weight (Supplementary Fig. 11b) and mutant life span (Supplementary Fig. 11c).

6) The authors should acknowledge that experiments carried out using tagged and other expression constructs were done in a wild type background (except the one rescue experiment), that is, that the endogenous *vaha* expression is also present, and that that might bias results, as they might have gain-of-function effects, especially since no quantification of expression is generally done. In reference 4 they use a different Gal4 construct to drive expression (*esg-Gal4*) in the midgut, and here they use the *NP1-Gal4*, which is more widespread in the intestine yet they see only expression in part of the midgut.

We agree with the reviewer's notion expressed in this comment. We have included the following sentence on page 6.

These experiments were done in a wild type background that contained unlabeled endogenous *Vaha*.

We also agree with the reviewer that different gut Gal4 drivers have been used in the experiments. In this manuscript, in addition to *NP1 Gal4*, we have included rescue of insulin secretion defect in *vaha* mutant using *Vaha Gal4* driver expressing *Vaha GFP* in Supplementary Figs. 12c, 12d.

7) Figures. In general, confocal figures should be bettered. The ones right now are very dark, with little detail. The same frame or plane of view image in phase contrast, or Nomark, should be furnished, to make clear where is the signal generated, and the extent of the tissue(s) surveyed in the pictures. In the paper differences in expression (fluorescent signals) are compared, but no quantitation is shown; this should be done, together with the *n* of the experiments, to back the claims in the text. Also, in several figures bars and lettering is often very small and hard to read (e.g., figure 1b). Figures should be revamped to make sure data is legible, figures are clear, adding scale bars where needed, for example. As they stand now, figures are not acceptable.

We have replaced most of the confocal images in the original submission with new panels. We have included bright field images in the replaced as well as new figures to give an idea of the extent of tissue surveyed in the pictures.

Main figures:

Original figs. 1c, 1d replaced with new 1c, 1d, and supplementary fig. 2

Fig. 2a replaced with new 1e, supplementary fig. 4a

Fig. 2d replaced with 2f

Fig. 3e replaced with 3j

Fig.6a replaced with 5a

Additional newly added main figures with brightfield images included are:

Figs. 2b (supplementary fig. 6 includes brightfield) 3d and 6a.

Newly added supplementary figs with brightfield images are:

1b, 3, 6, 8, 15d.

Quantitation data has now been included for images where DILP2HF fluorescent intensity or Vaha V5 fluorescent intensity is compared.

These include main figures:

Fig. 2b (confocal) and quantification in 2c, 2d, 2e

Fig. 3b (confocal) and quantification in 3c; 3d (confocal) and quantification in 3e, 3j (confocal) and quantification in 3k

Fig. 4a (confocal) and quantification in 4b; 4f (confocal) and quantification in 4g

Fig. 5a (confocal) and quantification in 5b

Fig. 6a (confocal) and quantification in 6b

Fig. 8g (quantification) and supplementary fig. 15c (confocal) and 8h (quantification) and supplementary fig. 15d (confocal).

The Supplementary figures include:

Supp fig.12a (confocal) and quantification 12b; supp fig. 12d (confocal) and quantification 12e

Supp fig.13a (confocal) and quantification 13b; Supp fig.13d(confocal) and quantification 13e

In general scale bars and units have been thickened in all figures. Missing scale bars have been added.

8) Accumulation of Vaha in brain cells is only partially coincident with ilp2 expressing IPCs, and some ilp2-positive cells do not accumulate Vaha. The authors mention that non-ilp2, Vaha expressing neurons should be characterized in the future, but no mention is made of ilp2-positive, Vaha negative cells. Is Vaha only accumulated in some of the IPC cells? Why?

We concur with the reviewer that in some instances we have observed ILP2 staining without Vaha co-staining in IPCs, especially in experiments involving Vaha GFP driven by NP1-Gal4 driver. Endogenous Vaha V5 shows much more consistent colocalization with ILP2. At this time, we are not sure why this is so. Perhaps this signifies some metabolic state that requires further investigation or latency/barrier for Vaha enrichment in these cells.

9) The sentence “We reason, yeast, the major supplier of steryl esters and triglycerides, and cornmeal, a minor contributor, could be the lipid sources that modulate Vaha in regular food.” needs backing.

We based our writing on the following information that 100 gm of corn meal contains 1.7 gm of fat, and 100 gm of yeast contains 1.7 gm of fat. However, careful delineation of all regulatory controls governing Vaha expression requires de-lipidation experiments and gene expression

analysis under different nutritional conditions. Since we could not experimentally support this conjecture, we have now removed this sentence.

10) Dilps should be changed to ilps throughout the text, since ilp is the official name of these genes in Flybase.

We have made the change from Dilp to ilp throughout the text.

Other points:

1) Perhaps add references in supplementary table 3.

We apologize for the omission, and we have now included references in supplementary table 3.

2) A revision of the text for typos and grammar. In general, the text is fine, but still some typos and error have not been corrected (e.g., We next examined, which processes could be on page 11, without the comma; use of the correct hyphen throughout the text).

We have revised the text for typos and grammar.

3) “Of these 280 w1118 pupae developed to adults and 303 vaha pupae developed to adults.”, from page 29: quantitation can be included.

The numbers have been added to the text on page 10.

4) IPC are usually capitalized.

IPC has been capitalized throughout the text.

5) Is “overnight” 14-16 hours?

Overnight fasting has usually been 16 hours in our experiments, and this is now indicated in the Methods section.

6) Shouldn't a supplementary figure showing the full extent of the gel be furnished for all Western experiments?

The full extent of Western blots (uncropped versions) has been now included in the source file accompanying the manuscript.

7) The depository for the metabolomics data should be mentioned in the text.

The following text is included in the Data Availability section on page 35:

Metabolomics data have been deposited to the EMBL-EBI MetaboLights database (DOI: 10.1093/nar/gkz1019, PMID:31691833) with the identifier MTBLS8441.

Reviewer #3 (Remarks to the Author):

Singh and colleagues present a very original study on a new gut-expressed and dietary-regulated lipase called Vaha, which relocates to the insulin-like peptide (Dilp) producing cells in the *Drosophila* brain (IPCs) to exert systemic developmental and metabolic effects by controlling Dilp secretion.

The authors use Vaha fusion transgenes and the heterologous Gal4/UAS system to (i) assess the localization of Vaha in the IPCs and (ii) the expression of the protein in the R4 region of the gut but not in IPCs. Global and tissue-specific mRNA expression data in the gut are provided in support of the findings based on transgene data. The authors demonstrate that Vaha localization at the level of protein secretion depends on the predicted Vaha signal sequence and on a functional LTP (provided by the fat body) at the level of protein uptake into the IPCs. The authors show the Vaha protein levels increase in response to high fat diet, which causes Vaha accumulation in the IPCs. Dilp secretion from the IPCs is impaired in global Vaha mutants and upon knockdown of the gene in the gut. Vaha mutants show developmental delay and – after aging - changes in the metabolite profile including increased total body glucose, DAG and TAG levels, while MAG levels are decreased. Recombinant Vaha has lipase activity accepting DAG, MAG and TAG as substrates. Overexpression of an Vaha lipase active site mutant in the gut does not impair IPC targeting but increases Dilp2 accumulation in the IPCs. Finally, the authors demonstrate that dietary supplementation with TAG, DAG and MAG reduced Dilp2 intensity in the IPCs suggesting increased secretion.

This is a very interesting study, which, however, deserves additional experimental support for the fascinating conclusions drawn and the regulatory concept deduced by the authors (see below). To this reviewer the order and prioritization of the presented data is not intuitive and readability would improve by restructuring. For example, recombinant protein data which prove that Vaha is a bona fide lipase and describe the substrate spectrum are presented in the last figure and not in the first along with the expression data. To my perception the authors provide no evidence that Vaha does not serve additional function(s) in the gut as it is expressed in regions and cells involved in lipid metabolism. So why not proceeding from the biochemical characterization to the expression before introducing the exciting finding that the lipase relocates to the IPCs? With regards to priority: Fig. 1 presents the protein localization data, which are also presented in Fig.2 using a different transgene construct. Why this redundancy?

We decided to tell the story as it unfolded in the lab. Hence the narrative first focuses on Vaha localizing to the IPCs, followed by characterization of *vaha* mutant phenotype and ending with biochemical characterization of substrate specificity of Vaha and that TAG, DAG and MAG facilitate ILP2 release in wild type flies while this event is compromised in *vaha* mutant.

We agree with the reviewer that Vaha could have additional functions in the gut, and we hope to delineate them in future experiments.

We have redone original Fig. 2a (now using CNS) and now included it in Fig. 1e. Since Fig. 1c shows overexpression data, we thought it would be useful to include endogenous Vaha V5 localization to IPCs. Although Vaha V5 was cloned in pUAST vector, the endogenous promoter drives the gene without the need for Gal4.

Points of concern:

1) According to Fig. 3, Vaha abundance is responsive to diet. Is this due to transcriptional regulation in the gut? According to the model of the authors, Vaha differential accumulation in the IPCs impact on Dilp secretion. This argues in favor of a vaha gain-of-function phenotype upon Vaha overexpression in the gut (on regular food). Do the authors observe such a phenotype and if so, is this phenotype absent in the *dilp2,3,5* mutant? Moreover, if the IPC-expressed Dilps mediate the developmental defects of vaha mutants, is the vaha *dilp2,3,5* double mutant phenotypically identical to the *dilp2,3,5* single mutant? This needs to be tested.

We examined *Vaha* transcript levels in flies fasted overnight and fed regular food or high fat or high sucrose food. *Vaha* transcript level is increased on high fat but not on high sucrose food. The following lines have been added to the main text on page 9:

We examined transcriptional regulation of Vaha by measuring *Vaha* transcript levels under these dietary conditions. *Vaha* transcript level was increased upon feeding high fat while it did not change significantly upon feeding high sucrose. (Supplementary Figs. 9c, 9d).

As suggested by the reviewer, we tested if Vaha overexpression in the gut influences ILP secretion in regular food and this data is included in Supplementary Figs. 13d, 13e and 13f. Vaha overexpression does lead to more ILP2HF release as monitored by immunostaining of IPCs and more ILP2HF in circulation as monitored by ELISA compared to control. The following text has been included on page 13:

On the other hand, gut specific overexpression of Vaha, resulted in reduced staining of ILP2HF in IPCs and increased levels of circulatory ILP2HF (Supplementary Figs. 13d-f).

Since we cannot follow insulin secretion in an insulin mutant, we evaluated other phenotypes upon Vaha expression in an insulin mutant. Gut specific overexpression of Vaha reduced glucose and TAG levels in wild type flies; however, their levels did not change in *ilp2,3,5/ilp2,3* mutant background. This data is included in supplementary figures (Figs. 14d and 14e).

As suggested by the reviewer, we also tested if ILPs mediate the developmental delay observed in *vaha* mutants by making *vaha* and *ilp2,3,5/2,3* combined mutants. Data shown in Fig. 6g suggest that the phenotype of the combined mutant is not different from the *ilp2,3,5/2,3* mutant. On the other hand, constitutive activation of IPCs in a *vaha* mutant background rescues the developmental delay. The following new section is now included on pages 14 and 15:

Metabolic changes and developmental delay in *vaha* mutants result from compromised insulin secretion

We explored the consequences of modulating IPC activity on the different phenotypes of *vaha* mutant. We asked if activating ILP secretion could rescue *vaha* mutant phenotypes. To achieve this, we depolarized the IPCs by expressing the bacterial sodium channel (NaChBac) under the control of dILP2 Gal4⁵³. Under these conditions, ILP2HF accumulation in the IPCs of *vaha* mutant was considerably reduced (Figs. 6a and 6b). The activation also rescued the metabolic changes (increased glucose and increased TAG) observed in the *vaha* mutant during fasting/refeeding (Figs. 6c and 6d). We also examined whether activation of the IPCs would rescue the developmental delay in *vaha* mutants and indeed it did so (Fig. 6g). On the other hand, we reduced ILP activity by using *ilp2,3,5 / ilp2,3* mutants which were viable compared to *ilp2,3,5 / ilp2,3,5* mutants that were lethal in our hands⁵⁴. We then made mutants of *vaha* with *ilp2,3,5 / ilp2,3* (combined mutant) and measured glucose and TAG levels in this background. As seen in Figs. 6e, 6f, the metabolite levels in the combined mutant were not different from *ilp2,3,5 / ilp2,3* mutants alone. We also monitored the development from embryos to adults in these mutant backgrounds. As shown in Fig. 6g, the eclosion time for 100% of the combined mutant flies was like that of the *ilp2,3,5 / ilp2,3* mutant.

2) According to the model Vaha exerts systemic metabolic effects by lipase function in the IPCs, while the expression of the gene in the gut might be required for its responsiveness to diet. The authors should take advantage of the wonderful tools they developed for an extended mechanistic analysis in the *vaha* mutant background. What are the phenotypes of flies rescued (or not) by targeted expression in the IPCs of Vaha, of the Vaha signal sequence mutant and of the Vaha active site mutant expression?

As suggested by the reviewer, we expressed the Vaha signal sequence deleted transgenic and Vaha active site mutant in the IPCs of *vaha* mutant using DILP2Gal4 driver. We monitored whether the insulin secretion defect and metabolic defects (increase in glucose and TAG) in *vaha* mutants are rescued by these transgenics. While the signal peptide deleted transgenic rescued

these defects, the active site mutant did not. A new main figure (Fig. 5) has been added and the following text is included on pages 13 and 14:

We next examined if targeted expression of Vaha in IPCs of *vaha* mutant could correct insulin secretion, glucose and TAG levels. Expression of UAS-Vaha GFP using the dILP2Gal4 driver in the *vaha* mutant rescued the insulin secretion defect as seen in Figs. 5a, 5b and the metabolic defects as seen in Figs. 5c and 5d. Since Vaha is a secreted protein, we addressed the local requirement of Vaha by expressing the signal peptide deleted transgenic (UAS Vaha Δ 30 GFP) in a similar manner. This also corrected the insulin secretion and metabolic defects suggesting that the presence of Vaha within IPCs is sufficient for this function. We next tested if lipase activity is important for the function of Vaha in the IPCs. Classical lipase function depends on a catalytic triad (Ser-His-Asp) mechanism with the Ser located in the middle of the GX SXG motif found in lipases that adopt the α/β hydrolase fold⁵². The Ser-His-Asp catalytic triad is also present in Vaha and we generated transgenic flies wherein each of these three active site residues was replaced by Ala. When this construct was expressed in the IPCs, ILP2HF levels were considerably higher compared to control suggesting lipase activity is essential for ILP2 release from IPCs (Figs. 5a, 5b). The increase in glucose and TAG levels in the *vaha* mutant were not corrected by expression of the active site mutant (Figs. 5c, 5d).

3) The rescue experiment provided by the authors uses a ubiquitous driver and accordingly is not appropriate to support the model that Vaha is expressed in the gut but acts in the IPCs. How does the fusion protein distribution/accumulation in the IPCs look like under these circumstances? A rescue experiment targeting *vaha* to its endogenous expression domain(s) using the Vaha Gal4 driver or to the gut using the NP1 driver is required.

As suggested by the reviewer, we carried out gut specific rescue experiments and this data is shown in main Figs. 4a, 4b and 4c. Gut specific UAS Vaha GFP expression using NP1 Gal4 driver rescued insulin secretion defect in *vaha* mutants. (page 11).

We also attempted rescue experiments using Vaha Gal4 driver expressing UAS Vaha GFP. This also corrected the insulin secretion defect in *vaha* mutant, and the data is included in Supplementary Figs. 12c, 12d.

The original rescue experiment involving ubiquitous expression of Vaha is shown in Supplementary Figs. 12a, 12b.

4) If Vaha lipase activity in the IPCs mediates the effect of dietary lipid supplementation (TAG, DAG, MAG) on Dilp release, are the effects shown in Fig. 6h absent in the *vaha* mutant?

As suggested by the reviewer, we fed *vaha* mutant flies with TAG, DAG or MAG and monitored DILP2HF release compared to control flies. While TAG and DAG could not elicit insulin release in *vaha* mutant, MAG had a partial rescue effect in the mutant flies. This data is now included in main Fig. 8h and Supplementary Fig. 15d. The following text is included on page 19:

We next measured ILP2HF release in *vaha* mutants supplemented with these lipids. While TAG and DAG did not cause release of ILP2HF, MAG could partially rescue ILP2HF secretion in the *vaha* mutant (Fig. 8h, Supplementary Fig. 15d). These data suggest that MAG and/or its downstream product(s) likely mediate the effects of Vaha.

5) The mechanism of Vaha lipase function in the control of dilp release in the IPCs is still obscure. Does Vaha act as general lipoprotein lipase? LTP mediates transport of Lpp-derived cargo to several tissues including imaginal discs and ovaries. The authors find increased DAG and decreased MAG levels in whole *vaha* mutant flies. This finding argues in favor of a more general action of Vaha on lipoproteins rather than a local effect at IPCs. In fact, modENCODE data report on *vaha* expression in the larval imaginal discs.

We agree with the reviewer that Vaha lipase function in the control of ILP release is still not clear and Vaha could function as a lipoprotein lipase. New data from experiments based on the reviewer's earlier suggestion where Vaha, or signal peptide deleted Vaha or active site mutant Vaha is expressed in the IPCs of *vaha* mutant supports the conclusion that locally active Vaha in IPCs is sufficient to rescue the observed phenotypes of the mutant (main Fig. 5).

qPCR experiments in the gut, CNS, brain, or head suggest Vaha transcript could not be detected in tissues other than gut (main Fig. 2a, Supplementary Fig. 5). Also, HCR experiments in wild type third instar larvae show *Vaha* staining in the gut but not in several other tissues including the imaginal discs (Supplementary Fig. 8).

6) While most assays were performed with 5-7d old adults, metabolites were measured in adults that are considerably older (30-35d). What was the reason for that? Is this phenotype restricted to this period or present in younger flies as well? To clarify this, the authors should include TAG/Glucose measurements of 5-7d old adults.

One of our reasons for metabolomic profiling was to interrogate if *vaha* mutant flies displayed diabetic features. We reasoned by 30-35 days, metabolic changes associated with diabetes might set in and would be more apparent than one week old flies. As suggested by the reviewer, we have carried out glucose and TAG measurements in homogenates from 5-7 day old control and

vaha mutant flies in our fasting/feeding experimental setting. Young *vaha* mutants also show increase in glucose and TAG levels. This data is shown in Figs. 4d, 4e and the following text is included in page 12:

To test if decreased ILP2HF secretion in the *vaha* mutant affects glucose and TAG levels, we measured glucose and TAG levels in homogenates from control, mutant and rescue flies fed regular food for one hour after overnight starvation. As can be seen in Figs.4d, 4e, *vaha* mutants show elevated glucose and TAG compared to control and these were rescued by gut specific expression of Vaha.

7) The statement that *vaha* is expressed exclusively(?) in enterocytes of the R4 needs more experimental support by co-staining with enterocyte markers. The NP1-Gal4 controlled knockdown, the abundance and the nuclear morphology data are in support that *vaha* is present in enterocytes but does not exclude its expression in other cell types of the gut. In fact, Fig. 2e seems to indicate that nuclear sizes of Vaha positive gut cells vary.

We attempted to show that enterocytes produce Vaha by two approaches. In the first approach we used anti Pdm1 (Nubbin) antibody which has been used as a marker for enterocytes. We stained wild type guts or guts dissected from flies expressing Vaha Gal4 > UAS mcherry.nls with anti Pdm1 antibody obtained from two different sources. As shown below, we failed to get enterocyte staining with these antibodies. In the second approach, we used Vnd LexA line based on Vnd Gal4 identified as being expressed in the R4 region of the gut (Lim YS et al., (2021) J Neurogenet. 35: 33-44). However, as shown below we detected diffuse staining in the entire gut. We have provided a high magnification view of a portion of the R4 region from this experiment. Vaha staining (pseudocolored orange) is observed in most cells with large nuclei (enterocytes) and is not seen in cells with small nuclei (other cell types in the gut, indicated by white arrows). The data from our attempts are provide for the perusal of the reviewers. In future, we will attempt to resolve which cell type in the gut secretes Vaha.

We have now moved the following sentence from the main text to the discussion section on page 21.

Among the different cell types in the gut, the intestinal stem cells, hormone secreting enteroendocrine cells, digestive and absorptive enterocytes, the latter are the predominant cell type and have large nuclear size. Based on these two criteria, it is likely, the enterocytes of the R4 region of the midgut secrete Vaha.

Immunostaining of guts dissected from flies expressing UAS nuclear localized mCherry under the control of Vaha Gal4 driver with anti Pdm1 antibody (rabbit, a gift from Dr. Cai Yu, Temasek Lifesciences Laboratory, Singapore) and anti DsRed antibody (mouse) or anti Nubbin antibody (Nub2D4, mouse) from DSHB and anti DsRed antibody (rabbit).

Immunostaining of guts dissected from flies expressing UAS nuclear localized Cherry under the control of Vaha Gal4 driver with anti GFP antibody (mouse) and anti DsRed antibody (rabbit).

8) Given the broad vaha expression in the R4 region of the gut compared to the the few cells in the brain, which accumulate Vaha, the relative amount of Vaha detected in gut vs. head is

surprising. How do the authors exclude that other head tissues except the brain accumulate Vaha? Is there direct evidence for Vaha detection in the hemolymph?

We tested if Vaha could be detected in hemolymph collected from Vaha V5 or NP1 Gal4> UAS Vaha GFP flies by Western blotting with V5 or GFP antibody. Vaha is detected in circulation and this data is shown in main Fig. 3a (V5) and Supplementary Fig. 9a (GFP). At this time, in the absence of a detailed mechanism for entry and enrichment of Vaha into IPCs it will be difficult to conjecture why other tissues do not accumulate Vaha. The following lines have been included in the text on page 8:

We first tested if Vaha is detected in circulation by collecting hemolymph from Vaha V5 flies or flies expressing Vaha GFP via NP1 Gal4 driver and performing Western analyses. Indeed, endogenously tagged Vaha and gut derived Vaha were detected in the hemolymph (Fig. 3a and Supplementary Fig. 9a).

9) Fig. 1a: Should it read NP1Gal4 in the legend?

Yes, and it has been corrected.

10) Fig. 3c and d: How is it possible that replicates on regular food show no SEM?

The original Figs. 3c, 3d were quantification of Western blot data. Two of the three replicates were run on the same gel, immunoblotted, and developed at the same time. The quantification values of these bands were close. One replicate was run at a different time. As a result, the developed films showed different densities for signal and background resulting in quantification values that were different from the other two values. So, in the original submission, we took each control as 100% and calculated the sample value. We have now done the calculation incorporating the differences that shows the spread in the control. The body and head blots were interchanged in the original submission, we have now corrected this mistake and we apologize for the oversight.

11) Fig. 3e needs a quantification.

The original Fig. 3e is now new Fig. 3j and the quantification is provided in Fig. 3k.

12) Fig. 4 What are the relative intensities normalized to? For example, in 4e the control mean is around 200%.

We thank the reviewer for bringing this to our notice. In our previous submission in several experiments an arbitrary value was set at 100% against which others were compared. Although in these measurements the comparisons are correct it lacks uniformity. We have now normalized the mean values of the control to hundred for all comparisons. This uniform representation makes for a better reading as implied by the reviewer.

13) Fig. 4: Does the genetic rescue of the mutant (as shown in a and b) also restore the secretion of Dilp2 (shown in c) to regular food and HFD?

The ubiquitous rescue has been replaced by gut specific rescue. Gut specific expression of UAS Vaha GFP did indeed rescue secretion defect in *vaha* mutant in regular and high fat food (Figs. 4a, 4b, 4c).

14) Fig. 4: According to b and c moderate changes in intensity translate into substantial changes in secretion in controls. This appear not to be true in the gut-directed RNAi situation (e and f). Why?

We do not know the exact reason for these differences. We do not know if variations in dynamics of insulin release in different genetic backgrounds could contribute to these differences.

15) According to Fig. 5c major phospholipid species are downregulated in the *vaha* mutant (as opposed to many TAG species). What does this mean? Do we look at late metabolic consequences of the developmental delay here?

As the reviewer suggested the downregulated phospholipid species could arise as a secondary consequence of insulin deregulation in *vaha* mutants which needs further investigation.

16) The “Methods” section is insufficient. For example, the description of the fly transgenes deserve revision. All *vaha* variants (also Vaha V5, Vaha Delta30 and the active site mutants) were cloned into pUAST and are accordingly Gal4-dependent constructs and no knock-ins. Accordingly, UAS *vaha* V5 etc. is more appropriate. Genomic positions used for cloning need to be precise. Descriptions like “... constructs contained 2kb upstream ...” are not best practice. According to the description the active site mutant changes the three catalytic center amino acids to Ala and was cloned in pUAST, but Supplemental data show that there is an (additional?) Vaha active site GFP fusion construct. Also, “Probes for in situ HCR ... were designed and manufactured by MI” is not sufficient. To the opinion of this reviewer “data not shown” is not acceptable. There is space in the supplement to present the data.

The Methods section describing the generation of different Vaha transgenics has been rewritten to include sufficient details. (pages 23, 24).

Although Vaha V5 and Vaha Δ 30 V5 were cloned in pUAST vector, the endogenous promoter drives the gene without the need for Gal4. This is now indicated in the text on page 24.

We apologize for the confusion about the active site mutant construct. There is only one construct which is a GFP fusion protein. This is now indicated in the text on page 24.

Generation of UAS Vaha GFP, UAS Vaha Δ 30 GFP and UAS-Vaha active site mutant GFP transgenic flies

A fusion protein of C-terminally EGFP tagged Vaha was designed. For this the open reading frame of CG8093 was fused in frame with cDNA sequence of EGFP (GenBank: AAB02574.1) with a 13 aa linker sequence coding for AVDGTAGPGSIAT between the two sequences. The cDNA coding for the fusion protein was cloned into pUAST using Not1-Xho1 site (synthesized and cloned by GenScript). Similarly, C-terminally EGFP tagged Vaha with sequences for the first 30 amino acids deleted from the N-terminus was synthesized and cloned into pUAST as a Not1-Xho1 fragment by GenScript. Vaha active site mutant construct tagged with C-terminal EGFP was generated wherein active site residues Serine (167), Aspartic acid (341) and Histidine (374) were replaced by Alanine and cloned as Not1-EcoR1 fragment into pUAST vector (synthesized by GenScript). The constructs were injected into *w¹¹¹⁸* embryos to generate transgenic flies (BestGene Inc).

Generation of genomic Vaha V5 and genomic Vaha Δ 30 V5 transgenic flies

1739 bp genomic fragment (containing protein-coding sequence of Vaha together with 0.5kb upstream regulatory region) was amplified by single fly PCR and cloned as EcoR1-Kpn1 fragment into pUAST vector (the upstream oligo 5'-GCG AAT TCT AAT TAT AGT AAT CAC TTT AAA AT - 3' maps 473 bp upstream of the start of CG8093 with an EcoR1 site; and the downstream oligo 5'-AAA ATA GGT ACC TTA CGT AGA ATC GAG ACC GAG GAG AGG GTT AGG GAT AGG CTT ACC TGC ATT ATT TAT GTC GTT AAT TAC C -3' has the sequence for V5 inserted just before the stop codon). The clone was confirmed by sequencing.

The Vaha Δ 30 V5 having the upstream sequence and V5 epitope tag as above but lacking the coding sequence for the first 30 amino acids of CG8093 was synthesized and cloned as an EcoR1-Not1 fragment into pUAST by GenScript. Transgenic flies were generated by BestGene Inc. Although the above two clones were generated in pUAST vectors, the endogenous promoter drives the expression of these genes without the need for Gal4 driver.

Generation of Vaha Gal4 transgenic flies

The Vaha-Gal4 construct includes DNA 2kb upstream of Vaha start codon (the sequence including 2kb upstream of the Start and 2 kb downstream of the Stop codons respectively is listed as 'extended gene sequence' of CG8093 in Flybase) that is fused in frame to the ATG of the Gal4 sequence (Gen Bank: K01486.1) and cloned as a Not1-EcoR1 fragment into pUAST (synthesized by GenScript). Transgenic flies were generated by BestGene Inc.

The *in situ* HCR methodology section has now been expanded (page 32).

In situ hybridization chain reaction (HCR)

HCRs were performed following the instructions provided by Molecular Instruments (MI) and based on published protocols⁷⁶. Probes for *in situ* HCR for *Vaha* (CG8093), *Actin* (CG7438) and the hairpins were designed and manufactured by MI. The probe set size was 20 for each gene and the amplifiers were B2 (Alexa fluor 647) for *Vaha* and B3 (Alexa fluor 546) for *Actin*. The transcripts for the two genes were used for designing the probes. While the company does not provide the individual probe binding sequences, the lot numbers provided by the company for the probes are as follows: CG7438 (*Actin*) – PRG995 and for CG8093 (*Vaha*) – PRG996 and will

aid in design and reproduction of the probes used in the study. The probes are designed to maximize target specificity while minimizing off-targeting complementarity.

We apologize for the data not shown. We have now included the data previously not shown in Supplementary Figures. These include LTP knockdown data (Supplementary Fig. 9b), *vaha* mutant fly size (Supplementary Fig. 11a), mutant weight (Supplementary Fig. 11b) and mutant life span (Supplementary Fig.11c).

REVIEWER COMMENTS

Reviewer #1 (Remarks to the Author):

The authors have invested significant effort in revising the manuscript, which has strengthened conclusions and an overall improved the study. I recommend publication of this work and have only a few minor comments below.

Fig. 3h: With only three data points and substantial overlap, it is surprising that they are statistically different?

In fig. 4g and 4f, IPC activity is shown using the CaLexA system. However, the images and stains appear weak and unconvincing. It would be unfortunate to publish this otherwise nice study with such poor-quality images. Even in the control group, it is difficult to discern IPC stains. Normally, CaLexA provides clear IPC stains. I think they should include better stains at least, or even better to perform both GFP (calcium activity) and DILP2 stains, allowing for clear visualization of IPCs in both genotypes.

For the sake of improved presentation, I recommend removing empty spaces in figures, such as Fig. 7, and aligning all elements neatly. Many images appear misaligned.

Reviewer #2 (Remarks to the Author):

The revised version of the manuscript is much improved, and critically, several key experiments that were before not quantitated and furnished have now been addressed, all of which strengthens the paper and its basic tenet: that the vaha gene is expressed in the gut, transported to the brain, and exerts its effect in the IPCs by metabolizing DAGs, most probably. Yet, I still find issue with some of the pictures provided. In figure one, besides the IPCs, there are other instances in the brain where signal is detected: Nothing is said about it, and it should be acknowledged, irrespective of whether this is the focus of the paper or not. It could be of potential interest whether other aspects of the vaha gene function are centered on these other cells (not explored here), which might be the groundwork for future work. Also, a close-up of the brain areas where the expression is found should also be

provided. Also, the double staining with DAPI and the merge images should be bettered, to actually show both stains and see the nuclei and cytoplasm; in this case, a higher magnification view should be provided. Same commentary goes for supplementary figures 1-4 and 6, where DAPI signal cannot be seen, and the pictures without having higher magnifications, are very hard to read. Better pictures should be provided. In general, all pictures of the supplementary figures have the same problems: they are very small, low magnification such that many time the relevant details cannot be discerned, and there is no real need to do this: higher magnifications should be provided, as well as merged images. Supplementary figure 7 shows sexual dimorphism in intestinal vaha expression, and that is not commented upon, and should be, since that could potentially be interesting, given the differences in metabolism between males and females. In general, the DAPI staining is not seen very well, and hard to define.

In figure 2 it is hard, if at all clear, to distinguish the differences in intestinal nuclei, and so, the argument about the co-localization is not shown clearly. Better pictures, and close-ups at higher magnification should be provided. Clearer pictures for figure 4f should also be provided.

A re-working of the figures is necessary, to bring them up to standards; other than that I have no comments on the paper.

Reviewer #3 (Remarks to the Author):

The authors presented a substantially improved and revised version of the manuscript, which appropriately addressed all of my concerns.

This is a conceptually very interesting study which will hopefully find a broad readership. Clarifying how IPC Vaha function translates to DILP-release control is a fascinating future question.

Reviewer #4 (Remarks to the Author):

The central finding of the manuscript presented by Singh et al. is the identification and characterization of a lipase that gets secreted by enterocytes in response to dietary cues to remotely control insulin secretion at insulin-producing cells in the brain. Although the exact

mechanism behind this observation remains unknown, this observation is an important advance in our understanding of how different organs communicate to process nutrient signals.

In the revised version of this manuscript, the authors address numerous issues that arose during the review process. In summary, these experiments allowed the authors to confirm and strengthen their original conclusions.

All specific major issues from my initial revision have been answered except for one, which the authors perceive as beyond the scope of the study. Therefore, I have no major concerns. Nevertheless, in my opinion there are still some clarifications needed in the methods section before publication.

First, the authors should describe how total DAG and MAG levels (presented in Figure 8) were calculated. If those values represent the sum of individual species detected in the LC/MS approach, this should be clarified in the method section. Since individual MAG and DAG species vary considerably according to the heat map in Figure 7 the authors should consider to add another supplementary figure showing the comprehensive panel of MAG and DAG species for each genotype.

Second, the description of the enzyme activity assay is somehow confusing. The authors state that 100 μ M cold substrate was used as a basis in every assay with different concentrations (25-500 nM) of 3H-substrate added on top of that. What were the actual concentrations then? Just increasing the amount of tracer in the assay would not substantially change the overall concentration of the substrate. Please clarify. Moreover, it would be helpful to have more details about substrate preparation. Since those lipids (except MAG) are essentially insoluble, it remains elusive how they were emulsified in the buffer to have a homogenous substrate.

RESPONSE TO REVIEWERS' COMMENTS

We appreciate that all the reviewers found our revised manuscript substantially strengthened due to the significant effort invested in improving several experiments based on their suggestions. There were suggestions and comments from the reviewers on the revised version. We have now addressed these comments and provided a point-by-point response below. We are thankful and value the advice we have received and recognize that performing these experiments have allowed us to consolidate and strengthen our conclusions.

SUMMARY OF MAJOR CHANGES

We have provided improved images in Main Fig. 4f and quantitation in Fig. 4g (comment from reviewers 1 and 2). This figure deals with measuring the activity of the insulin producing cells in control and *vaha* mutant on regular and high fat food using the CaLexA system. We had to bring in copies of the CD8 GFP reporter transgene to improve the sensitivity of the CaLexA system. Since the reporter transgene and the *vaha* mutant are on the same chromosome, this necessitated meiotic recombination and took us some time.

We have also included many new and have improved existing Supplementary Figures (comment from reviewer 2) by providing high magnification images and high resolution images. Since we received comments in the original submission to include images of the brain and ventral nerve cord and during re-revision to include images of the IPC region, we have tried to accommodate both suggestions. Wherever possible, the main figure shows the brain and ventral nerve cord, or brain images and a supplementary figure shows a high magnification view of the IPC region of the same brain or vice versa.

Main Fig. 1a (gut + brain) and Supplementary Fig. 1c (additional brain high mag)

Main Fig. 1c (brain + VNC) and Supplementary Fig. 2a (IPC)

Main Fig. 1d (brain + VNC) and Supplementary Fig. 2b (IPC)

Main Fig. 1e (brain + VNC) and Supplementary Fig. 4a (IPC)

Main Fig. 2b (IPC) and Supplementary Figs. 6a and 6b (brain)

Main Fig. 4f (IPC) and Supplementary Figs. 13a and 13b (brain)

Supplementary Figs. 3a (brain) and 3b (IPC)

Additionally, we have also split Supplementary Figs 6 and 8 into two parts (a and b) to accommodate bigger images.

We have removed white space and aligned all figures as suggested by reviewer 1.

We have performed kinetic assays (Figs.8b, 8c, 8d) again, provided a Supplementary Table (S4) and additional details in the methods section for the kinetic assays (corrections based on suggestions from reviewer 4).

POINT-BY-POINT RESPONSE TO REVIEWERS' COMMENTS

The reviewers' comments are in black, and our response is in blue. The highlighted versions are the changes made in the text.

REVIEWER COMMENTS

Reviewer #1 (Remarks to the Author):

The authors have invested significant effort in revising the manuscript, which has strengthened conclusions and an overall improved the study. I recommend publication of this work and have only a few minor comments below.

Fig. 3h: With only three data points and substantial overlap, it is surprising that they are statistically different?

This figure is the quantification of the Western blot data showing Vaha V5 protein expression is higher on high fat food compared to regular food during the one hour of feeding after fasting which is the experimental setting used in the manuscript. As mentioned before, two of the three replicates were run on the same gel, immunoblotted, and developed at the same time. The quantification values of these bands were close. One replicate was run at a different time. As a result, the developed films showed different densities for signal and background resulting in quantification values that were different from the other two values. Nevertheless, in a paired t test, there is difference of 48%, 39%, 20% among three biological replicates with a mean difference of 35% between the high fat and regular food groups and a two tailed P value of 0.04 which is statistically significant.

In fig. 4g and 4f, IPC activity is shown using the CaLexA system. However, the images and stains appear weak and unconvincing. It would be unfortunate to publish this otherwise nice study with such poor-quality images. Even in the control group, it is difficult to discern IPC stains. Normally, CaLexA provides clear IPC stains. I think they should include better stains at least, or even better to perform both GFP (calcium activity) and DILP2 stains, allowing for clear visualization of IPCs in both genotypes.

Our original experiment to measure IPC activity using the CaLexA system included one copy of the LexAop-CD2GFP transgene as a reporter for calcium activation. Increasing the copy number of GFP has been shown to result in more robust reporter gene expression and improve the sensitivity of the CaLexA system (Masuyama et al., (2012) J Neurogenet 26(1): 89-102). Therefore, we have now used flies bearing LexAop-CD8-GFP-2A-CD8-GFP in addition to LexAop-CD2GFP. Generating these flies in the *vaha* mutant background necessitated recombination since the *vaha* mutant and the LexAop-CD8-GFP-2A-CD8-GFP transgene are both on the second chromosome. As suggested by the reviewer, we have also included ILP2HF (epitope tagged ILP2 expressed under its own promoter) for clear visualization of the IPCs. With three copies of the reporter gene, we have been able to improve the images as shown in Fig. 4f which shows LexA GFP staining in the IPC region in control and *vaha* mutants on regular food and high fat diet. Supplementary Figs. 13a (regular food) and 13b (HFD) show all the panels including GFP, ILP2HF, DAPI and the merged images. The quantification of GFP fluorescence in the control and mutant backgrounds is provided in Fig. 4g. The following text has been included on page 13.

To test if IPC activity is compromised in the *vaha* mutant, we used the CaLexA (calcium-dependent nuclear import of LexA) system since IPC activation is accompanied by an increase in intracellular calcium levels. Here, when calcium levels rise in the IPCs, CaLexA enters the nucleus and binds the LexA operator to activate downstream CD8 GFP and CD2 GFP reporter expression which can be visualized by staining with an antiGFP antibody⁵². IPC activity thus measured in the *vaha* mutant is reduced compared to the control on regular food as well as HFD (Fig. 4f and quantification in Fig. 4g). In this experiment, ILP2HF staining was also included to mark the IPCs and Supplementary Figs. 13a and 13b show all the panels including LexA GFP, ILP2HF, DAPI and the merged images on regular food and HFD respectively.

For the sake of improved presentation, I recommend removing empty spaces in figures, such as Fig. 7, and aligning all elements neatly. Many images appear misaligned.

Following the reviewer's suggestion, we have improved presentation by removing empty spaces in figures and aligned all the images in the figures.

Reviewer #2 (Remarks to the Author):

The revised version of the manuscript is much improved, and critically, several key experiments that were before not quantitated and furnished have now been addressed, all of which strengthens the paper and its basic tenet: that the *vaha* gene is expressed in the gut, transported to the brain, and exerts its effect in the IPCs by metabolizing DAGs, most probably. Yet, I still find issue with some of the pictures provided. In figure one, besides the IPCs, there are other instances in the brain where signal is detected: Nothing is said about it, and it should be acknowledged, irrespective of whether this is the focus of the paper or not. It could be of potential interest whether other aspects of the *vaha* gene function are centered on these other cells (not explored here), which might be the groundwork for future work. Also, a close-up of the brain areas where the expression is found should also be provided. Also, the double staining with DAPI and the merge images should be bettered, to actually show both stains and see the nuclei and cytoplasm; in this case, a higher magnification view should be provided. Same commentary goes for supplementary figures 1-4 and 6, where DAPI signal cannot be seen, and the pictures without having higher magnifications, are very hard to read. Better pictures should be provided. In

general, all pictures of the supplementary figures have the same problems: they are very small, low magnification such that many time the relevant details cannot be discerned, and there is no real need to do this: higher magnifications should be provided, as well as merged images. Supplementary figure 7 shows sexual dimorphism in intestinal vaha expression, and that is not commented upon, and should be, since that could potentially be interesting, given the differences in metabolism between males and females. In general, the DAPI staining is not seen very well, and hard to define.

Below is a point-by-point response to the reviewer's comments.

In figure one, besides the IPCs, there are other instances in the brain where signal is detected: Nothing is said about it, and it should be acknowledged, irrespective of whether this is the focus of the paper or not. It could be of potential interest whether other aspects of the vaha gene function are centered on these other cells (not explored here), which might be the groundwork for future work. Also, a close-up of the brain areas where the expression is found should also be provided. Also, the double staining with DAPI and the merge images should be bettered, to actually show both stains and see the nuclei and cytoplasm; in this case, a higher magnification view should be provided.

We agree with the reviewer that although not explored in this study, the other areas in the brain where Vaha GFP signal is detected when expressed using the NP1 Gal4 driver could be of potential future interest. As suggested by the reviewer, we have added a supplementary figure (Supplementary Figure 1c) that documents the IPCs and other areas in the brain at both low and high magnification views. The following lines have been added to the main text on page 5.

While the focus of this study is the intense Vaha GFP staining in the PI region, GFP staining was also observed in the subesophageal ganglion area (marked 2 in Supplementary Fig.1c) and a few additional areas in the brain which would be of interest to characterize in a future study (marked 3, 4, 5 in Supplementary Fig. 1c).

Same commentary goes for supplementary figures 1-4 and 6, where DAPI signal cannot be seen, and the pictures without having higher magnifications, are very hard to read. Better pictures should be provided. In general, all pictures of the supplementary figures have the same problems: they are very small, low magnification such that many time the relevant details cannot be discerned, and there is no real need to do this: higher magnifications should be provided, as well as merged images.

As suggested by the reviewer, we have reworked the supplementary figures to improve them.

The original Supplementary Fig. 2 which had reproduced Main Figs. 1c and 1d (brain + VNC) at higher magnification has now been replaced with a new Supplementary Fig. 2. It shows high magnification views of the IPC region of the brain shown in main Fig. 1c when Vaha GFP is

expressed using NP1 Gal4 (Supplementary Fig. 2a). Supplementary Fig. 2b shows high magnification views of the IPC region of the brain shown in Main Fig. 1d when Vaha Δ 30GFP is expressed using NP1 Gal4 driver. These are referred to in the text on page 6.

Higher magnification views of the IPC region in Figs. 1c and 1d are shown in Supplementary Figs. 2a and Fig. 2b.

We have improved Supplementary Fig. 3a to be able to better visualize the DAPI staining. Additionally, a high magnification view of the IPC region when Vaha GFP is driven by a second gut driver (Mex Gal4) is now added as Supplementary Fig. 3b to visualize the IPC region at better resolution. This is referred to in the text on page 6.

(Supplementary Fig. 3a and high magnification view of IPCs in Supplementary Fig. 3b).

As suggested by the reviewer, in Supplementary Fig. 4a, the original view with brain and VNC is replaced by a high magnification view of the IPC region when Vaha V5 is driven by its endogenous promoter. This is referred to in the text on page 6.

(Fig. 1e and high magnification view of IPCs in Supplementary Fig. 4a).

Supplementary Figure 6 is broken down into two parts, 6a and 6b to accommodate bigger images. 6a shows NP1 Gal4 > Vaha RNAi while 6b shows data from Elav Gal4 > Vaha RNAi and Ppl Gal4 > Vaha RNAi. This is referred to in the text on page 7.

Supplementary Figs. 6a and 6b which show all the panels including brightfield, DAPI, Vaha V5, ILP2HF, and merged images).

Supplementary Figure 8 is divided in two parts 8a and 8b so bigger images could be provided. This is referred to in the text on page 8.

Supplementary figure 7 shows sexual dimorphism in intestinal vaha expression, and that is not commented upon, and should be, since that could potentially be interesting, given the differences in metabolism between males and females. In general, the DAPI staining is not seen very well, and hard to define.

We agree with the reviewer that potential sexual dimorphism in intestinal Vaha expression could be interesting. Since this aspect has not been investigated in detail, we have not included a comment in the results section describing Supplementary Figure 7. However, the following lines have been included in the text in the discussion section on page 22 to acknowledge this possibility.

Sexual dimorphism in the R4 region of the midgut has been observed particularly for carbohydrate metabolism genes¹⁵. Since *Vaha* is expressed in the R4 region, it would be of interest in the future to explore if there could be sex differences in intestinal *Vaha* gene expression.

Also, to provide better resolution, bigger images have been provided for Supplementary Figure 7.

In figure 2 it is hard, if at all clear, to distinguish the differences in intestinal nuclei, and so, the argument about the co-localization is not shown clearly. Better pictures, and close-ups at higher magnification should be provided.

In our original submission, we had speculated that enterocytes likely produce *Vaha* based on differences in size of intestinal nuclei. In the revision process, we attempted to show enterocytes produce *Vaha*. However, as mentioned in the previous response to reviewers, we could not unequivocally resolve which cell type in the gut secretes *Vaha* since we could not get the antibodies to Pdm1 (a marker for enterocytes) to work well in our hands in colocalization experiments. Therefore, we had removed the lines that suggested enterocytes of the R4 region likely secrete *Vaha* from the results section and moved them to discussion section. We had provided a few images only for reviewers' perusal that *Vaha* staining is observed in most cells with large nuclei and not seen in cells with smaller nuclei. These were not included in the manuscript. Our conclusion from Figure 2g is that *Vaha* is expressed in the R4 region of the midgut, and we have refrained from drawing any inference about differences in intestinal nuclei and expression of *Vaha* in the results section. We hope to resolve the intestinal cell type that produces *Vaha* in the future with the generation of additional tools.

Clearer pictures for figure 4f should also be provided.

A re-working of the figures is necessary, to bring them up to standards; other than that I have no comments on the paper.

We have now provided improved images for Fig. 4f. Our original experiment to measure IPC activity using the CaLexA system included one copy of the LexAop-CD2GFP transgene as a reporter for calcium activation. Increasing the copy number of GFP has been shown to result in more robust reporter gene expression and improve the sensitivity of the CaLexA system (Masuyama et al., (2012) J Neurogenet 26(1): 89-102). Therefore, we have now used flies bearing LexAop-CD8-GFP-2A-CD8-GFP in addition to LexAop-CD2GFP. Generating these flies in the *vaha* mutant background necessitated recombination, since the *vaha* mutant and the LexAop-CD8-GFP-2A-CD8-GFP transgene are both on the second chromosome. We have also included ILP2HF (epitope tagged ILP2 expressed under its own promoter) for clear visualization of the IPCs. With three copies of the reporter gene, we have been able to improve the images as shown in Fig. 4f which shows LexA GFP staining in the IPC region in control and *vaha* mutants on regular food and high fat diet. Supplementary Figs. 13a (regular food) and 13b (HFD) show

all the panels including GFP, ILP2HF, DAPI and the merged images. The quantification of GFP fluorescence in the control and mutant backgrounds is provided in Fig. 4g. The following text has been included on page 13.

To test if IPC activity is compromised in the *vaha* mutant, we used the CaLexA (calcium-dependent nuclear import of LexA) system since IPC activation is accompanied by an increase in intracellular calcium levels. Here, when calcium levels rise in the IPCs, CaLexA enters the nucleus and binds the LexA operator to activate downstream CD8 GFP and CD2 GFP reporter expression which can be visualized by staining with an antiGFP antibody⁵². IPC activity thus measured in the *vaha* mutant is reduced compared to the control on regular food as well as HFD (Fig. 4f and quantification in Fig. 4g). In this experiment, ILP2HF staining was also included to mark the IPCs and Supplementary Figs. 13a and 13b show all the panels including LexA GFP, ILP2HF, DAPI and the merged images on regular food and HFD respectively.

Reviewer #3 (Remarks to the Author):

The authors presented a substantially improved and revised version of the manuscript, which appropriately addressed all of my concerns. This is a conceptually very interesting study which will hopefully find a broad readership. Clarifying how IPC Vaha function translates to DILP-release control is a fascinating future question.

We thank the reviewer for finding our study conceptually interesting. We agree with the reviewer that how IPC Vaha function translates to DILP-release control is a fascinating question and we will strive to solve it in the future.

Reviewer #4 (Remarks to the Author):

The central finding of the manuscript presented by Singh et al. is the identification and characterization of a lipase that gets secreted by enterocytes in response to dietary cues to remotely control insulin secretion at insulin-producing cells in the brain. Although the exact mechanism behind this observation remains unknown, this observation is an important advance in our understanding of how different organs communicate to process nutrient signals. In the revised version of this manuscript, the authors address numerous issues that arose during

the review process. In summary, these experiments allowed the authors to confirm and strengthen their original conclusions.

All specific major issues from my initial revision have been answered except for one, which the authors perceive as beyond the scope of the study. Therefore, I have no major concerns.

Nevertheless, in my opinion there are still some clarifications needed in the methods section before publication.

First, the authors should describe how total DAG and MAG levels (presented in Figure 8) were calculated. If those values represent the sum of individual species detected in the LC/MS approach, this should be clarified in the method section. Since individual MAG and DAG species vary considerably according to the heat map in Figure 7 the authors should consider to add another supplementary figure showing the comprehensive panel of MAG and DAG species for each genotype.

As pointed out by the reviewer, total DAG level (Fig. 8e) and MAG level (Fig. 8f) represent the sum of individual species in control and mutant groups. These estimations, and methods were performed and provided by Metabolon (North Carolina, USA). As suggested by the reviewer, a supplementary table (Supplementary Table 4) showing DAG and MAG species that were detected by mass spectrometry for control and *vaha* mutant has been now included. The following lines have been included in the text on pages 18 and 19.

Furthermore, the increase in total DAG and decrease in MAG levels in *vaha* mutants observed in our lipidomic profiling suggests that the lipolytic cascade is interrupted at the stage of DAG hydrolysis (Figs. 8e, 8f). The individual species of DAG and MAG used to calculate the total DAG and MAG concentrations in the control and mutant groups are shown in Supplementary Table 4.

The following lines have been added to the methods section under Metabolic profiling on pages 30 and 31.

Lipids were extracted from samples via a modified Bligh-Dyer extraction using methanol/water/dichloromethane in the presence of deuterated internal standards. The extracts were concentrated under nitrogen and reconstituted in 0.25 mL of 10 mM ammonium acetate dichloromethane:methanol (50:50). The extracts were transferred to inserts and placed in vials for infusion-MS analysis, performed on a Shimadzu LC with nano PEEK tubing and the Sciex

SelexIon-5500 QTRAP. The samples were analyzed via both positive and negative mode electrospray. The 5500 QTRAP scan was performed in MRM mode with the total of more than 1,100 MRMs. Individual lipid species were quantified by taking the peak area ratios of target compounds and their assigned internal standards, then multiplying by the concentration of internal standard added to the sample. Lipid class concentrations were calculated from the sum of all molecular species within a class.

Second, the description of the enzyme activity assay is somehow confusing. The authors state that 100 μM cold substrate was used as a basis in every assay with different concentrations (25-500 nM) of ^3H -substrate added on top of that. What were the actual concentrations then? Just increasing the amount of tracer in the assay would not substantially change the overall concentration of the substrate. Please clarify. Moreover, it would be helpful to have more details about substrate preparation. Since those lipids (except MAG) are essentially insoluble, it remains elusive how they were emulsified in the buffer to have a homogenous substrate.

We thank the reviewer for raising this point. We agree with reviewer that increasing the hot substrate (tracer) would not change the overall concentration of the substrate substantially. This was a mistake on our part, we apologize for this oversight and are grateful to the reviewer for bringing this error to our notice. We have performed the assays again with each of the substrates (TAG, DAG, MAG) by varying the cold substrate concentration and using the hot substrate as a tracer. The X-axes in Figs. 8b, 8c and 8d now represent the actual concentration of the substrate used (cold + hot lipid). This resulted in the K_m in each instance increasing from nanomolar to micromolar range. The new data is now included in Figs. 8b, 8c and 8d. The overall conclusion drawn from this experiment that Vaha enzyme has the highest K_{cat}/K_m for DAG remains unchanged.

The following changes were made to the text on pages 18 and 19.

To explore the substrate preference of the enzyme, we used a well-established assay that involves incubating the enzyme with radiolabeled [^3H] triolein (TAG) or [^3H] diolein (DAG) or [^3H] monolein (MAG) as substrate⁶⁰. The lipolytic activity of the enzyme leads to the release of radiolabeled fatty acids which are resolved by thin layer chromatography, scraped, and counted. Using this assay, the substrate concentration curves for TAG, DAG and MAG were generated and the kinetic parameters K_m , k_{cat} and k_{cat}/K_m were calculated (Figs. 8b, 8c and 8d

respectively). The purified lipase showed highest affinity for DAG with a K_m of $17.95 \times 10^{-6}M$, followed by TAG with a K_m of $19.71 \times 10^{-6}M$, and then MAG with a K_m of $25 \times 10^{-6}M$. The ratio of k_{cat}/K_m is a useful indicator of the relative efficiency of the enzyme when it has more than one substrate. Based on the observation that the enzyme has highest k_{cat}/K_m values for DAG, Vaha is likely a DAG lipase (Fig. 8c).

As suggested by the reviewer, details about substrate preparation have been added to the methods under the section, Determination of Vaha enzyme activity using different substrates. The following text is included on page 33.

Determination of Vaha enzyme activity using different substrates

Vaha activity was measured using 3 different substrates - [3H] TAG (Perkin Elmer NET431001MC), [3H] DAG (American Radiolabeled Chemicals ART0643) and [3H] MAG (American Radiolabeled Chemicals ART1158). The cold substrates were from Cayman Chemical Company TAG 26871, DAG 26896, MAG 16537. The cold substrates were emulsified with 5 mM CHAPS and sonicated for 5-10 sec using a probe sonicator (Ultrasonic Processor, Cole-Parmer). To determine substrate specificity of lipase, we used 10 μM -100 μM of TAG/DAG/MAG and 0.1% of respective radiolabeled [3H] substrate in the total substrate concentration. The FPLC purified enzyme was added to the assay mixture containing 50 mM Tris-HCl (pH 7.4), cold substrate, and [3H] substrate in a total volume of 200 μl . The reaction mixture was incubated at 30°C for 15 min and terminated by adding acidified water and 0.6 ml of chloroform: methanol (1:2 v/v). The lipids were extracted using the Bligh and Dyer method and the extracted lipids were separated on TLC silica gel plates (Sigma, 1055540001) using the petroleum ether: diethyl ether: acetic acid (70:30:1 v/v) solvent system. The released fatty acid was quantified in a scintillation counter (TRI-CARB 4910TR, PerkinElmer).

REVIEWERS' COMMENTS

Reviewer #1 (Remarks to the Author):

The authors have made commendable efforts in revising the manuscript, notably enhancing the quality of their stains, which has significantly strengthened the study's conclusions and overall quality. I am pleased to recommend the publication of this work.

Reviewer #2 (Remarks to the Author):

The current version of the paper is much improved over the previous version, solving some of the main issues raised before: e.g., the figures are much improved, and the details provided allow assessment of the results. The paper, overall, is now sound and presents exciting and new data in the field, that should be considered for publication.

I will still ask the authors to answer these minor points:

a) on page 21 the authors refer to eiger and TNF-alpha as two different genes, and to my knowledge, eiger is the sole TNF-alpha homolog in flies, at least as referred to in Flybase. Please clarify or correct.

b) On page 22 the authors state the enterocytes secrete vaha; I think that although likely, it is still a speculation at this point, and the authors should either refer to it in such a way, or provide evidence to substantiate the claim.

c) PCR cycling parameters should be furnished in the methods section of the paper for the different PCR protocols.

d) Vaha mode of action in figure 8i should be described and explained in the corresponding figure legend.

e) In supplementary figures 7 and 8 additional sites within the gut (especially in the hindgut region) show expression of vaha transcript. These are not mentioned or discussed in the

text, and they should be acknowledged and also discussed. It would be of interest to explore whether this expression is significant for the phenotype(s), especially in reference of the source of secreted vaha (see point b).

Reviewer #4 (Remarks to the Author):

In the revised version of the manuscript the authors present improved enzyme activity assays and add a detailed table of MAG and DAG species. These additions/improvements add to the comprehensiveness of the manuscript and further strengthen the conclusion of the authors. My questions have been fully addressed. Overall, the authors did a great job and provide an exciting piece of research.

REVIEWERS' COMMENTS

Reviewer #1 (Remarks to the Author):

The authors have made commendable efforts in revising the manuscript, notably enhancing the quality of their stains, which has significantly strengthened the study's conclusions and overall quality. I am pleased to recommend the publication of this work.

We appreciate the positive feedback from the reviewer recommending our work for publication. We would like to thank the reviewer for the comments which have helped us strengthen the manuscript.

Reviewer #2 (Remarks to the Author):

The current version of the paper is much improved over the previous version, solving some of the main issues raised before: e.g., the figures are much improved, and the details provided allow assessment of the results. The paper, overall, is now sound and presents exciting and new data in the field, that should be considered for publication.

We appreciate that the reviewer approves of our manuscript and would like to thank the reviewer for the comments which have helped us improve the manuscript.

I will still ask the authors to answer these minor points:

a) on page 21 the authors refer to eiger and TNF-alpha as two different genes, and to my knowledge, eiger is the sole TNF-alpha homolog in flies, at least as referred to in Flybase. Please clarify or correct.

We thank the reviewer for pointing this out. We have now corrected the sentence on page 21 as follows:

The negative regulators include the *Drosophila* TNF- α homolog Eiger, and insulin-like growth factor Impl2^{33,70}.

b) On page 22 the authors state the enterocytes secrete vaha; I think that although likely, it is still a speculation at this point, and the authors should either refer to it in such a way, or provide evidence to substantiate the claim.

We agree with the reviewer that we have not been able to provide proof that Vaha is secreted by enterocytes. We have modified the sentence on page 22 as follows:

Among the different cell types in the gut, the intestinal stem cells, hormone secreting enteroendocrine cells, digestive and absorptive enterocytes, the latter are the predominant cell type and have large nuclear size. Based on these two criteria, **we speculate**, the enterocytes of the R4 region of the midgut secrete Vaha.

c) PCR cycling parameters should be furnished in the methods section of the paper for the different PCR protocols.

We have added PCR cycling parameters in the methods section on page 34 as follows:

Quantitative reverse transcription PCR

To measure changes in gene expression, total RNA was extracted from 10 whole flies, 30 guts, or 40 heads by TRIZOL RNA isolation protocol. Total RNA was extracted from 10 brains and 10 CNS using PicoPure RNA isolation kit (ThermoFisher Scientific). The isolated RNA samples were reverse transcribed by transcriptor first strand cDNA synthesis kit (Roche). Gene expression was analyzed by real-time qPCR using the iTaq universal SYBR Green supermix (BIO-RAD) in a QuantStudio™ 5 Real-Time PCR System (Applied Biosystems). **The RT-qPCR cycling conditions are as follows: initial denaturation at 95°C/5 min followed by denaturation (95°C/15 sec) and annealing and extension (60°C/60 sec) for 40 cycles.**

d) Vaha mode of action in figure 8i should be described and explained in the corresponding figure legend.

We apologize for this omission. The following line has been included in the legend to figure 8i on page 53.

i A model of Vaha-mediated inter-organ communication in ILP2 release.

Vaha (dark blue circles) secreted from R4 region of the midgut (light blue) concentrates in IPCs (green) in a process requiring LTP (black circles) from the fat body (yellow) to regulate ILP2 release.

e) In supplementary figures 7 and 8 additional sites within the gut (especially in the hindgut region) show expression of vaha transcript. These are not mentioned or discussed in the text, and they should be acknowledged and also discussed. It would be of interest to explore whether this expression is significant for the phenotype(s), especially in reference of the source of secreted vaha (see point b).

We agree with the reviewer that additional staining is observed in the HCR images of adult and larval gut shown in supplementary figures 7 and 8b respectively. We are not sure that this staining is specific to *Vaha* transcript since it does not appear cellular like the R4 region in high magnification images. Nevertheless, we have added the following sentence on page 8.

In addition to the midgut, both the adult and larval guts show some staining in the hind gut region for *Vaha*, the specificity of which requires further exploration (Supplementary Figs. 7 and 8b).

Reviewer #4 (Remarks to the Author):

In the revised version of the manuscript the authors present improved enzyme activity assays and add a detailed table of MAG and DAG species. These additions/improvements add to the comprehensiveness of the manuscript and further strengthen the conclusion of the authors. My questions have been fully addressed. Overall, the authors did a great job and provide an exciting piece of research.

We appreciate the positive feedback from the reviewer. We would like to thank the reviewer for the comments which have helped us strengthen the manuscript.